# Hierarchical Multi-Stage Recovery Framework for Kronecker Compressed Sensing

**Yanbin He, Geethu Joseph**
Signal Processing Systems Group, Delft University of Technology
`{y.he-1,g.joseph}@tudelft.nl`

## Abstract

In this paper, we study the Kronecker compressed sensing problem, which focuses on recovering sparse vectors using linear measurements obtained using the Kronecker product of two or more matrices. We first introduce the *hierarchical view* of the Kronecker compressed sensing, showing that the Kronecker product measurement matrix probes the sparse vector from different levels, following a block-wise and hierarchical structure. Leveraging this insight, we develop a versatile multi-stage sparse recovery algorithmic framework and tailor it to three different sparsity models: standard, hierarchical, and Kronecker-supported. We further analyze the restricted isometry property of Kronecker product matrices under different sparsity models, and provide theoretical recovery guarantees for our multi-stage algorithm. Simulations demonstrate that our method achieves comparable recovery performance to other state-of-the-art techniques while substantially reducing runtime owing to the hierarchical, multi-stage recovery process.

## 1 Introduction

Kronecker compressed sensing (KCS) is a measurement framework that employs the Kronecker product of multiple factor matrices as a measurement matrix, capturing multidimensional signal structure while reducing measurement complexity. It appears in many acquisition systems, such as sensor arrays in communication systems (He & Joseph, 2025a) or separable filters in imaging (Friedland et al., 2014). We focus on the general KCS problem with canonical form,

$$\boldsymbol{y} = \boldsymbol{H}\boldsymbol{x} + \boldsymbol{n} = (\boldsymbol{H}_I \otimes \boldsymbol{H}_{I-1} \otimes \cdots \otimes \boldsymbol{H}_1)\,\boldsymbol{x} + \boldsymbol{n} = \left(\otimes_{i=I}^{1}\boldsymbol{H}_i\right)\boldsymbol{x} + \boldsymbol{n}. \tag{1}$$

Here, $\boldsymbol{x} \in \mathbb{R}^{\bar{N}}$ is the *unknown* sparse vector and $\boldsymbol{y} \in \mathbb{R}^{\bar{M}}$ is the noisy measurements via a *known* measurement matrix $\boldsymbol{H} = \otimes_{i=I}^{1}\boldsymbol{H}_i$, where each factor matrix $\boldsymbol{H}_i \in \mathbb{R}^{M_i \times N_i}$ has *full row rank*.

A key challenge in solving Equation 1 is the high dimensionality of the multidimensional signal $\boldsymbol{x}$. It grows rapidly with both the number and size of factor matrices $\boldsymbol{H}_i$, e.g., $\mathcal{O}(N^I)$ if $N_i = \mathcal{O}(N)$. Another challenge is exploiting sparsity patterns as prior knowledge. Beyond simple sparsity, the nonzero elements in $\boldsymbol{x}$ often exhibit more complex but regulated patterns. We consider three prevalent models. The first model is the standard sparsity, where the nonzero entries can be positioned arbitrarily. This model is ubiquitous and has been applied to various fields, such as image processing (Duarte & Baraniuk, 2010; Li & Bernal, 2017; Zhao et al., 2019), system identification (Sun et al., 2022; Yuan et al., 2019), regression (Ament & Gomes, 2021), and communications (Berger et al., 2010; Xiao et al., 2024). The second model, hierarchical sparsity, considers a vector $\boldsymbol{x}$ partitioned into blocks at multiple levels with sparsity structured across these levels. For example, in massive machine-type communication (Wunder et al., 2017; Roth et al., 2018; 2020), only a subset of devices are active (device-level sparsity), and each active device sends a sparse signal, forming a two-level hierarchical structured sparsity pattern on $\boldsymbol{x}$. The third model, Kronecker-supported sparsity (or block tensor sparsity) (He & Joseph, 2025a; 2023; Caiafa & Cichocki, 2013; Zhao et al., 2019; Boyer & Haardt, 2016), assumes the support of $\boldsymbol{x}$ is the Kronecker product of multiple binary support vectors. This pattern arises in radar imaging and wireless communications, where signals are separable across dimensions (He & Joseph, 2023; Xu et al., 2022; He & Joseph, 2025d). Motivated by varied sparsity patterns, we focus on efficient methods for KCS with structured sparsity.

This paper introduces a novel *hierarchical view* on KCS, showing how its dimension-wise measuring structure can be used to design and analyze efficient recovery methods to exploit structured sparsity effectively. Our main contributions are as follows:

- *Hierarchical View*: We establish that when measuring via Kronecker product matrices, each factor matrix in the Kronecker product captures the vector at a distinct hierarchical level. It provides a unified perspective for handling different sparsity models within a single framework.

- *Unified Algorithm*: We design a multi-stage sparse recovery algorithm using the hierarchical view. By leveraging the Kronecker structure of $\boldsymbol{H}$ through tensor operation and investigating the underlying structure, our method achieves a significant complexity reduction, e.g., reducing from $\mathcal{O}((MN)^I)$ (He & Joseph, 2025a) to $\mathcal{O}(MN^I)$ regarding Kronecker-supported sparse vector recovery, and accommodates the mentioned sparsity patterns within a single, flexible framework.

- *Theoretical Guarantees*: We establish a unified restricted isometry property (RIP) analysis for KCS covering the standard, hierarchical, and Kronecker-supported sparsity. It proves that sparsity at each hierarchical level, rather than total sparsity, drives the recovery. Our result improves the RIP-based bound for KCS with standard sparsity and provides a cohesive understanding of structured sparsity. We also provide a RIP-based recovery guarantee for our unified algorithm.

**Related works:** The Kronecker product measurement matrix is introduced for compressed imaging in Rivenson & Stern (2009). KCS is formalized in Duarte & Baraniuk (2011a) tailored to hyperspectral imaging, with an RIP analysis for KCS with standard sparsity (Duarte & Baraniuk, 2011a;b). It bounds the restricted isometry constant (RIC) of the Kronecker product using the RIC of factor matrices $\boldsymbol{H}_i$. However, the recovery algorithm fails to leverage the Kronecker structure in $\boldsymbol{H}$. To leverage this structure, Kronecker orthogonal matching pursuit (KroOMP) (Caiafa & Cichocki, 2013) adopts tensor operations. Nonetheless, it still incurs a high complexity of $\mathcal{O}(N^I)$, and lacks theoretical analysis. Friedland et al. (2014; 2015) presents two algorithms: one uses tensor unfolding for sequential recovery in dimension, and the other uses approximate Tucker decomposition to recover along each dimension for compressible image and video representation and recovery. Still, both approaches are limited to standard sparsity. Li & Bernal (2017) decomposes the unfolding-based approach into multiple *independent* subproblems for hyperspectral imaging. Yet, it fails to exploit joint sparsity patterns and is not immediately extendable to other sparsity patterns.

KCS with structured sparse recovery is also investigated in the literature. For *hierarchically sparse* vectors, Roth et al. (2020) discusses the hierarchical hard thresholding pursuit (HiHTP), adapting classic hard thresholding pursuit (HTP) with a tailored RIP and coherence analysis for channel estimation for massive multiple-input multiple-output systems (Wunder et al., 2019). However, it fails to incorporate the Kronecker structure in $\boldsymbol{H}$, leading to higher computational costs. For *Kronecker-supported sparsity*, both greedy and Bayesian methods have been explored. An orthogonal matching pursuit (OMP)-based algorithm offers reduced complexity (Caiafa & Cichocki, 2012; Caiafa & Cichocki, 2013) but performs poorly in noisy settings (He & Joseph, 2025a). Bayesian algorithms, designed for applications such as hyperspectral image processing (Zhao et al., 2019) and wireless communication (He & Joseph, 2025a; Chang & Su, 2021; Xu et al., 2022), use a structured prior distribution. They suffer from poor generalization and high complexity (He & Joseph, 2025b). Besides, both OMP-based and Bayesian algorithms lack theoretical guarantees. Recently, He & Joseph (2025c) provides an algorithm and RIP analysis for KCS for the $I = 2$ case. However, the analysis is decoupled from the algorithm. Also, it relies heavily on specific matrix properties, making the generalization to higher orders ($I > 2$) nontrivial.

To summarize, existing approaches reveal several literature gaps. First, KCS methods mostly ignore the structures of $\boldsymbol{H}$, relying on generic solvers, while our method is specifically designed to leverage the Kronecker structure through tensor operations. Second, current methods are largely tailored to a single sparsity pattern and cannot be generalized, whereas our work provides a unified framework for multiple patterns. Third, many methods suffer from high computational complexity, while our approach is efficient and low-complexity. Besides, no prior work offers a unified RIP analysis of Kronecker-structured matrices across various sparsity patterns, nor a recovery framework for different sparsity patterns with RIP-based guarantees, which are our central theoretical contributions.

**Notation and tensor preliminaries:** We use $[I]$ to denote the set $\{1, 2, \cdots, I\}$ for any scalar $I$ and $\boldsymbol{I}_N$ to denote the $N \times N$ identity matrix. The symbols $\otimes$ and $\times_j$ denote Kronecker and $j$th mode product, respectively. The $j$th mode unfolding $\boldsymbol{T}_{(j)}$ of tensor $\mathsf{T} \in \mathbb{R}^{N_1 \times N_2 \times \cdots \times N_I}$ is $\left[\boldsymbol{T}_{(j)}\right]_{n_j, k} =$

$[\mathsf{T}]_{n_1, n_2, \ldots, n_I}$ for $j \in [I]$ with $k = 1 + \sum_{\ell=1, \ell \neq j}^{I} \left( \prod_{p=1, p \neq j}^{\ell-1} N_p \right) (n_l - 1)$, with $n_j \in [N_j]$. Also, $[\boldsymbol{T}_{(j)}]_{n_j, k}$ is $(n_j, k)$th matrix entry, and $[\mathsf{T}]_{n_1, n_2, \cdots, n_I}$ is the $(n_1, \cdots, n_I)$th tensor entry. The $i$th mode product of $\boldsymbol{D}_i \in \mathbb{R}^{N_i \times M_i}$ with $\mathsf{T}$ is $\mathsf{M} = \mathsf{T} \times_i \boldsymbol{D}_i \in \mathbb{R}^{M_1 \times \cdots \times M_{i-1} \times N_i \times M_{i+1} \times \cdots \times M_I}$. The $i$th mode unfolding of $\mathsf{M}$ is $M_{(i)} = \boldsymbol{D}_i \boldsymbol{T}_{(i)}$ (Kolda & Bader, 2009).

## 2 HIERARCHICAL VIEW OF THE KRONECKER-STRUCTURED MEASURING

Our hierarchical view builds on the Kronecker structure in Equation 1, interpreting the measurement matrix as probing the signal's sparsity across multiple block-wise and hierarchical levels. To illustrate this, we first introduce the hierarchical block partition of a sparse vector $\boldsymbol{x} \in \mathbb{R}^{\bar{N}}$.

**Hierarchical partition:** We first partition $\boldsymbol{x}$ in Equation 1 into $N_I$ equal-length blocks, denoting the $I$th level blocks as $\{\boldsymbol{x}_{(n_I)}\}_{n_I=1}^{N_I} \in \mathbb{R}^{\prod_{i=1}^{I-1} N_i}$. Each $\boldsymbol{x}_{(n_I)}$ is further partitioned into $N_{I-1}$ blocks, denoted as $(I-1)$th level blocks $\{\boldsymbol{x}_{(n_{I-1}, n_I)}\}_{n_{I-1}=1}^{N_{I-1}} \in \mathbb{R}^{\prod_{i=1}^{I-2} N_i}$. We continue until we reach blocks of length $N_1$ at the second level. The first-level blocks are the individual entries of $\boldsymbol{x}$.

For brevity, we use $\boldsymbol{x}_{\mathrm{n_j}}$ to denote a block in the $j$th level with length $\prod_{i=1}^{j-1} N_i$ and encapsulation $\mathrm{n_j} := (n_j, \cdots, n_{I-1}, n_I)$. An encapsulation $\mathrm{n_j} := (n_j, \cdots, n_{I-1}, n_I)$ can be viewed as a coordinate for blocks in this hierarchical block structure. Also, set $[\![\boldsymbol{x}_{\mathrm{n_j}}]\!]$ contains all $N_j$ *child blocks* that share the same parent block at the level $j+1$ as that of $\boldsymbol{x}_{\mathrm{n_j}}$. We illustrate a hierarchical partition for $\boldsymbol{x} \in \mathbb{R}^{40}$ in Figure 1, where $[\![\boldsymbol{x}_{(1,3)}]\!] = [\![\boldsymbol{x}_{(2,3)}]\!] = \{\boldsymbol{x}_{(1,3)}, \boldsymbol{x}_{(2,3)}\}$ as they share the parent $\boldsymbol{x}_{(3)}$.

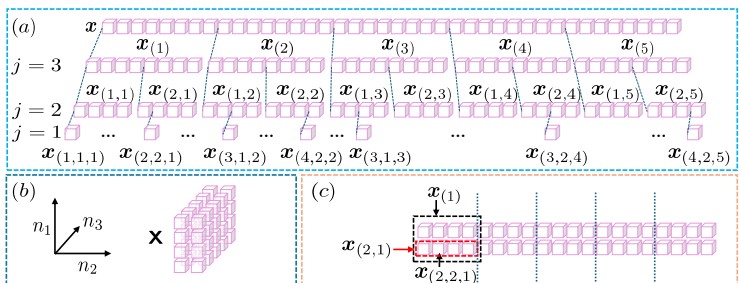

Figure 1: (a) Hierarchical partition for $\boldsymbol{x} \in \mathbb{R}^{40}$ with $I = 3$, $N_3 = 5$, $N_2 = 2$, $N_1 = 4$, and $\bar{N} = 40$. (b) Reordered tensor $\mathsf{X}$. (c) Mode unfolding $\boldsymbol{X}_{(2)}$ and the relation between the $n_{I-1}$th row within the $n_I$th column block and the $(I-1)$th level child block $\boldsymbol{x}_{n_{I-1}}$ with $n_{I-1} = (2, 1)$ and $I = 3$.

**Hierarchical view:** We first focus on the noiseless version of Equation 1 reformulated using tensors,

$$\mathsf{T} := \mathsf{Y} = \mathsf{X} \times_1 \boldsymbol{H}_1 \cdots \times_I \boldsymbol{H}_I,$$

where the first mode unfolding satisfy $\text{vec}(\boldsymbol{X}_{(1)}) = \boldsymbol{x}$ and $\text{vec}(\boldsymbol{T}_{(1)}) = \text{vec}(\boldsymbol{Y}_{(1)}) = \boldsymbol{y}$. Unfolding $\mathsf{T}$ on the $I$th mode leads to

$$\boldsymbol{T}_{(I)} = \boldsymbol{H}_I \boldsymbol{X}_{(I)} \left( \otimes_{i=I-1}^{1} \boldsymbol{H}_i^{\top} \right) = \boldsymbol{H}_I \boldsymbol{U}_I \in \mathbb{R}^{M_I \times \prod_{i=I-1}^{I} M_i}.$$

Here, $\boldsymbol{U}_I = \boldsymbol{X}_{(I)} \left( \otimes_{i=I-1}^{1} \boldsymbol{H}_i^{\top} \right) \in \mathbb{R}^{N_I \times \prod_{i=I-1}^{I} M_i}$ and $\boldsymbol{X}_{(I)} \in \mathbb{R}^{N_I \times \prod_{i=I-1}^{I} N_i}$ whose $n_I$th row is the $I$th level block $\boldsymbol{x}_{\mathrm{n_I}}$ with $\mathrm{n_I} = (n_I)$. Therefore, matrix $\boldsymbol{H}_I$ acts on $\boldsymbol{U}_I$, and a zero row in $\boldsymbol{U}_I$ indicates that the corresponding $I$th level block is entirely zero. Hence, matrix $\boldsymbol{H}_I$ captures the sparsity pattern of the $I$th-level blocks.

For the $(I-1)$th level, we fold $\boldsymbol{U}_I$ into a new tensor $\mathsf{T}$, whose $I$th mode unfolding $\boldsymbol{T}_{(I)} = \boldsymbol{U}_I$, as

$$\mathsf{T} = \mathsf{X} \times_1 \boldsymbol{H}_1 \cdots \times_{I-1} \boldsymbol{H}_{I-1} \times_I \boldsymbol{I}_{N_I}.$$

Unfolding $\mathsf{T}$ along its $(I-1)$th mode gives

$$\boldsymbol{T}_{(I-1)} = \boldsymbol{H}_{I-1} \boldsymbol{X}_{(I-1)} \left( \boldsymbol{I}_{N_I} \otimes \left( \otimes_{i=I-2}^{1} \boldsymbol{H}_i^{\top} \right) \right) = \boldsymbol{H}_{I-1} \boldsymbol{U}_{I-1} \in \mathbb{R}^{M_{I-1} \times N_I \prod_{i=I-2}^{I} M_i}.$$

Here, $\boldsymbol{X}_{(I-1)} \in \mathbb{R}^{N_{I-1} \times \bar{N}/N_{I-1}}$ has $N_I$ column blocks, with $n_I$th block corresponding to $\boldsymbol{x}_{\mathrm{n_I}}$ with $\mathrm{n_I} = (n_I)$. Within the $n_I$th column block, the $n_{I-1}$th row is the $(I-1)$th level child block $\boldsymbol{x}_{n_{I-1}}$

with $n_{I-1} = (n_{I-1}, n_I)$, as illustrated in Figure 1c. The Kronecker product $\boldsymbol{I}_{N_I} \otimes \left(\otimes_{i=I-2}^1 \boldsymbol{H}_i^\top\right)$ is a block matrix and preserves the column block structure in $\boldsymbol{U}_{I-1}$. Column blocks of $\boldsymbol{U}_{I-1}$ are associated with $I$-level blocks, and the rows of a column block correspond to the $(I-1)$ level blocks. Hence, the zero rows in each column block of $\boldsymbol{U}_{I-1}$ indicate that the corresponding $(I-1)$th level blocks are entirely zero. Therefore, $\boldsymbol{H}_{I-1}$ captures the sparsity pattern at the $(I-1)$th-level blocks. For a general $j$th level, we define $\mathsf{T} = \mathsf{X} \times_1 \boldsymbol{H}_1 \times_2 \boldsymbol{H}_2 \cdots \times_j \boldsymbol{H}_j \times_{j+1} \boldsymbol{I}_{N_{j+1}} \cdots \times_I \boldsymbol{I}_{N_I}$, and

$$\boldsymbol{T}_{(j)} = \boldsymbol{H}_j \boldsymbol{X}_{(j)} \left(\boldsymbol{I}_{\prod_{i=I}^{j+1} N_i} \otimes \left(\otimes_{i=j-1}^1 \boldsymbol{H}_i\right)\right)^\top \in \mathbb{R}^{M_j \times \prod_{i=I}^{j+1} N_i \prod_{i=j-1}^1 M_i},$$

is its $j$th unfolding. Similar to the column block structure at $(I-1)$th level, we have the following.

**Lemma 1.** *Consider a sparse tensor $\mathsf{X}$ reordered from a sparse vector $\boldsymbol{x}$ such that $\mathrm{vec}(\boldsymbol{X}_{(1)}) = \boldsymbol{x}$. For the $j$th mode unfolding of $\mathsf{X}$, i.e., $\boldsymbol{X}_{(j)}$, and with full row rank $\boldsymbol{H}_i$'s, the matrix*

$$\boldsymbol{U}_j := \boldsymbol{X}_{(j)} \left(\boldsymbol{I}_{\prod_{i=I}^{j+1} N_i} \otimes \left(\otimes_{i=j-1}^1 \boldsymbol{H}_i\right)\right)^\top \in \mathbb{R}^{N_j \times \prod_{i=I}^{j+1} N_i \prod_{i=j-1}^1 M_i},$$

*can be divided into $\prod_{i=I}^{j+1} N_i$ column blocks. Each block is indexed by an encapsulation $\mathrm{n_{j+1}}$ with $\mathrm{n_{j+1}} = (n_{j+1}, \cdots, n_I)$ for $n_k \in [N_k]$ for $k = j+1, \ldots, I$. The number of nonzero rows in a column block indexed by $\mathrm{n_{j+1}}$ equals the number of nonzero blocks in $[\![\boldsymbol{x}_{\mathrm{n_j}}]\!]$ with $\mathrm{n_j} = (n_j, n_{j+1}, \cdots, n_I)$.*

Lemma 1 implies that matrix $\boldsymbol{H}_j$ actually captures the sparsity at the $j$th level blocks, which we refer to as the *hierarchical view* of KCS. The above perspective can also be interpreted directly from Equation 1. The Kronecker product matrix $\boldsymbol{H}$ has a recursive column-block structure: each block of columns is obtained by taking the Kronecker product of a column of $\boldsymbol{H}_I$ with $\otimes_{i=I-1}^1 \boldsymbol{H}_i$, which itself has a column block structure. This recursive structure aligns with the hierarchical partition block of $\boldsymbol{x}$. Hence, in this hierarchical framework, factor matrices $\{\boldsymbol{H}_i\}_{i=I}^1$ operate at different levels: for any $p, q$ with $p > q$, $\boldsymbol{H}_q$ first measures each $q$th level block of $\boldsymbol{x}$, the resulting measurements of all blocks are then processed by $\boldsymbol{H}_p$, which captures sparsity at a higher level.

## 3 MULTI-STAGE SPARSE RECOVERY ALGORITHM

We aim to recover $\boldsymbol{x}$ in Equation 1 from noisy measurement $\boldsymbol{y}$, given $\{\boldsymbol{H}_i\}_{i=1}^I$. Guided by the hierarchical view in Section 2, we next present a recovery framework that handles each $\boldsymbol{H}_i$ sequentially. We formally define the following three considered sparsity models.

**Sparsity 1** (Standard sparsity). *A vector $\boldsymbol{x} \in \mathbb{R}^{\bar{N}}$ is $s$ sparse if $\boldsymbol{x}$ contains at most $s$ nonzeros.*

**Sparsity 2** (Hierarchical sparsity). *A vector $\boldsymbol{x} \in \mathbb{R}^{\bar{N}}$ is $\boldsymbol{s}$ hierarchically sparse with $\boldsymbol{s} := (s_I, s_{I-1}, \cdots, s_1)$ if it has a hierarchical partition defined by $\{N_j\}_{j=1}^I$, and at each level $j \in [I]$, every set $[\![\boldsymbol{x}_{\mathrm{n_j}}]\!]$ contains at most $s_i$ nonzero blocks.*

**Sparsity 3** (Kronecker-supported sparsity). *A vector $\boldsymbol{x} \in \mathbb{R}^{\bar{N}}$ is $\boldsymbol{s}$ Kronecker supported sparse if its support is the Kronecker product of $s_j$ sparse support vectors $\boldsymbol{b}_j \in \{0, 1\}^{N_j}$ for $j \in [I]$.*

We note that the Kronecker-supported sparsity is a special case of hierarchical sparsity, where at each level $j \in [I]$, the $s_j$ nonzero blocks $\boldsymbol{x}_{\mathrm{n_j}}$ share the same support. See Appendix B for illustration.

Our framework first solves for $\boldsymbol{U}_I = \boldsymbol{X}_{(I)} \left(\otimes_{i=I-1}^1 \boldsymbol{H}_i\right)^\top$ from unfolding along $I$th mode using

$$\boldsymbol{T}_{(I)} := \boldsymbol{Y}_{(I)} = \boldsymbol{H}_I \boldsymbol{U}_I + \boldsymbol{N}_{(I)}. \tag{2}$$

Here, $\boldsymbol{U}_I$ exhibits a row sparsity pattern where a zero row in $\boldsymbol{U}_I$ corresponds to an all-zero $I$th level block $\boldsymbol{x}_{\mathrm{n_I}}$. Thus, recovering $\boldsymbol{U}_I$ from Equation 2 is a multiple measurement vector (MMV) problem and solved using MMV algorithms such as simultaneous OMP (SOMP), simultaneous iterative hard thresholding (SIHT), simultaneous HTP (SHTP), or MMV sparse Bayesian learning (MMV-SBL).

Let the estimate of $\boldsymbol{U}_I$ be $\tilde{U}_I$ with error $\boldsymbol{E}_I$ modeling the estimation error and residual noise, $\tilde{U}_I = \boldsymbol{U}_I + \boldsymbol{E}_I$. In the second step, we treat $\tilde{U}_I$ as the noisy measurement and $\boldsymbol{E}_I$ as noise, reorder them into tensor $\mathsf{T}$ and $\mathsf{N}$ such that $\mathsf{T}_{(I)} = \tilde{U}_I$ and $\mathsf{N}_{(I)} = \boldsymbol{E}_I$, to obtain $\mathsf{T} = \mathsf{X} \times_1 \boldsymbol{H}_1 \cdots \times_{I-1} \boldsymbol{H}_{I-1} \times_I \boldsymbol{I}_{N_I} + \mathsf{N}$. Unfolding $\mathsf{T}$ along its $(I-1)$th mode as

$$\boldsymbol{T}_{(I-1)} = \boldsymbol{H}_{I-1} \boldsymbol{U}_{I-1} + \boldsymbol{N}_{(I-1)}. \tag{3}$$

For standard and hierarchical sparsity models, the supports of different $(I-1)$th level blocks of $\boldsymbol{x}$ are different. By Lemma 1, zero $(I-1)$th level blocks leads to the zero rows in each column block in $\boldsymbol{U}_{I-1}$, making it a concatenation of $N_I$ row sparse matrices $[\boldsymbol{U}_{I-1}]_{\mathrm{n_I}} := [\boldsymbol{X}_{(I-1)}]_{\mathrm{n_I}} \left(\otimes_{i=I-2}^{1} \boldsymbol{H}_i\right)^{\top}$ for $\mathrm{n_I} = (n_I)$ and $n_I \in [N_I]$. We thus partition Equation 3 into $N_I$ independent MMV problems as

$$[\boldsymbol{T}_{(I-1)}]_{\mathrm{n_I}} = \boldsymbol{H}_{I-1}[\boldsymbol{U}_{I-1}]_{\mathrm{n_I}} + [\boldsymbol{N}_{(I-1)}]_{\mathrm{n_I}},$$

and solve them (sequentially or in parallel) using MMV solvers. Concatenating estimates $\tilde{\boldsymbol{U}}_{I-1} := [[\tilde{\boldsymbol{U}}_{I-1}]_1, [\tilde{\boldsymbol{U}}_{I-1}]_2, \cdots, [\tilde{\boldsymbol{U}}_{I-1}]_{N_I}]$ gives the final solution, where $[\tilde{\boldsymbol{U}}_{I-1}]_{\mathrm{n_I}} = [\boldsymbol{U}_{I-1}]_{\mathrm{n_I}} + [\boldsymbol{E}_{I-1}]_{\mathrm{n_I}}$. However, for the Kronecker-supported sparsity, Equation 3 is a single MMV problem because the support is common across the $(I-1)$th level blocks.

Generalizing, for $j$th mode unfolding step, with measurement $\tilde{\boldsymbol{U}}_{j+1}$ from the previous step,

$$\tilde{\boldsymbol{U}}_{j+1} = \boldsymbol{U}_{j+1} + \boldsymbol{E}_{j+1} = \boldsymbol{X}_{(j+1)} \left(\boldsymbol{I}_{\prod_{i=I}^{j+2} N_i} \otimes \left(\otimes_{i=j}^{1} \boldsymbol{H}_i\right)\right)^{\top} + \boldsymbol{E}_{j+1}. \qquad (4)$$

We unfold the measurement tensor formed from $\tilde{\boldsymbol{U}}_{j+1}$ along its $j$th mode as

$$\boldsymbol{T}_{(j)} = \boldsymbol{H}_j \boldsymbol{U}_j + \boldsymbol{N}_{(j)}. \qquad (5)$$

Lemma 1 reduces Equation 5 to $\prod_{i=I}^{j+1} N_i$ independent MMV problems for standard and hierarchical sparsity. Sparsity varies across MMVs for the standard model (defined by total sparsity rather than level-wise sparsity) but remains identical in the hierarchical model. For Kronecker-supported sparsity, Equation 5 is a single MMV due to shared block support. While mixed models with single and multiple MMVs at different levels are possible, we focus on these three main cases for brevity, leading to the Multi-Stage Recovery (MSR) algorithm, summarized in Algorithm 1.

---

**Algorithm 1** Multi-Stage Recovery (MSR)

---

**Input:** Measurement $\boldsymbol{y}$, dictionaries $\{\boldsymbol{H}_i\}_{i=1}^{I} \in \mathbb{R}^{M_i \times N_i}$
1: Fold $\boldsymbol{y}$ to $\mathsf{Y}$ according to the dimensions of dictionaries $\{\boldsymbol{H}_i\}_{i=1}^{I}$, and initialize $\mathsf{T} = \mathsf{Y}$
2: **for** $j = I, I-1, \cdots, 1$ **do**
3:     Obtain the $j$th mode unfolding of $\mathsf{T}$, i.e., $\boldsymbol{T}_{(j)}$
4:     Solve Equation 5 for $\boldsymbol{U}_j$ via a compressed sensing algorithm to get estimate $\tilde{\boldsymbol{U}}_j$
5:     Fold $\tilde{\boldsymbol{U}}_j$ back to $\mathsf{T}$ such that the $j$th mode unfolding of $\mathsf{T}$, i.e., $\boldsymbol{T}_{(j)}$ is $\tilde{\boldsymbol{U}}_j$
6: **end for**
**Output:** Estimated sparse vector $\hat{\boldsymbol{x}} = \mathrm{vec}(\tilde{\boldsymbol{U}}_1)$

---

**Complexity:** We compare the complexity of MSR variants with existing methods for each sparsity model, assuming Equation 5 is solved sequentially, and $M_i = \mathcal{O}(M)$, $N_i = \mathcal{O}(N)$ for $i \in [I]$ with $I < M < N$. For standard sparsity, MSR with OMP matches the time complexity of KroOMP (Caiafa & Cichocki, 2013), but reduces space complexity from $\mathcal{O}(N^I)$ to $\mathcal{O}(M^{I-1}N)$. For hierarchical sparsity, our MSR with HTP has time complexity $\mathcal{O}(MN^I)$ and space complexity $\mathcal{O}(M^{I-1}N)$, improving over HiHTP (Roth et al., 2020) with time and space complexities of $\mathcal{O}(M^2N^2)$ for $I = 2$. For Kronecker-supported sparsity, MSR with SBL lowers time complexity to $\mathcal{O}(MN^I)$ and space complexity to $\mathcal{O}(N^I)$ compared to AM- and SVD-KroSBL (He & Joseph, 2025a) with both complexities $\mathcal{O}(M^IN^I)$. The improvements are due to $i)$ the exploitation of the Kronecker structure through tensor operation, reducing the dimensionality; and $ii)$ leveraging the MMV structure from Lemma 1. We refer to Table 3 in Appendix F for a comprehensive comparison.

## 4 UNIFIED ANALYSIS FOR STRUCTURED SPARSITY MODELS

We establish a unified RIP analysis via a generalized notion of RIP called the $(\boldsymbol{s}, \mathrm{N})$-RIP condition with $\boldsymbol{s} := (s_I, s_{I-1}, \cdots, s_1)$ and $\mathrm{N} := (N_I, N_{I-1}, \cdots, N_1)$ defined by the dimension of factor matrices in KCS. To this end, we introduce the generalized $(\boldsymbol{s}, \mathrm{N})$ sparsity model, tailored to the KCS problem, which reflects a hierarchical view where sparsity at each level affects recovery.

**Sparsity 4** (Generalized sparsity). *Consider KCS with $\boldsymbol{H}_i \in \mathbb{R}^{M_i \times N_i}$. A vector $\boldsymbol{x} \in \mathbb{R}^{\bar{N}}$ is $(\boldsymbol{s}, \mathrm{N})$ sparse if for tensor $\mathsf{X} \in \mathbb{R}^{N_1 \times \cdots \times N_I}$ reordered from $\boldsymbol{x}$ using $\mathrm{N} := (N_I, N_{I-1}, \cdots, N_1)$, the maximum number of nonzero rows of each of the column blocks of its $j$th mode unfolding $\boldsymbol{X}_{(j)}$ is $s_j$.*

**Relation to other models:** We relate the above model to the standard, hierarchical, Kronecker-supported, and block sparsity models. The standard sparsity model is not a special case of $(\boldsymbol{s}, \mathrm{N})$ sparsity, but the set of $s$ sparse vectors is contained in a union of $(\boldsymbol{s}, \mathrm{N})$ sparse vectors.

**Lemma 2.** *Let set $\mathcal{S}$ contains all $s$ standard sparse vectors in $\mathbb{R}^{\bar{N}}$, and $\mathcal{S}_{\boldsymbol{s}}$ contains all $(\boldsymbol{s}, \mathrm{N})$ sparse vectors in $\mathbb{R}^{\bar{N}}$ for a given $(\boldsymbol{s}, \mathrm{N})$. Then, $\mathcal{S} \subset \cup_{\boldsymbol{s} \in f_{\mathrm{N}}(s)} \mathcal{S}_{\boldsymbol{s}}$, where $f_{\mathrm{N}}(s) = \{\boldsymbol{s} : \sum_{i=1}^{I} s_i \leq s + (I-1),\ 1 \leq s_i \leq s\}$.*

Hierarchical sparsity is a special case of $(\boldsymbol{s}, \mathrm{N})$ sparsity when the hierarchical partition structure matches the dimensions of factor matrices in the Kronecker measurement matrix. If, additionally, all the column blocks of $j$th mode unfolding $\boldsymbol{X}_{(j)}$ share the same support regarding nonzero rows, then we arrive at the Kronecker-supported sparsity. Block sparsity can also be viewed as $(\boldsymbol{s}, \mathrm{N})$ sparsity with $I = 2$ when the block boundary matches the hierarchical partition structure.

We next define the $(\boldsymbol{s}, \mathrm{N})$-RIP condition for a Kronecker product matrix $\boldsymbol{H}$.

**Definition 1** $((\boldsymbol{s}, \mathrm{N})$-RIP)**.** *A Kronecker product matrix $\boldsymbol{H} = \otimes_{i=I}^{1} \boldsymbol{H}_i$ with $\boldsymbol{H}_i \in \mathbb{R}^{M_i \times N_i}$ satisfies $(\boldsymbol{s}, \mathrm{N})$-RIP if there exists $\delta \in (0, 1)$ such that for all $(\boldsymbol{s}, \mathrm{N})$ sparse $\boldsymbol{x} \in \mathbb{R}^{\bar{N}}$, it satisfies $(1-\delta)\|\boldsymbol{x}\|_2^2 \leq \|\boldsymbol{H}\boldsymbol{x}\|_2^2 \leq (1+\delta)\|\boldsymbol{x}\|_2^2$. The smallest feasible $\delta$, denoted as $\delta_{(\boldsymbol{s}, \mathrm{N})}(\boldsymbol{H})$, is the $(\boldsymbol{s}, \mathrm{N})$-RIC of $\boldsymbol{H}$.*

Under our models, $(\boldsymbol{s}, \mathrm{N})$-RIP is defined over the unions of subspaces, thus can be used to guarantee the success of recovery algorithms, such as iterative hard thresholding (IHT) and HTP (Blumensath, 2011). In general, such guarantees are established using the upper bound of the RICs. Therefore, we first derive the upper bound of $\delta_{(\boldsymbol{s}, \mathrm{N})}(\boldsymbol{H})$, then discuss its implications for different sparsity models, and finally discuss the associated recovery algorithms and guarantees. Here, we denote the standard $s$-RIC of matrix $\boldsymbol{H}$ as $\delta_s(\boldsymbol{H})$.

**Theorem 1.** *The $(\boldsymbol{s}, \mathrm{N})$-RIC of Kronecker product dictionary $\boldsymbol{H} = \otimes_{i=I}^{1} \boldsymbol{H}_i$, i.e., $\delta_{(\boldsymbol{s}, \mathrm{N})}(\boldsymbol{H})$, satisfies $\delta_{(\boldsymbol{s}, \mathrm{N})}(\boldsymbol{H}) \leq \prod_{i=I}^{1}(1 + \delta_{s_i}(\boldsymbol{H}_i)) - 1$.*

The above result immediately applies to hierarchical and Kronecker-supported sparsity, as both are special cases of $(\boldsymbol{s}, \mathrm{N})$ sparsity. For Kronecker-supported sparsity, a tighter bound could be expected due to its additional joint sparsity structure arising from the shared support across the nonzero block. However, improving the RIC bound by exploiting this additional joint sparsity is difficult. As noted in Li & Petropulu (2013); Eldar & Mishali (2009), RIP analysis considers the worst-case performance and does not guarantee that MMV outperforms the SMV case. So, our bound shows no improvement, and deriving a stronger RIP-based condition for the MMV model is an open problem.

Theorem 1 can also be tailored to standard sparsity using Lemma 2.

**Corollary 1.** *Consider the Kronecker product $\boldsymbol{H} = \otimes_{i=I}^{1} \boldsymbol{H}_i$. For any $s$, the $s$-RIC of $\boldsymbol{H}$ satisfies $\delta_s(\boldsymbol{H}) \leq \max_{\boldsymbol{s} \in f_{\mathrm{N}}(s)} \delta_{(\boldsymbol{s}, \mathrm{N})}(\boldsymbol{H}) \leq \max_{\boldsymbol{s} \in f_{\mathrm{N}}(s)} \prod_{i=1}^{I}(1 + \delta_{s_i}(\boldsymbol{H}_i)) - 1$.*

The $s$-RIC bound corroborates that only the sparsity level at different levels of blocks explicitly affects the $s$-RIC of Kronecker-structured $\boldsymbol{H}$. Also, a known upper RIC bound is $\delta_s(\boldsymbol{H}) \leq \prod_{i=1}^{I}(1 + \delta_s(\boldsymbol{H}_i)) - 1$ (Duarte & Baraniuk, 2011a). Our bound slightly improves this bound:

$$\max_{\boldsymbol{s} \in f_{\mathrm{N}}(s)} \prod_{i=1}^{I}(1 + \delta_{s_i}(\boldsymbol{H}_i)) - 1 \leq \prod_{i=1}^{I}(1 + \delta_s(\boldsymbol{H}_i)) - 1,$$

because $\delta_s$ is a non-decreasing function of $s$ (Foucart & Rauhut, 2013) and $s_i^* \leq s$ for all $i \in [I]$ and the equality cannot be achieved simultaneously.

**Maximum sparsity level:** Corollary 1 indicates that recovering $s$ standard sparse vectors via KCS with $M_i < N_i$ is only guaranteed when $s < \min_i N_i$, as it is a worst-case analysis. When $s = \min_i N_i$ with $j = \arg \min_i N_i$, a worst-case scenario is $s_j = s = N_j$ and $s_i = 1$ for all $i \neq j$. Then, $\delta_{s_j} = \|\boldsymbol{H}_j^\top \boldsymbol{H}_j - \boldsymbol{I}_{N_j}\|_2 \geq 1$, making $\boldsymbol{H}_j$ is a non-injective map, and recovery is impossible. This also indicates that it is only possible to recover block-sparse vectors with block length smaller than $\min_i N_i$. However, recovery is still possible for $s \geq \min_i N_i$ in structured sparsity settings.

**Measurement bounds for classical methods:** We discuss the implications of Theorem 1 on measurement bounds for recovering $(\boldsymbol{s}, \mathrm{N})$-sparse vectors using classical iterative algorithms, namely

IHT and HTP. For both algorithms, at iteration $k$, the support is updated via thresholding operator $L_{\mathcal{S}}$ as $\mathcal{T}^{k+1} = L_{\mathcal{S}}\left(\boldsymbol{x}^k + \boldsymbol{H}^\top\left(\boldsymbol{y} - \boldsymbol{H}\boldsymbol{x}^k\right)\right)$. The thresholding operator depends on the sparsity model. For standard $s$ sparse, $L_{\mathcal{S}}$ returns the support of the $s$ largest entries of $\boldsymbol{x}$ in amplitude (Foucart & Rauhut, 2013). For $\boldsymbol{s}$ hierarchically sparse, it selects the top $s_1$ entries within each first-level block, then recursively picks top $s_2, \ldots, s_I$ blocks at higher levels based on the $\ell_2$ norm, as in Roth et al. (2020). However, finding the thresholding operator $L_{\mathcal{S}}$ for $\boldsymbol{s}$ Kronecker-supported sparse vectors is NP-hard and not available in the literature. For example, when $I = 2$, it reduces to selecting rows and columns whose intersection maximizes the squared sum, equivalent to the NP-hard maximum weight biclique problem. A practical alternative is to first select the top $s_I$ blocks at the $I$th level by $\ell_2$ norm, then recursively sum norms across matching indices at each lower level and select the top $s_{I-1}, \ldots, s_1$ blocks; this is the approach we use in simulations for comparison. Then, IHT applies a simple projection while HTP solves a least-squares problem on the support,

$$\boldsymbol{x}^{k+1} = \left(\boldsymbol{x}^k + \boldsymbol{H}^\top\left(\boldsymbol{y} - \boldsymbol{H}\boldsymbol{x}^k\right)\right)_{\mathcal{T}^{k+1}}, \tag{IHT}$$

$$\boldsymbol{x}^{k+1} = \arg\min_{\boldsymbol{x}\in\mathbb{R}^{\bar{N}}} \|\boldsymbol{y} - \boldsymbol{H}\boldsymbol{x}\|_2, \;\; \text{supp}(\boldsymbol{x}) \in \mathcal{T}^{k+1}, \tag{HTP}$$

where operator $(\cdot)_{\mathcal{T}^{k+1}}$ only preserves the entries within the set $\mathcal{T}^{k+1}$ and sets the others to zero.

We next discuss the implications for measurement bounds. It is known that for IHT and HTP to recover a vector from a union of subspaces, tailoring the thresholding operator $L_{\mathcal{S}}$ to the union and having an RIC below $1/\sqrt{3}$ over that union is sufficient to guarantee convergence to the ground truth (Foucart & Rauhut, 2013; Roth et al., 2020). So, our results shows that $\max_{\boldsymbol{s}\in f_{\mathrm{N}}(3s)} \delta_{(\boldsymbol{s},\mathrm{N})} < 1/\sqrt{3}$ (for $s$ standard sparsity) and $\delta_{(3\boldsymbol{s},\mathrm{N})} < 1/\sqrt{3}$ (for $\boldsymbol{s}$ hierarchical sparsity) are sufficient for the success of IHT and HTP. However, it does not guarantee the recovery of the $\boldsymbol{s}$ Kronecker-supported sparse vectors as the thresholding operator is suboptimal.

To compare the measurement bound for KCS, we consider the simplest case with $I = 2$ and $s = \mathcal{O}(s_1 s_2)$ for $\boldsymbol{s} \in f_{\mathrm{N}}(s)$, and Gaussian factor matrices $\boldsymbol{H}_i$'s. For recovering $s$ standard sparse vectors, our Corollary 1 implies that each $\boldsymbol{H}_i$ satisfies the $s_i$-RIP, requiring $M_i = \mathcal{O}(s_i \log N_i)$ (Foucart & Rauhut, 2013). So, the total measurement bound scales as $\bar{M} = \mathcal{O}(s_1 s_2 \log N_1 \log N_2)$ improving over the existing bound $\bar{M} = \mathcal{O}(s^2 \log N_1 \log N_2) = \mathcal{O}(s_1^2 s_2^2 \log N_1 \log N_2)$ (Duarte & Baraniuk, 2011a). In comparison, standard compressed sensing with *fully unstructured Gaussian matrix* requires only $\mathcal{O}(s_1 s_2 \log N_1 N_2)$ measurements, which is smaller due to greater flexibility and randomness in measurement. However, KCS exploits the multidimensional structure to reduce the computational complexity during recovery. For the recovery of $\boldsymbol{s}$ hierarchical sparse vectors, Corollary 1 suggests a measurement bound $\mathcal{O}(s_1 s_2 \log N_1 \log N_2)$, while a *fully unstructured Gaussian matrix* requires only $\mathcal{O}(s_1 s_2 \log N_1 + s_2 \log N_2)$ (Roth et al., 2020).

**Measurement bounds for our MSR:** We now establish recovery guarantees for MSR with IHT and HTP using the RICs of factor matrices.

**Theorem 2.** *Consider the sparse recovery problem, $\boldsymbol{y} = \left(\otimes_{i=I}^1 \boldsymbol{H}_i\right)\boldsymbol{x} + \boldsymbol{n}$. Define tensors $\mathsf{X}$ and $\mathsf{N}$, which are reshaped from $\boldsymbol{x}$ and $\boldsymbol{n}$, respectively, using the dimensions of $\boldsymbol{H}_i$'s. If $\boldsymbol{x}$ is an $s$ standard sparse vector and the factor matrices $\boldsymbol{H}_i$ for $i \in [I]$ satisfy $\delta_{3s_i}(\boldsymbol{H}_i) < 1/\sqrt{3}$ for $\forall \boldsymbol{s} \in f_{\mathrm{N}}(s)$, then the estimate $\hat{\boldsymbol{x}}$ of $\boldsymbol{x}$ using $k$-iteration IHT or HTP in Algorithm 1, satisfies*

$$\|\hat{\boldsymbol{x}} - \boldsymbol{x}\|_2 \leq \max_{\boldsymbol{s}\in f_{\mathrm{N}}(s)} \sum_{n_2,\cdots,n_I} \left(\sum_{i=1}^I \prod_{j=1}^{i-1} \tau_j \alpha_i^k \left\|[\boldsymbol{U}_i]_{\mathrm{n}_{i+1}}\right\|_{\mathrm{F}} + \prod_{i=1}^I \tau_i \|\mathsf{N}\|_{\mathrm{F}}\right),$$

*where $[\boldsymbol{U}_i]_{\mathrm{n}_{i+1}} = [\boldsymbol{X}_{(i)}]_{\mathrm{n}_{i+1}} \left(\otimes_{l=i-1}^1 \boldsymbol{H}_l\right)^\top$, and if $\boldsymbol{x}$ is an $\boldsymbol{s}$ hierarchically sparse vector, and the factor matrices $\boldsymbol{H}_i$ for $i \in [I]$ satisfy $\delta_{3s_i}(\boldsymbol{H}_i) < 1/\sqrt{3}$, then the estimate $\hat{\boldsymbol{x}}$ of $\boldsymbol{x}$ using $k$-iteration IHT or HTP in Algorithm 1, satisfies*

$$\|\hat{\boldsymbol{x}} - \boldsymbol{x}\|_2 \leq \sum_{n_2,\cdots,n_I} \left(\sum_{i=1}^I \prod_{j=1}^{i-1} \tau_j \alpha_i^k \left\|[\boldsymbol{U}_i]_{\mathrm{n}_{i+1}}\right\|_{\mathrm{F}} + \prod_{i=1}^I \tau_i \|\mathsf{N}\|_{\mathrm{F}}\right),$$

*and if $\boldsymbol{x}$ is an $\boldsymbol{s}$ Kronecker-supported sparse vector, there is*

$$\|\hat{\boldsymbol{x}} - \boldsymbol{x}\|_2 \leq \sum_{i=1}^I \prod_{j=1}^{i-1} \tau_j \alpha_i^k \left\|\boldsymbol{U}_i\right\|_{\mathrm{F}} + \prod_{i=1}^I \tau_i \|\mathsf{N}\|_{\mathrm{F}},$$

*where* $\boldsymbol{U}_i = \boldsymbol{X}_{(i)} \left( \boldsymbol{I}_{\prod_{l=I}^{i+1} N_l} \otimes \left( \otimes_{l=i-1}^1 \boldsymbol{H}_l \right) \right)^\top$, *with* $\alpha_i < 1$, *and* $\tau_i$ *are*

$$\textit{MSIHT: } \alpha_i = \sqrt{3}\delta_{3s_i}(\boldsymbol{H}_i); \ \tau_i = (1 - \alpha_i^k)\frac{\sqrt{3(1 + \delta_{2s_i}(\boldsymbol{H}_i))}}{1 - \alpha_i}, \textit{ and}$$

$$\textit{MSHTP: } \alpha_i = \sqrt{2\delta_{3s_i}^2(\boldsymbol{H}_i)/(1 - \delta_{2s_i}^2(\boldsymbol{H}_i))}; \ \tau_i = (1 - \alpha_i^k)\frac{\sqrt{2(1 - \delta_{2s_i}(\boldsymbol{H}_i))} + \sqrt{1 + \delta_{s_i}(\boldsymbol{H}_i)}}{(1 - \delta_{2s_i}(\boldsymbol{H}_i))(1 - \alpha_i)}.$$

As the number of iterations $k \to \infty$, the error bound reduces to $\tau_1 \prod_{i=2}^I \tau_i N_i \|\mathbf{N}\|_F$ for the standard and hierarchical sparsity, and $\prod_{i=1}^I \tau_i \|\mathbf{N}\|_F$ for Kronecker-supported sparsity. So, MSIHT and MSHTP approach the true value within a constant factor of measurement noise power. Although factors $\tau_1 \prod_{i=2}^I \tau_i N_i$ and $\prod_{i=1}^I \tau_i$ suggest error propagation as the algorithm proceeds from $j = I$ till $j = 1$ and scale with the problem dimension, this amplification is not observed in practice (see Figure 3). The bound for $\boldsymbol{s}$ Kronecker-supported sparsity is tighter than that for the other two models because it solves a single MMV problem, resulting a collective error bound, instead of a looser bounds due to the sum of each individual MMV bound. While our MSR's measurement bound scales the same as classical methods due to a shared requirement on the $s_i$-RIP of $\boldsymbol{H}_i$'s, it can have a larger error from propagation, potentially requiring more iterations or $\boldsymbol{H}_i$'s with smaller $s_i$-RICs. However, a key advantage of MSR is that it provides recovery guarantees for the Kronecker-supported sparsity model, unlike classical IHT and HTP-based methods.

## 5 NUMERICAL EVALUATIONS

For numerical results, we combine MSR with MMV-SBL (Wipf & Rao, 2007), SIHT (Blanchard et al., 2014), SHTP (Blanchard et al., 2014), and SOMP (Tropp et al., 2006), and the resulting algorithms are referred to as MSSBL, MSIHT, MSHTP, and MSOMP, respectively. Our benchmark for the standard sparsity is KroOMP (Caiafa & Cichocki, 2013). Here, we omit computationally intensive SBL and OMP whose results are identical to KroOMP. For hierarchical sparsity, our benchmark is the state-of-the-art HiHTP (Roth et al., 2020). For Kronecker-structured support sparsity, we benchmark with the state-of-the-art AM- and SVD-KroSBL (He & Joseph, 2025a). Unlike the OMP/SBL-based algorithms, the IHT/HTP-based algorithms need the true sparsity level $\boldsymbol{s}$ as input.

For all three models, we set $M_i = M$, $N_i = N$, and $s_i = s$ for $i \in [I]$. For the $\boldsymbol{s}$ standard sparsity, we opt for $\boldsymbol{H} = \otimes_{i=I}^1 \boldsymbol{H}_i$ with $I = 2$, $M = 64$, and $N = 80$. The entries of $\boldsymbol{H}_i$ and the nonzero entries of $\boldsymbol{x}$ are drawn independently from the standard normal distribution. We set $s = 15$, and the support is randomly drawn from a uniform distribution. For $\boldsymbol{s}$ hierarchically sparse vectors, we also opt for $I = 2$, $M = 64$, $N = 80$, and $s = 15$. Here, supports are generated by first selecting $s$ blocks uniformly at random, then assigning support within each block uniformly. In the Kronecker-supported sparsity model, we opt for $I = 3$, $M = 15$, $N = 18$, and $s = 4$. The measurement noise is zero mean white Gaussian noise whose variance is determined by SNR (dB) $= 10 \log_{10} \mathbb{E}\{\|\boldsymbol{H}\boldsymbol{x}\|_2^2 / \|\boldsymbol{n}\|_2^2\}$ of $\{3, 5, \cdots, 23, 25\}$.

Our metrics are runtime and the normalized squared error NSE $= \|\boldsymbol{x} - \hat{\boldsymbol{x}}\|_2^2 / \|\boldsymbol{x}\|_2^2$, where $\boldsymbol{x}$ is the ground truth and $\hat{\boldsymbol{x}}$ is the estimated vector. The results are shown in Figure 2 and Table 1, with the figure showing median and 25%/75% quartiles, and the table showing averages. The NSE for recovering an $\boldsymbol{s}$ standard sparse vector is shown in Figure 2a. Compared to KroOMP, MSOMP provides similar performance regarding NSE but needs one to three orders less runtime, as in Table 1. MSSBL outperforms KroOMP in all SNR cases with one or two orders less runtime. The NSE for hierarchical sparsity is shown in Figure 2b using only the HTP/IHT-based algorithms (full comparison in Appendix H). Our MSHTP/MSIHT offers similar performance to HTP and HiHTP, and IHT. However, MSHTP requires two orders less runtime than HTP and one order less runtime than HiHTP; and MSIHT requires two orders less runtime than IHT. The NSE for Kronecker-supported sparsity is shown in Figure 2c. Our MSSBL consistently achieves a comparable NSE and is two or three orders faster than AM- and SVD-KroSBL. In summary, MSR variants achieve similar or better accuracy than existing methods while drastically reducing computation time.

Figure 3 shows how the NSE and runtime (median with 25%/75% quartiles) of MSSBL, MSHTP, and MSIHT scale with the problem dimension, focusing on hierarchical sparsity. We choose $I = 3$ and SNR as 20dB and vary $N = \{50, 60, \cdots, 110\}$, so that the problem dimension $\bar{N} = N^I =$

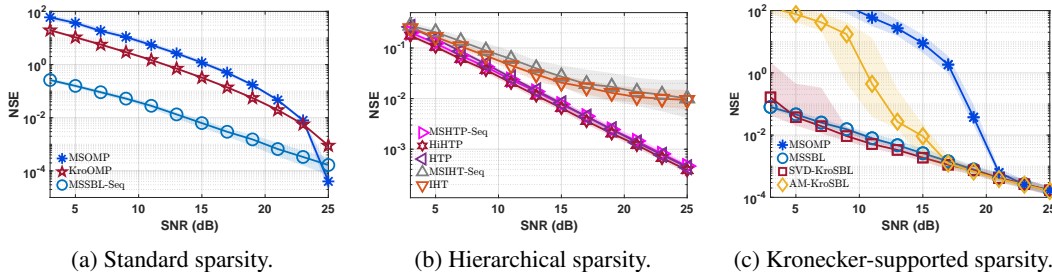

(a) Standard sparsity.  (b) Hierarchical sparsity.  (c) Kronecker-supported sparsity.

Figure 2: NSE as a function of SNR.

Table 1: Average runtime in seconds. **Bold**: the best result.

| SNR | 3 dB | 7 dB | 11 dB | 15 dB | 19 dB | 23 dB |
|---|---|---|---|---|---|---|
| Recovery of $s$ sparse vectors | | | | | | |
| MSOMP-Seq | **0.4256** | **0.4119** | **0.3827** | 0.3329 | 0.2204 | **0.0568** |
| KroOMP (Caiafa & Cichocki, 2013) | 130.5405 | 108.0526 | 76.6942 | 39.9844 | 11.5774 | 0.7525 |
| MSSBL-Seq | 1.8191 | 1.1016 | 0.5758 | **0.2218** | **0.1417** | 0.1141 |
| Recovery of $s$ hierarchically sparse vectors | | | | | | |
| MSHTP-Seq | **0.0379** | **0.0305** | **0.0297** | **0.0247** | **0.0186** | **0.0168** |
| HiHTP (Roth et al., 2020) | 0.6512 | 0.5493 | 0.5204 | 0.5444 | 0.4398 | 0.4574 |
| HTP | 2.2436 | 1.7170 | 1.3256 | 0.8450 | 0.8264 | 0.5311 |
| MSIHT-Seq | 0.0500 | 0.0510 | 0.0532 | 0.0509 | 0.0450 | 0.0434 |
| IHT | 8.2437 | 8.2412 | 8.2554 | 8.2917 | 8.2889 | 8.2789 |
| Recovery of $s$ Kronecker-supported sparse vectors | | | | | | |
| MSOMP | **0.0042** | **0.0041** | **0.0040** | **0.0038** | **0.0026** | **0.0015** |
| MSSBL | 0.0728 | 0.0587 | 0.0447 | 0.0279 | 0.0119 | 0.0051 |
| SVD-KroSBL (He & Joseph, 2025a) | 37.1233 | 26.9816 | 14.2405 | 8.6036 | 5.4067 | 4.0681 |
| AM-KroSBL (He & Joseph, 2025a) | 55.9532 | 63.4676 | 75.9727 | 74.5840 | 51.7089 | 34.1331 |

$125000, 216000, \cdots, 1331000$, where $M = \left\lceil (0.6\bar{N})^{1/I} \right\rceil$ and $s = \lceil 0.4N \rceil$. As expected, parallel implementation is faster than sequential. MSSBL has the best NSE but is slower than MSIHT and MSHTP. The MSIHT is worse than MSHTP due to IHT's slow convergence (Foucart & Rauhut, 2013). Overall, our MSR efficiently handles large dimensional KCS problems.

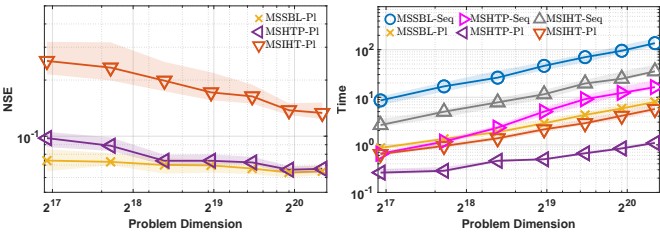

Figure 3: NSE and runtime of MSR as functions of problem dimension $\bar{N}$.

**Application to wideband massive multiple-input multiple-output (MIMO) channel estimation**: Massive MIMO has been a key enabler for the fifth generation communication. For data transmission, an important task is to estimate the channel by processing the received pilot signals sent from user. We focus on the orthogonal frequency-division multiplexing (OFDM)-based wideband massive MIMO channel estimation, where we consider a base station with a half-wavelength spacing uniform linear array equipped with $N_a$ elements serving one single antenna user. Due to the environment reflection, we consider $L$ impinging angles, each containing up to $K_L$ delays. The maximum delay is $\alpha T_s$ with $\alpha \leq 1$ where $T_s$ is the OFDM symbol duration. The number of subcarriers of the OFDM symbol is $N_s$. The channel matrix $C$ is the superposition of impinging waves characterized by delays and angles as $C = \sum_{l=1}^{L} \sum_{k_l=1}^{K_L} \rho_{l,k_l} d(\tau_{l,k_l}) a^H(\theta_l)$ (Haghighatshoar & Caire, 2017; Chen & Yang, 2016), where $\rho_{l,k_l} \in \mathbb{C}$ is the complex gain of the path corresponding to the $k_l$th delay of the $l$th angle, $d(\tau_{l,k_l}) := [1, e^{-j2\pi\tau_{l,k_l}/T_s}, \cdots, e^{-j2\pi(N_s-1)\tau_{l,k_l}/T_s}]^\top$ is the delay manifold vector of the delay $\tau_{l,k_l}$, while $a(\theta_l) := [1, e^{-j2\pi\theta_l}, \cdots, e^{-j2\pi(N_a-1)\theta_l}]^\top$ is the steering vector for $\theta_l \in [0, 1]$ representing the equivalent $l$th impinging angle (Wunder et al.,

2019). Due to the significant path loss, the received signal is transmitted through a limited number of paths, making the channel intrinsically sparse over two sparsifying bases. The first is obtained by sampling the delay range $[0, T_s]$ with $N_s$ samples as $\{nT_s/N_s\}_{n=0}^{N_s-1}$, leading to a delay basis $\boldsymbol{H}_d := [\boldsymbol{d}(0), \boldsymbol{d}(T_s/N_s), \cdots, \boldsymbol{d}((N_d-1)T_s/N_s)] \in \mathbb{C}^{N_s \times N_d}$, with $N_d := \lfloor \alpha N_s \rfloor$. The second is obtained by sampling the angular domain as $\{n/N_a\}_{n=0}^{N_a-1}$, yielding angle basis $\boldsymbol{H}_a := [\boldsymbol{a}(0), \boldsymbol{a}(1/N_a), \cdots, \boldsymbol{a}(1-1/N_a)] \in \mathbb{C}^{N_a \times N_a}$. Then the channel can be represented as $\boldsymbol{C} = \boldsymbol{H}_d \boldsymbol{X} \boldsymbol{H}_a^H$, where $\boldsymbol{X} \in \mathbb{C}^{N_d \times N_a}$ is the sparse representation with up to $LK_L$ nonzeros.

To reduce the overhead, one may use a subset of OFDM subcarriers and array elements for channel estimation. Denote the pilot as $\boldsymbol{p} \in \mathbb{C}^{M_d}$, where $M_d \leq N_s$ is the number of subcarriers in a subset. Let $\boldsymbol{S}_d \in \{0,1\}^{M_d \times N_s}$ be the sampling matrix for subcarriers and $\boldsymbol{S}_a \in \{0,1\}^{M_a \times N_a}$ be the sampling matrix for array where only $M_a$ out of $N_a$ elements are chosen. We write the received signal as $\boldsymbol{Y} = \text{diag}(\boldsymbol{p})\boldsymbol{S}_d \boldsymbol{C} \boldsymbol{S}_a^\top + \boldsymbol{N} \in \mathbb{C}^{M_d \times M_a}$ with $\boldsymbol{N}$ being noise (Wunder et al., 2019). Plugging in the sparse channel representation and vectorizing both sides of the equation, we have

$$\boldsymbol{y} = ((\boldsymbol{S}_a \boldsymbol{H}_a^*) \otimes (\text{diag}(\boldsymbol{p})\boldsymbol{S}_d \boldsymbol{H}_d)) \, \boldsymbol{x} + \boldsymbol{n},$$

where $\boldsymbol{y} = \text{vec}(\boldsymbol{Y})$, $\boldsymbol{x} = \text{vec}(\boldsymbol{X})$, $\boldsymbol{n} = \text{vec}(\boldsymbol{N})$, and $(\cdot)^*$ is the conjugate. Denoting $\boldsymbol{S}_a \boldsymbol{H}_a^* = \boldsymbol{H}_2 \in \mathbb{C}^{M_a \times N_a}$, $\text{diag}(\boldsymbol{p})\boldsymbol{S}_d \boldsymbol{H}_d = \boldsymbol{H}_1 \in \mathbb{C}^{M_d \times N_d}$, and $I = 2$, the channel estimation problem is a Kronecker compressed sensing problem with $\boldsymbol{x}$ being $\boldsymbol{s} = (L, K_L)$ hierarchically sparse. For simulation, we consider $N_a = 512$, $L = \{5, 10, 15, 20, 25, 35, 50, 75, 100\}$, $K_L = 3$, $N_s = 1024$ OFDM subcarriers, and $\alpha = 0.5$ for the maximum delay. We fix $M_a = \lceil 0.3N_a \rceil$ and $M_d = \lceil 0.1N_s \rceil$, making $\boldsymbol{H}_2 \in \mathbb{C}^{154 \times 512}$ and $\boldsymbol{H}_1 \in \mathbb{C}^{103 \times 512}$. Both angles $\{\theta_l\}$ and delays $\{\tau_{l,k_l}\}$ are generated independently and uniformly over the sampling grid, while path gains $\{\rho_{l,k_l}\}$ are drawn from a standard normal distribution (Wunder et al., 2019). The measurement noise is zero mean white Gaussian noise whose variance is determined by SNR (dB) $= 10 \log_{10} \mathbb{E}\{\|((\boldsymbol{S}_a \boldsymbol{H}_a^*) \otimes (\text{diag}(\boldsymbol{p})\boldsymbol{S}_d \boldsymbol{H}_d))\boldsymbol{x}\|_2^2/\|\boldsymbol{n}\|_2^2\}$ of 20dB. We evaluate NSE $:= \|\boldsymbol{C} - \boldsymbol{H}_d \hat{\boldsymbol{X}} \boldsymbol{H}_a^H\|_F^2/\|\boldsymbol{C}\|_F^2$ and runtime and compare MSHTP and MSOMP to HiIHT/HiHTP in (Wunder et al., 2019). Results are obtained by two hundred independent trials.

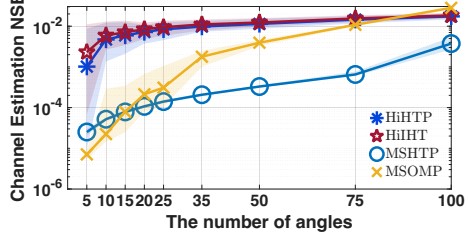

Figure 4: NSE of different schemes.

| #angles $L$ | HiHTP | HiIHT | MSHTP | MSOMP |
|---|---|---|---|---|
| 5 | 0.0738 | 4.1465 | 0.0195 | 0.0080 |
| 10 | 0.1303 | 4.1792 | 0.0404 | 0.0135 |
| 15 | 0.1444 | 4.1287 | 0.0550 | 0.0189 |
| 20 | 0.1957 | 4.1337 | 0.0769 | 0.0247 |
| 25 | 0.2107 | 4.1670 | 0.0920 | 0.0303 |
| 35 | 0.3117 | 4.1311 | 0.1243 | 0.0426 |
| 50 | 0.5073 | 4.1571 | 0.1816 | 0.0641 |
| 75 | 0.8801 | 4.2211 | 0.2706 | 0.1188 |
| 100 | 1.3406 | 4.2272 | 0.2386 | 0.2139 |

Table 2: Average runtime in seconds.

We present NSE of channel estimation and average runtime in Figure 4 and Table 2, respectively. We observe that MSHTP and MSOMP provide better performance than HiHTP and HiIHT in most cases, with one or two orders less runtime. MSOMP's relatively higher NSE with large $L$ is because it wrongly identifies many insignificant paths (smaller $|\rho_{l,k_l}|$), since it does not require the true sparsity level $(L, K_L)$ as input. However, we still observe that the significant paths are estimated accurately and efficiently, making MSOMP a practical option for the channel estimation task.

## 6 CONCLUSION

We investigated the Kronecker compressed sensing problem for signals with multiple sparsity structures. We presented a novel hierarchical view, comprehending that each factor matrix in the Kronecker product dictionary senses the sparse signal at a different level, obeying a hierarchical structure. This insight led to a computationally efficient, multi-stage recovery framework that achieved performance comparable to state-of-the-art methods with one order or less runtime. On the theoretical front, we unified the RIP analysis for Kronecker product matrices across various structured sparsity models, and also established the recovery guarantee for our multi-stage recovery algorithm. This hierarchical framework opens promising avenues for designing new algorithms to accommodate more structured patterns and provide efficient solutions to many applications.

## 7 REPRODUCIBILITY STATEMENT

All conditions required to reproduce the results are included in Section 5 and Appendix H. Our implementation and data for reproducing figures and tables are available at https://github.com/YanbinHe/msr.

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

## A    PROOF OF LEMMA 1

The proof proceeds in two parts: first, establishing the column block structure, and second, analyzing the sparsity of each block.

For the first step, we observe that matrix $\left(\boldsymbol{I}_{\prod_{i=I}^{j+1} N_i} \otimes \left(\otimes_{i=j-1}^{1} \boldsymbol{H}_i\right)\right)^{\top}$ is a *block-diagonal matrix*. It has $\prod_{i=I}^{j+1} N_i$ identical diagonal blocks, each equal to $\left(\otimes_{i=j-1}^{1} \boldsymbol{H}_i\right)^{\top}$. To match this structure, we partition the columns of the unfolded matrix $\boldsymbol{X}_{(j)}$ into $\prod_{i=I}^{j+1} N_i$ column blocks. The standard column ordering in tensor unfolding places elements with higher-level indices $(n_{j+1}, \ldots, n_I)$ further apart. Consequently, we can partition $\boldsymbol{X}_{(j)}$ into $\prod_{i=j+1}^{I} N_i$ column blocks, where each block corresponds to a unique *encapsulation* $\mathrm{n}_{j+1} = (n_{j+1}, \ldots, n_I)$ as

$$\boldsymbol{X}_{(j)} = \begin{bmatrix} \boldsymbol{X}_{(j),(1,\ldots,1)} & \cdots & \boldsymbol{X}_{(j),(N_{j+1},\ldots,N_I)} \end{bmatrix}.$$

Since $\left(\boldsymbol{I}_{\prod_{i=I}^{j+1} N_i} \otimes \left(\otimes_{i=j-1}^{1} \boldsymbol{H}_i\right)\right)^{\top}$ is block-diagonal, the multiplication with $\boldsymbol{X}_{(j)}$ decouples and operates on each of these blocks independently,

$$\boldsymbol{X}_{(j)} \left(\boldsymbol{I}_{\prod_{i=I}^{j+1} N_i} \otimes \left(\otimes_{i=j-1}^{1} \boldsymbol{H}_i\right)\right)^{\top}$$
$$= \begin{bmatrix} \boldsymbol{X}_{(j),(1,\ldots,1)} \left(\otimes_{i=j-1}^{1} \boldsymbol{H}_i\right)^{\top} & \cdots & \boldsymbol{X}_{(j),(N_{j+1},\ldots,N_I)} \left(\otimes_{i=j-1}^{1} \boldsymbol{H}_i\right)^{\top} \end{bmatrix}.$$

This confirms that the resulting matrix is also composed of $\prod_{i=j+1}^{I} N_i$ column blocks, each indexed by $\mathrm{n}_{j+1}$ and given by $\boldsymbol{X}_{(j),\mathrm{n}_{j+1}} \left(\otimes_{i=j-1}^{1} \boldsymbol{H}_i\right)^{\top}$.

For the second step, consider a column block indexed by a fixed $\mathrm{n}_{j+1}$, i.e., $\boldsymbol{X}_{(j),\mathrm{n}_{j+1}} \left(\otimes_{i=j-1}^{1} \boldsymbol{H}_i\right)^{\top}$. The rows of this block are indexed by $n_j \in [N_j]$. The $k$th row of $\boldsymbol{X}_{(j),\mathrm{n}_{j+1}} \left(\otimes_{i=j-1}^{1} \boldsymbol{H}_i\right)^{\top}$ will be nonzero *if and only if* the $k$th row of $\boldsymbol{X}_{(j),\mathrm{n}_{j+1}}$ contains nonzeros due to the full row rankness. Moreover, the $k$th row of $\boldsymbol{X}_{(j),\mathrm{n}_{j+1}}$ is the hierarchical block $\boldsymbol{x}_{\mathrm{n}_j}$ where the encapsulation is $\mathrm{n}_j = (k, n_{j+1}, \ldots, n_I)$. This equivalence follows because the indices of the entries in the $k$th row of $\boldsymbol{X}_{(j),\mathrm{n}_{j+1}}$ align exactly with those of the hierarchical block $\boldsymbol{x}_{\mathrm{n}_j}$ with $\mathrm{n}_j = (k, n_{j+1}, \ldots, n_I)$. Hence, the $k$th row of $\boldsymbol{X}_{(j),\mathrm{n}_{j+1}}$ and the hierarchical block $\boldsymbol{x}_{\mathrm{n}_j}$ with $\mathrm{n}_j = (k, n_{j+1}, \ldots, n_I)$ contain identical entries with identical order. Thus, the number of non-zero rows in $\boldsymbol{X}_{(j),\mathrm{n}_{j+1}} \left(\otimes_{i=j-1}^{1} \boldsymbol{H}_i\right)^{\top}$ is the number of hierarchical blocks $\{\boldsymbol{x}_{\mathrm{n}_j}\}$ (within the parent block defined by $\mathrm{n}_{j+1}$) that contain at least one non-zero element, which concludes the proof.

**Illustrative Example**: We provide an example in Figure 5 for the proof of Lemma 1. In Figure 5(a), we consider the same vector $\boldsymbol{x} \in \mathbb{R}^{40}$ with $I = 3, N_3 = 5, N_2 = 2, N_1 = 4, \bar{N} = 40$ as in Figure 1, and mark $s = 3$ nonzero entries using colored cubes. Figure 5(b), (c), and (d) illustrate the reordered tensor $\mathbf{X}$, its mode unfolding $\boldsymbol{X}_{(j)}$ with $j = 2$, and how $\boldsymbol{U}_2$ is computed, respectively.

The first step of the proof corresponds to Figure 5(c). To see why there is a block column structure, we first investigate how the unfolding matrix $\boldsymbol{X}_{(2)}$ is obtained. Since the unfolding tensor mode is $j = 2$, the row of the unfolding matrix $\boldsymbol{X}_{(2)}$ is indexed by $n_2 = 1, 2$. The column index $k$ is determined by $n_3$ and $n_1$ jointly as $k = 1 + (n_1 - 1) + N_1(n_3 - 1)$, according to the definition in Section 1. When $n_1$ increments by one, $k$ increases by 1; when $n_3$ increments by one, $k$ increases by $N_1$.

To arrange the columns of $\boldsymbol{X}_{(2)}$, we fix $n_3$ and let $n_1$ runs through $1, 2, 3, 4$, and then increase $n_3$ by one and let $n_1$ runs through $1, 2, 3, 4$ again, as shown in Figure 5(c). This indicates that $n_3$ indexes $\prod_{i=I}^{j+1} N_i = N_3 = 5$ column blocks, each containing $\prod_{i=1}^{j-1} N_i = N_1 = 4$ columns. Besides, the matrix $\left(\boldsymbol{I}_{\prod_{i=I}^{j+1} N_i} \otimes \left(\otimes_{i=j-1}^{1} \boldsymbol{H}_i\right)\right)^{\top}$ in this case reduces to the block diagonal matrix $(\boldsymbol{I}_5 \otimes \boldsymbol{H}_1)^{\top}$ in Figure 5(d), matching the column block structure of $\boldsymbol{X}_{(2)}$. Therefore, $\boldsymbol{U}_2$ can be divided into $N_3 = 5$ column blocks given by the product of the column blocks of $\boldsymbol{X}_{(2)}$ and $\boldsymbol{H}_1$, where each block of $\boldsymbol{U}_2$ is also indexed by an encapsulation $\mathrm{n}_3 = (n_3)$ for $n_3 \in [N_3]$.

For the second step, to understand why the number of nonzero rows in a column block indexed by $\mathrm{n}_{j+1}$ in $\boldsymbol{U}_j$ equals the number of nonzero blocks in $[\![\boldsymbol{x}_{\mathrm{n}_j}]\!]$ with $\mathrm{n}_j = (n_j, n_{j+1}, \cdots, n_I)$, we

examine Figure 5(d). Consider $n_3 = 4$. In $\boldsymbol{U}_2$, it corresponds to the fourth column block given by $\boldsymbol{X}_{(2),(4)}\boldsymbol{H}_1^\top$, where $\boldsymbol{X}_{(2),(4)}$ is the fourth column block of $\boldsymbol{X}_{(2)}$. Also, we have

$$[\![\boldsymbol{x}_{n_2}]\!] = [\![\boldsymbol{x}_{(1,4)}]\!] = \{\boldsymbol{x}_{(1,4)}, \boldsymbol{x}_{(2,4)}\}.$$

Each element of $[\![\boldsymbol{x}_{(1,4)}]\!]$ corresponds to one row of the column block $\boldsymbol{X}_{(2),(4)}$. Only $\boldsymbol{x}_{(2,4)}$ is nonzero leading to a nonzero row in $\boldsymbol{X}_{(2),(4)}\boldsymbol{H}_1^\top$. This demonstrates that the number of nonzero rows in a column block indexed by $n_{j+1}$ in $\boldsymbol{U}_j$ equals the number of nonzero blocks in $[\![\boldsymbol{x}_{n_j}]\!]$ with $n_j = (n_j, n_{j+1}, \cdots, n_I)$.

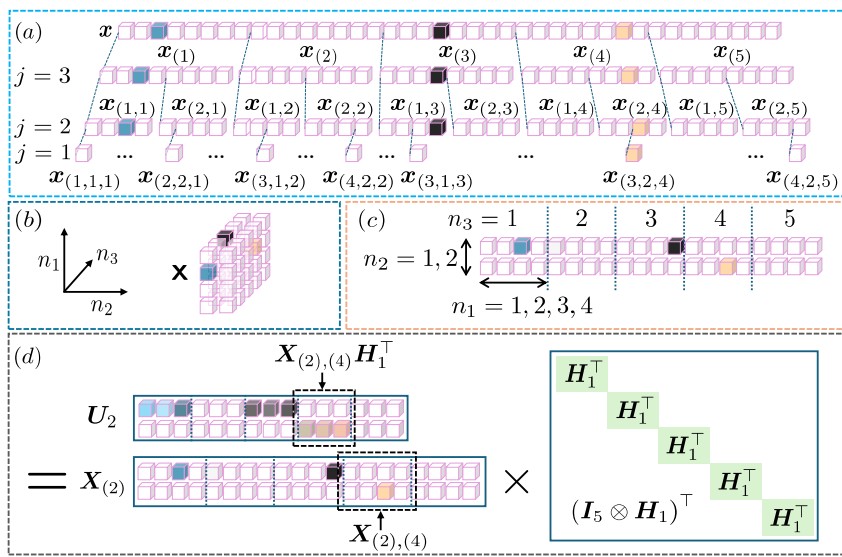

Figure 5: An illustrative example of the proof of Lemma 1 with $\boldsymbol{H}_1 \in \mathbb{R}^{3 \times 4}$.

## B    ILLUSTRATIONS OF DIFFERENT SPARSITY PATTERNS

In this section, we provide examples of sparsity patterns considered in this paper in Figure 6, using the same vector shown in Figure 1.

In Figure 6(a), we present the standard sparsity with $s = 3$. Three nonzero entries $\boldsymbol{x}_{(3,1,1)}$, $\boldsymbol{x}_{(4,1,3)}$, and $\boldsymbol{x}_{(3,2,4)}$ are arbitrarily positioned. Take $\boldsymbol{x}_{(3,2,4)}$ as an example. Its encapsulation $(3, 2, 4)$ means $\boldsymbol{x}_{(3,2,4)}$ is the third entry of the block indexed by encapsulation $(2, 4)$, i.e., $\boldsymbol{x}_{(2,4)}$, while $\boldsymbol{x}_{(2,4)}$ means it is the second block of the block indexed by encapsulation $\boldsymbol{x}_{(4)}$. Then $\boldsymbol{x}_{(4)}$ is the fourth block of vector $\boldsymbol{x}$.

In Figure 6(b), we show an example of $\boldsymbol{s} = (s_3, s_2, s_1) = (2, 1, 2)$ hierarchical sparsity. For the third level blocks, the set $[\![\boldsymbol{x}_{(2)}]\!]$ contains all blocks that share the same parent block as $\boldsymbol{x}_{(2)}$, meaning $[\![\boldsymbol{x}_{(2)}]\!] = \{\boldsymbol{x}_{(1)}, \boldsymbol{x}_{(2)}, \boldsymbol{x}_{(3)}, \boldsymbol{x}_{(4)}, \boldsymbol{x}_{(5)}\} = [\![\boldsymbol{x}_{(1)}]\!] = [\![\boldsymbol{x}_{(3)}]\!] = [\![\boldsymbol{x}_{(4)}]\!] = [\![\boldsymbol{x}_{(5)}]\!]$. Since $s_3 = 2$, according to the definition of Sparsity 2, $[\![\boldsymbol{x}_{(2)}]\!]$ contains at most $s_3 = 2$ nonzero blocks, which are $\boldsymbol{x}_{(2)}$ and $\boldsymbol{x}_{(4)}$. For the second level sparsity $s_2$, we take $\boldsymbol{x}_{(2)}$ and its child blocks $\boldsymbol{x}_{(1,2)}$ and $\boldsymbol{x}_{(2,2)}$ as an example. Since $s_2 = 1$, it means that $[\![\boldsymbol{x}_{(1,2)}]\!] = \{\boldsymbol{x}_{(1,2)}, \boldsymbol{x}_{(2,2)}\} = [\![\boldsymbol{x}_{(2,2)}]\!]$ contains at most $s_2$ nonzero block, which is $\boldsymbol{x}_{(1,2)}$. Similarly, $[\![\boldsymbol{x}_{(1,4)}]\!]$ contains at most $s_2$ nonzero block. For the first level sparsity $s_1 = 2$, we take $\boldsymbol{x}_{(1,2)}$ and its child blocks as an example. There should be at most $s_1 = 2$ nonzero blocks in the set of children of $\boldsymbol{x}_{(1,2)}$, which are $\boldsymbol{x}_{(2,1,2)}$ and $\boldsymbol{x}_{(3,1,2)}$ in $[\![\boldsymbol{x}_{(3,1,2)}]\!]$. Since this is the first level, a block corresponds to an individual element of $\boldsymbol{x}$.

Figure 6(c) illustrates the $\boldsymbol{s} = (s_3, s_2, s_1) = (2, 1, 2)$ Kronecker-supported sparsity with $\boldsymbol{b}_3 = [0, 1, 0, 1, 0]$, $\boldsymbol{b}_2 = [1, 0]$, and $\boldsymbol{b}_1 = [0, 1, 1, 0]$. Its support is then $\boldsymbol{b}_3 \otimes \boldsymbol{b}_2 \otimes \boldsymbol{b}_1$.

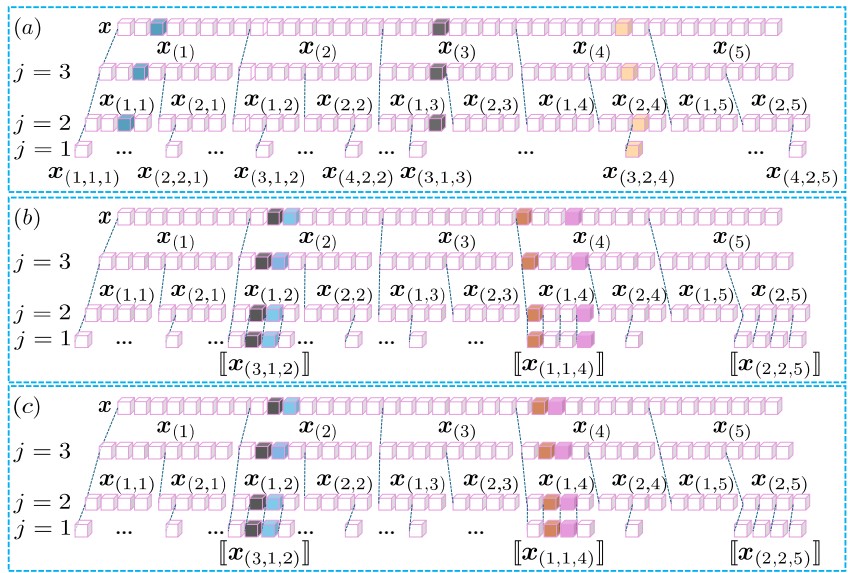

Figure 6: Examples of different sparsity patterns. (a) Standard sparsity. (b) $(2, 1, 2)$ hierarchical sparsity. (c) $(2, 1, 2)$ Kronecker-supported sparsity.

## C   PROOF OF LEMMA 2

Let $\boldsymbol{x}$ be an $s$-sparse vector. We denote $k_j$ as the total number of nonzero blocks within all $j$th-level blocks of $\boldsymbol{x}$. Clearly, $k_1 = s$ and $k_{I+1} = 1$. Then, each nonzero block in the $j + 1$th level can have at most $k_j - (k_{j+1} - 1)$ number of nonzero $j$th level blocks. This occurs in the most unbalanced case, where $k_{j+1} - 1$ blocks have only one nonzero $j$th level block while the remaining block has $k_j - (k_{j+1} - 1)$ nonzero $j$th level blocks. This observation leads to the upper bound for the sparsity level, $s_j \leq k_j - (k_{j+1} - 1)$, which yields $\sum_{i=1}^{I} s_i \leq s + (I - 1)$. So, any $s$ sparse vector $\boldsymbol{x} \in \cup_{\boldsymbol{s} \in f_{\mathrm{N}}(s)} \mathcal{S}_{\boldsymbol{s}}$.

## D   PROOF OF THEOREM 1

For any $\boldsymbol{x}$, we note that Equation 1 bounds $\|\boldsymbol{H}\boldsymbol{x}\|_2^2$. Following the hierarchical view, we note

$$\|\boldsymbol{H}\boldsymbol{x}\|_2^2 = \|\mathbf{X} \times_1 \boldsymbol{H}_1 \cdots \times_I \boldsymbol{H}_I\|_{\mathrm{F}}^2 = \|\boldsymbol{H}_I \boldsymbol{X}_{(I)} \left( \otimes_{i=I-1}^1 \boldsymbol{H}_i \right)^\top \|_{\mathrm{F}}^2.$$

Using the RIC of $\boldsymbol{H}_I$, we have

$$(1 - \delta_{s_I}) \|\boldsymbol{X}_{(I)} \left( \otimes_{i=I-1}^1 \boldsymbol{H}_i \right)^\top \|_{\mathrm{F}}^2 \leq \|\boldsymbol{H}\boldsymbol{x}\|_2^2 \leq (1 + \delta_{s_I}) \|\boldsymbol{X}_{(I)} \left( \otimes_{i=I-1}^1 \boldsymbol{H}_i \right)^\top \|_{\mathrm{F}}^2.$$

We also note that $\|\boldsymbol{X}_{(I)} \left( \otimes_{i=I-1}^1 \boldsymbol{H}_i \right)^\top \|_{\mathrm{F}}^2 = \|\boldsymbol{H}_{I-1} \boldsymbol{X}_{(I-1)} \left( \boldsymbol{I}_{N_I} \otimes \left( \otimes_{i=I-2}^1 \boldsymbol{H}_i \right) \right)^\top \|_{\mathrm{F}}^2$ due to the tensor folding and unfolding. Therefore, using RIC of $\boldsymbol{H}_{I-1}$, we arrive at

$$(1 - \delta_{s_I})(1 - \delta_{s_{I-1}})) \|\boldsymbol{X}_{(I-1)} \left( \boldsymbol{I}_{N_I} \otimes \left( \otimes_{i=I-2}^1 \boldsymbol{H}_i \right) \right)^\top \|_{\mathrm{F}}^2$$
$$\leq \|\boldsymbol{H}\boldsymbol{x}\|_2^2 \leq (1 + \delta_{s_I})(1 + \delta_{s_{I-1}}) \|\boldsymbol{X}_{(I)} \left( \otimes_{i=I-1}^1 \boldsymbol{H}_i \right)^\top \|_{\mathrm{F}}^2$$

Repeating these steps recursively, following the analysis in the hierarchical view, we obtain

$$\prod_{i=1}^{I}(1 - \delta_{s_i}) \|\boldsymbol{X}_{(1)} \left( \otimes_{i=I-1}^1 \boldsymbol{I}_{N_i} \right)^\top \|_{\mathrm{F}}^2 \leq \|\boldsymbol{H}\boldsymbol{x}\|_2^2 \leq \prod_{i=1}^{I}(1 + \delta_{s_i}) \|\boldsymbol{X}_{(1)} \left( \otimes_{i=I-1}^1 \boldsymbol{I}_{N_i} \right)^\top \|_{\mathrm{F}}^2.$$

Since $\boldsymbol{X}_{(1)} \left( \otimes_{i=I-1}^1 \boldsymbol{I}_{N_i} \right)^\top = \boldsymbol{X}_{(1)}$ and $\|\boldsymbol{X}_{(1)}\|_{\mathrm{F}}^2 = \|\boldsymbol{x}\|_2^2$,

$$\prod_{i=1}^{I}(1 - \delta_{s_i}) \|\boldsymbol{x}\|_2^2 \leq \|\boldsymbol{H}\boldsymbol{x}\|_2^2 \leq \prod_{i=1}^{I}(1 + \delta_{s_i}) \|\boldsymbol{x}\|_2^2.$$

Hence, we derive

$$\delta_{(\boldsymbol{s},\mathrm{N})}(\boldsymbol{H}) \leq \max\{1 - \prod_{i=1}^{I}(1 - \delta_{s_i})), \prod_{i=1}^{I}(1 + \delta_{s_i}) - 1\} = \prod_{i=1}^{I}(1 + \delta_{s_i}) - 1,$$

which completes the proof.

**Remark:** We note that the high-level proof strategy of our Theorem 1 and Roth et al. (2020, Theorem 4) is similar in that both aim to sequentially unwrap the effect of the Kronecker product. The key difference is that we employ tensor representations and operations such as tensor unfolding, enabling a straightforward, flip-operator-free proof. This formulation clearly demonstrates how the sparse signal $\boldsymbol{x}$ (or its tensor form $\mathbf{X}$ and its unfolding $\boldsymbol{X}_{(j)}$) is measured by factor matrix $\boldsymbol{H}_j$ through a linear transformation $\boldsymbol{X}_{(j)}\left(\boldsymbol{I}_{\prod_{i=I}^{j+1} N_i} \otimes \left(\otimes_{i=j-1}^{1} \boldsymbol{H}_i\right)\right)^{\top}$. The row sparsity of $\boldsymbol{X}_{(j)}\left(\boldsymbol{I}_{\prod_{i=I}^{j+1} N_i} \otimes \left(\otimes_{i=j-1}^{1} \boldsymbol{H}_i\right)\right)^{\top}$ is dictated by the sparsity of our hierarchical block partition as in Section 3. The aspect of this multi-stage measurement framework is missing in Roth et al. (2020). Thus, Roth et al. (2020, Theorem 4) focuses solely on hierarchical sparsity while our multi-stage framework provides a general perspective that defines generalized sparsity, where standard, hierarchical, and Kronecker-supported sparsity are special cases for analysis and recovery. This proof also explains why standard RIP cannot be improved beyond hierarchical sparsity, clarifies the maximum achievable sparsity level, and shows why the corresponding bounds are fundamentally tight. It further provides insight into why proofs for Kronecker-supported sparsity can be strengthened, drawing analogies to standard RIP and MMV analyses.

## E    COMPLETE RESULTS ON THE NUMBER OF MEASUREMENTS

In this section, we present the measurement bounds for unstructured $\boldsymbol{H}$ with different sparsity patterns. Let $\boldsymbol{H} \in \mathbb{R}^{\bar{M} \times \bar{N}}$ has independent and identically distributed standard Gaussian. For

$$\bar{M} = \mathcal{O}\left(s \ln(\frac{e\bar{N}}{s})\right)$$

where $c$ is a positive constant, $s$ sparse vectors can be recovered from the measurement of $\boldsymbol{H}$ with high probability (Foucart & Rauhut, 2013). Also, if

$$\bar{M} = \mathcal{O}\left(\sum_{i=1}^{I}\prod_{j=i}^{I} s_j \ln(\frac{eN_i}{s_i}) + \prod_{i=1}^{I} s_i\right),$$

$\boldsymbol{s}$ hierarchical sparse vectors can be recovered from the measurement of $\boldsymbol{H}$ with high probability (Roth et al., 2020). These two results lead to the discussed measurement bounds in Section 4.

## F    COMPLEXITY COMPARISON

We comprehensively analyze the complexity of our MSR algorithm to demonstrate the benefit of exploiting the Kronecker structure of $\boldsymbol{H}$ via the hierarchical view. We consider MSR combined with MMV-SBL (Wipf & Rao, 2007), SIHT (Blanchard et al., 2014), SHTP (Blanchard et al., 2014), and SOMP (Tropp et al., 2006) as sparse recovery algorithms, referred to as MSSBL, MSIHT, MSHTP, and MSOMP, respectively. We also use `Seq` and `Pl` to represent the *sequential* and *parallel* implementation of Equation 5. Assume $M_i$'s are $\mathcal{O}(M)$, $N_i$'s are $\mathcal{O}(N)$ for $i \in [I]$, and $I < M < N$. We compare the time and space complexities of our algorithms with those of other state-of-the-art algorithms. For the recovery of $s$ sparse vectors, we include SBL (Wipf & Rao, 2004), OMP, and KroOMP (Caiafa & Cichocki, 2013) as benchmarks. For the recovery of $\boldsymbol{s}$ hierarchically sparse vectors, HiHTP (Roth et al., 2020), IHT, and HTP are used as benchmarks. We note that only the exact implementation of HiHTP for $I = 2$ is given in (Roth et al., 2020). Regarding recovering $\boldsymbol{s}$ Kronecker-supported sparse vectors, we consider AM- and SVD-KroSBL (He & Joseph, 2025a) for benchmarking.

For the recovery of $s$ standard sparse vectors, our MSSBL and MSOMP substantially reduce both the time and space complexity compared to their traditional counterparts. In terms of time complexity,

our MSSBL ($\mathcal{O}(M^2N^I)$ for Seq and $\mathcal{O}(M^IN)$ for Pl) is superior than SBL ($\mathcal{O}(M^{2I}N^I)$), while the time complexity of MSOMP ($\mathcal{O}(MN^I)$ for Seq and $\mathcal{O}(M^IN)$ for Pl) is also lower than OMP with $\mathcal{O}(M^IN^I)$. Moreover, both MSSBL and MSOMP avoid $\mathcal{O}(MN)^I$ in space complexity and have $\mathcal{O}(M^{I-1}N)$ for Seq and $\mathcal{O}(MN^I)$ for Pl. Compared to KroOMP with time complexity $\mathcal{O}(MN^I)$ and space complexity $\mathcal{O}(N^I)$, MSOMP-Seq achieves the same time complexity but with a much lower space complexity $\mathcal{O}(M^{I-1}N)$. Alternatively, we can achieve a much lower time complexity $\mathcal{O}(M^IN)$ by parallel implementation, at the cost of a slightly higher space complexity of $\mathcal{O}(MN^I)$.

The computational gains are particularly significant in the context of structured sparsity. For both hierarchically sparse and Kronecker-supported sparse vectors, classical methods like IHT and HTP exhibit a time and space complexity of $\mathcal{O}(M^IN^I)$. Our MSIHT-Seq, MSHTP-Seq, and MSSBL-Seq have time complexity $\mathcal{O}(MN^I)$, $\mathcal{O}(MN^I)$, and $\mathcal{O}(M^2N^I)$, respectively, and $\mathcal{O}(M^{I-1}N)$ for space complexity. Compared to HiHTP, our MSSBL-Seq has the same time complexity $\mathcal{O}(M^2N^2)$ while MSSBL-Pl has a lower space complexity ($\mathcal{O}(MN^2)$ compared to $\mathcal{O}(M^2N^2)$ of HiHTP.

Similarly, for Kronecker-supported sparse recovery, when compared to AM-KroSBL and SVD-KroSBL, the MSSBL algorithm demonstrates lower time complexity from $\mathcal{O}(M^IN^I)$ to $\mathcal{O}(MN^I)$ and space complexity from $\mathcal{O}(M^IN^I)$ to $\mathcal{O}(N^I)$. MSIHT and MSHTP exhibit the same or even lower time and space complexities than MSSBL, which is lower than AM-KroSBL and SVD-KroSBL, demonstrating the superiority of our multi-stage framework. We list all the time and space complexity of the algorithms in Table 3. We use $R_{\text{EM}}$, $R_{\text{OMP}}$, $R_{\text{HTP}}$, $R_{\text{IHT}}$, and $R_{\text{AM}}$ to denote the number of EM, OMP, HTP, IHT, and AM iterations. These values can vary for different algorithms and experimental settings.

Table 3: Complexity of different algorithms in different sparse recovery problems.

| Method | Time Complexity | Space Complexity |
|---|---|---|
| Recovery of $s$ sparse vectors | | |
| MSSBL-Seq | $\mathcal{O}\big(R_{\text{EM}}(M^2N^I + MN^I)\big)$ | $\mathcal{O}(M^{I-1}N)$ |
| MSSBL-Pl | $\mathcal{O}\big(R_{\text{EM}}(IM^2N + M^IN)\big)$ | $\mathcal{O}(MN^I)$ |
| MSOMP-Seq | $\mathcal{O}\big(R_{\text{OMP}}MN^I + R_{\text{OMP}}^3 N^{I-1} + R_{\text{OMP}}^2 MN^{I-1}\big)$ | $\mathcal{O}(M^{I-1}N)$ |
| MSOMP-Pl | $\mathcal{O}\big(R_{\text{OMP}}M^IN + R_{\text{OMP}}^2 M^I + R_{\text{OMP}}^3 M^{I-1}\big)$ | $\mathcal{O}(MN^I)$ |
| KroOMP | $\mathcal{O}\big(R_{\text{OMP}}MN^I + R_{\text{OMP}}^2 M^I + R_{\text{OMP}}^2 MN + R_{\text{OMP}}^3\big)$ | $\mathcal{O}(N^I)$ |
| SBL | $\mathcal{O}\big(R_{\text{EM}}M^{2I}N^I\big)$ | $\mathcal{O}((MN)^I)$ |
| OMP | $\mathcal{O}\big(R_{\text{OMP}}(MN)^I + R_{\text{OMP}}^3 + R_{\text{OMP}}^2 M^I\big)$ | $\mathcal{O}((MN)^I)$ |
| Recovery of $s$ hierarchically sparse vectors | | |
| MSSBL-Seq | $\mathcal{O}\big(R_{\text{EM}}(M^2N^I + MN^I)\big)$ | $\mathcal{O}(M^{I-1}N)$ |
| MSSBL-Pl | $\mathcal{O}\big(R_{\text{EM}}(IM^2N + M^IN)\big)$ | $\mathcal{O}(MN^I)$ |
| MSHTP-Seq | $\mathcal{O}\big(R_{\text{HTP}}(MN^I + \max_i s_i^2 MN^{I-1})\big)$ | $\mathcal{O}(M^{I-1}N)$ |
| MSIHT-Seq | $\mathcal{O}\big(R_{\text{IHT}}MN^I\big)$ | $\mathcal{O}(M^{I-1}N)$ |
| HiHTP Roth et al. (2020) ($I = 2$) | $\mathcal{O}\big(R_{\text{HTP}}((s_1 s_2)^2 M^2 + (MN)^2)\big)$ | $\mathcal{O}((MN)^2)$ |
| IHT | $\mathcal{O}\big(R_{\text{IHT}}(MN)^I\big)$ | $\mathcal{O}((MN)^I)$ |
| HTP | $\mathcal{O}\big(R_{\text{HTP}}((MN)^I + (\prod_{i=1}^I s_i)^2 M^I)\big)$ | $\mathcal{O}((MN)^I)$ |
| Recovery of $s$ Kronecker-supported sparse vectors | | |
| MSSBL | $\mathcal{O}\big(R_{\text{EM}}(IM^2N + MN^I)\big)$ | $\mathcal{O}(N^I)$ |
| MSIHT | $\mathcal{O}\big(R_{\text{IHT}}MN^I\big)$ | $\mathcal{O}(N^I)$ |
| MSHTP | $\mathcal{O}\big(R_{\text{HTP}}MN^I + R_{\text{HTP}}M \sum_{i=1}^I s_i^2\big)$ | $\mathcal{O}(N^I)$ |
| MSOMP | $\mathcal{O}\big(R_{\text{OMP}}^3 N^{I-1} + R_{\text{OMP}}^2 MN^{I-1} + R_{\text{OMP}}MN^I\big)$ | $\mathcal{O}(N^I)$ |
| AM-KroSBL He & Joseph (2025a) | $\mathcal{O}\big(R_{\text{EM}}(R_{\text{AM}}IN^I + (MN)^I)\big)$ | $\mathcal{O}((MN)^I)$ |
| SVD-KroSBL He & Joseph (2025a) | $\mathcal{O}\big(R_{\text{EM}}(N^{I+1} + (MN)^I)\big)$ | $\mathcal{O}((MN)^I)$ |
| IHT | $\mathcal{O}\big(R_{\text{IHT}}(MN)^I\big)$ | $\mathcal{O}((MN)^I)$ |
| HTP | $\mathcal{O}\big(R_{\text{HTP}}((MN)^I + (\prod_{i=1}^I s_i)^2 M^I)\big)$ | $\mathcal{O}((MN)^I)$ |

Table 3 compares the complexity of different versions of MSR to different traditional compressed sensing algorithms. The conclusion can be extended to a more general case. Consider $N > M > 1$ and $I > 1$ with $M, N, I \in \mathbb{Z}$. In a prototypical compressed sensing problem $\boldsymbol{y}_{\text{p}} = \boldsymbol{H}_{\text{p}}\boldsymbol{x}_{\text{p}}$ with $\boldsymbol{H}_{\text{p}} \in \mathbb{R}^{M_{\text{p}} \times N_{\text{p}}}$ being a dense, unstructured measurement matrix, consider a general compressed sensing algorithm that has complexity $\mathcal{O}(M_{\text{p}}^a N_{\text{p}}^b)$ to recover $\boldsymbol{x}_{\text{p}}$ with $a, b \geq 1$. We note that $a, b \geq 1$ is a fair consideration since computing $\boldsymbol{H}_{\text{p}}\boldsymbol{x}_{\text{p}}$ already requires $\mathcal{O}(M_{\text{p}}N_{\text{p}})$. Then applying this algorithm to Equation 1 induces a time complexity of $\mathcal{O}(M^{aI}N^{bI})$. If we combine the same compressed sensing algorithm with our MSR, *regardless of the special structure or the MMV property of $\boldsymbol{U}_j$*, step 4 in Algorithm 1 is simply solving $N^{I-j}M^{j-1}$ compressed sensing subproblems where each

has $M$ measurements and $N$ unknowns, inducing a per level complexity of $\mathcal{O}(M^a N^b N^{I-j} M^{j-1})$. Considering all steps from $j = I$ to $1$, the total complexity is given by

$$\sum_{j=1}^{I} M^a N^b N^{I-j} M^{j-1} = M^{a-1} N^{b+I} \frac{M(N^I - M^I)}{N^I(N-M)} = M^a N^b \frac{N^I - M^I}{N - M}.$$

To compare to $M^{aI} N^{bI}$, we consider the ratio

$$\frac{M^a N^b (N^I - M^I)}{M^{aI} N^{bI}(N-M)} = \frac{N^I - M^I}{M^{a(I-1)} N^{b(I-1)}(N-M)} \overset{(i)}{\leq} \frac{N^I - M^I}{M^{I-1} N^{I-1}(N-M)} \overset{(ii)}{<} \frac{I}{M^{I-1}} \overset{(iii)}{\leq} 1,$$

where $(i)$ holds since the ratio is a decreasing function of $a$ and $b$, $(ii)$ is due to

$$N^I - M^I = (N-M)\sum_{i=1}^{I} N^{I-i} M^{i-1} < I N^{I-1}(N-M),$$

and $(iii)$ holds since $M^{I-1} \geq 2^{I-1} \geq I$ for $\forall I, M > 1, I, M \in \mathbb{Z}$, we conclude that $\frac{I}{M^{I-1}} \leq 1$, and thus $M^a N^b \frac{N^I - M^I}{N-M} < M^{aI} N^{bI}$ for any $N > M > 1$, $I > 1$, and $a, b \geq 1$ with $M, N, I \in \mathbb{Z}$. Hence, in general, our MSR has lower computational complexity than traditional compressed sensing algorithms when both are applied to Equation 1. We note that when $I = 1$, Equation 1 reduces to the traditional compressed sensing problem and MSR has identical complexity to that of a traditional compressed sensing algorithm.

## G  Proof of Theorem 2

Before the proof of Theorem 2, we introduce four aiding lemmas.

**Lemma 3.** *(Foucart & Rauhut, 2013, Lemma 6.16) Given a vector $\boldsymbol{v} \in \mathbb{R}^N$ and an index set $\mathcal{T} \subset [N]$, there is*

$$\|((\boldsymbol{I}_N - \boldsymbol{H}^\top \boldsymbol{H})\boldsymbol{v})_{\mathcal{T}}\|_2 \leq \delta_t \|\boldsymbol{v}\|_2,$$

*if the cardinality of the union of $\mathcal{T}$ and the support set of $\boldsymbol{v}$ is not exceeding $t$.*

**Lemma 4.** *(Foucart & Rauhut, 2013, Lemma 6.20) Given vector $\boldsymbol{n} \in \mathbb{R}^N$ and set $\mathcal{T} \subset [N]$ with cardinality not exceeding $s$, then*

$$\|(\boldsymbol{H}^\top \boldsymbol{n})_{\mathcal{T}}\|_2 \leq \sqrt{1 + \delta_s}\|\boldsymbol{n}\|_2.$$

**Lemma 5.** *For sparse matrix $\boldsymbol{X}$ with row support $\mathcal{T}$ with $\mathrm{card}(\mathcal{T}) \leq s$, and $\boldsymbol{N} \in \mathbb{R}^{M \times N}$, the sequence $\{\boldsymbol{X}^k\}$ defined by SIHT or SHTP for solving an MMV problem $\boldsymbol{Y} = \boldsymbol{H}\boldsymbol{X} + \boldsymbol{N}$ with $\boldsymbol{X}^0 = \boldsymbol{0}$, satisfies for any $k \geq 0$,*

$$\|\boldsymbol{X}^k - \boldsymbol{X}\|_F \leq \alpha^k \|\boldsymbol{X}\|_F + \tau \|\boldsymbol{N}\|_F,$$

*where*

$$\text{for SIHT: } \alpha = \sqrt{3}\delta_{3s}, \tau = \sqrt{3(1 + \delta_{2s})}\frac{1 - \alpha^k}{1 - \alpha}, \text{ and}$$

$$\text{for SHTP: } \alpha = \sqrt{\frac{2\delta_{3s}^2}{1 - \delta_{2s}^2}}, \tau = \frac{(\sqrt{2(1 - \delta_{2s})} + \sqrt{1 + \delta_s})(1 - \alpha^k)}{(1 - \delta_{2s})(1 - \alpha)}.$$

*Proof.* The proof closely follows the technique in Foucart & Rauhut (2013, Theorem 6.18) and Blanchard et al. (2014). Here, we extend the SMV case in Foucart & Rauhut (2013, Theorem 6.18) to the MMV case. In the MMV case, the thresholding operator retains the rows of $\boldsymbol{X}^k + \boldsymbol{H}^\top(\boldsymbol{Y} - \boldsymbol{H}\boldsymbol{X}^k)$ with the $s$ largest row $\ell_2$ norms, and then we have

$$\|\left(\boldsymbol{X}^k + \boldsymbol{H}^\top(\boldsymbol{Y} - \boldsymbol{H}\boldsymbol{X}^k)\right)_{\mathcal{T}}\|_F^2 \leq \|\left(\boldsymbol{X}^k + \boldsymbol{H}^\top(\boldsymbol{Y} - \boldsymbol{H}\boldsymbol{X}^k)\right)_{\mathcal{T}^{k+1}}\|_F^2.$$

Removing the common rows from both sides, we arrive at

$$\|\left(\boldsymbol{X}^k + \boldsymbol{H}^\top(\boldsymbol{Y} - \boldsymbol{H}\boldsymbol{X}^k)\right)_{\mathcal{T} \setminus \mathcal{T}^{k+1}}\|_F^2 \leq \|\left(\boldsymbol{X}^k + \boldsymbol{H}^\top(\boldsymbol{Y} - \boldsymbol{H}\boldsymbol{X}^k)\right)_{\mathcal{T}^{k+1} \setminus \mathcal{T}}\|_F^2.$$

IHT proceeds with $X^{k+1} = \left(X^k + H^\top \left(Y - HX^k\right)\right)_{\mathcal{T}^{k+1}}$. Since $(X^{k+1})_{\mathcal{T}\setminus\mathcal{T}^{k+1}} = 0$ and $(X)_{\mathcal{T}^{k+1}\setminus\mathcal{T}} = 0$, we get

$$\| \left(X - X^{k+1} + X^k - X + H^\top \left(Y - HX^k\right)\right)_{\mathcal{T}\setminus\mathcal{T}^{k+1}} \|_{\mathrm{F}}$$
$$\leq \| \left(X^k - X + H^\top \left(Y - HX^k\right)\right)_{\mathcal{T}^{k+1}\setminus\mathcal{T}} \|_{\mathrm{F}}.$$

Applying reverse triangle inequality to the left-hand side and rearranging, we arrive at

$$\| \left(X - X^{k+1}\right)_{\mathcal{T}\setminus\mathcal{T}^{k+1}} \|_{\mathrm{F}} \leq \| \left(X^k - X + H^\top \left(Y - HX^k\right)\right)_{\mathcal{T}^{k+1}\setminus\mathcal{T}} \|_{\mathrm{F}}$$
$$+ \| \left(X^k - X + H^\top \left(Y - HX^k\right)\right)_{\mathcal{T}\setminus\mathcal{T}^{k+1}} \|_{\mathrm{F}}$$
$$\leq \sqrt{2}\| \left(X^k - X + H^\top \left(Y - HX^k\right)\right)_{\mathcal{T}\Delta\mathcal{T}^{k+1}} \|_{\mathrm{F}},$$

where $\mathcal{T}\Delta\mathcal{T}^{k+1} = (\mathcal{T} \setminus \mathcal{T}^{k+1}) \cup (\mathcal{T}^{k+1} \setminus \mathcal{T})$ denoting the symmetric difference of the sets $\mathcal{T}$ and $\mathcal{T}^{k+1}$. Therefore, we obtain the error in the $k$th iteration as

$$\|X^{k+1} - X\|_{\mathrm{F}}^2 = \| \left(X^{k+1} - X\right)_{\mathcal{T}^{k+1}} \|_{\mathrm{F}}^2 + \| \left(X^{k+1} - X\right)_{\mathcal{T}\setminus\mathcal{T}^{k+1}} \|_{\mathrm{F}}^2$$
$$= \| \left(X^k + H^\top \left(Y - HX^k\right) - X\right)_{\mathcal{T}^{k+1}} \|_{\mathrm{F}}^2 + \| \left(X^{k+1} - X\right)_{\mathcal{T}\setminus\mathcal{T}^{k+1}} \|_{\mathrm{F}}^2$$
$$\leq \| \left(X^k + H^\top \left(Y - HX^k\right) - X\right)_{\mathcal{T}^{k+1}} \|_{\mathrm{F}}^2$$
$$+ 2\| \left(X^k - X + H^\top \left(Y - HX^k\right)\right)_{\mathcal{T}\Delta\mathcal{T}^{k+1}} \|_{\mathrm{F}}^2$$
$$\leq 3\| \left(X^k - X + H^\top \left(Y - HX^k\right)\right)_{\mathcal{T}\cup\mathcal{T}^{k+1}} \|_{\mathrm{F}}^2.$$

Considering $Y = HX + N$, we then have

$$\|X^{k+1} - X\|_{\mathrm{F}} \leq \sqrt{3}\| \left(X^k - X + H^\top \left(Y - HX^k\right)\right)_{\mathcal{T}\cup\mathcal{T}^{k+1}} \|_{\mathrm{F}}$$
$$= \sqrt{3}\| \left(\left(I - H^\top H\right) \left(X^k - X\right) + H^\top N\right)_{\mathcal{T}\cup\mathcal{T}^{k+1}} \|_{\mathrm{F}}$$
$$\leq \sqrt{3}\| \left(\left(I - H^\top H\right) \left(X^k - X\right)\right)_{\mathcal{T}\cup\mathcal{T}^{k+1}} \|_{\mathrm{F}} + \sqrt{3}\| \left(H^\top N\right)_{\mathcal{T}\cup\mathcal{T}^{k+1}} \|_{\mathrm{F}}$$
$$\leq \sqrt{3}\delta_{3s}\|X^k - X\|_{\mathrm{F}} + \sqrt{3(1 + \delta_{2s})}\|N\|_{\mathrm{F}},$$

where the last step is the direct consequence of Lemma 3 and Lemma 4. To see this, we note

$$\| \left(\left(I - H^\top H\right) \left(X^k - X\right)\right)_{\mathcal{T}\cup\mathcal{T}^{k+1}} \|_{\mathrm{F}}^2 = \sum_n \| \left(\left(I - H^\top H\right) [X^k - X]_n\right)_{\mathcal{T}\cup\mathcal{T}^{k+1}} \|_2^2$$
$$\leq \delta_{3s}^2 \sum_n \|[X^k - X]_n\|_2^2 = \delta_{3s}^2 \|X^k - X\|_{\mathrm{F}}^2,$$

where $[X^k - X]_n$ is the $n$th column of matrix $X^k - X$. We can derive similar argument for $\sqrt{1 + \delta_{2s}}\|N\|_{\mathrm{F}}$. Conclusion for HTP has been given in Blanchard et al. (2014, Theorem 3). This concludes the proof.

$\square$

**Lemma 6.** *For the sparse recovery problem in the stage of unfolding $j$th $(j \leq I - 1)$ mode of tensor*

$$\mathsf{T} = \mathsf{X} \times_1 H_1 \times_2 H_2 \cdots \times_j H_j \times_{j+1} I_{N_{j+1}} \cdots \times_I I_{N_I} + \mathsf{N},$$

*where the sparse tensor $\mathsf{X}$ corresponds to $s$ standard sparse $x$ or $s$ hierarchically sparse $x$. Its $j$th mode unfolding leads to*

$$T_{(j)} = H_j U_j + N_{(j)} = H_j X_{(j)} \left(I_{\prod_{i=I}^{j+1} N_i} \otimes \left(\otimes_{i=j-1}^1 H_i\right)\right)^\top + N_{(j)}. \tag{6}$$

*Then the estimate of $U_j$, denoted as $\tilde{U}_j$ and obtained through IHT or HTP, satisfies*

$$\|[\tilde{U}_j]_{\mathrm{n_{j+1}}} - [U_j]_{\mathrm{n_{j+1}}}\|_{\mathrm{F}} \leq \alpha_j^k \|[U_j]_{\mathrm{n_{j+1}}}\|_{\mathrm{F}} + \tau_j \|[\tilde{U}_{j+1} - U_{j+1}]_{\mathrm{n_{j+2}}}\|_{\mathrm{F}}, \tag{7}$$

*where $[\tilde{U}_j]_{\mathrm{n_{j+1}}}$ and $[U_j]_{\mathrm{n_{j+1}}} := [X_{(j)}]_{\mathrm{n_{j+1}}} \left(\otimes_{i=j-1}^1 H_i\right)^\top$ denote the $\mathrm{n_{j+1}}$th column block of $\tilde{U}_j$ and $U_j$, respectively. Here, encapsulation $\mathrm{n_{j+1}} := (n_{j+1}, \cdots, n_{I-1}, n_I)$ is the index for the column block. The indices $\{n_i\}_{i=j+2}^I$ in encapsulation $\mathrm{n_{j+2}} := (n_{j+2}, \cdots, n_{I-1}, n_I)$ have the same value as the indices $\{n_i\}_{i=j+2}^I$ in encapsulation $\mathrm{n_{j+1}}$, i.e., the block indexed by $\mathrm{n_{j+2}}$ should be a parent block of the block indexed by $\mathrm{n_{j+1}}$. Constants $\mu_j$ and $\tau_j$ depend on the iteration number $k$ and matrix $H_j$.*

*Proof.* According to Lemma 1, we solve Equation 6 by separating it into $\prod_{i=I}^{j+1} N_i$ MMV problem, where each MMV problem is indexed by an encapsulation $\mathrm{n}_{j+1}$. Suppose we consider a fixed encapsulation $\mathrm{n}_{j+1}^* = (n_{j+1}^*, \cdots, n_I^*)$, and consider the MMV problem indexed by $\mathrm{n}_{j+1}^*$ as

$$[\boldsymbol{T}_{(j)}]_{\mathrm{n}_{j+1}^*} = \boldsymbol{H}_j [\boldsymbol{U}_j]_{\mathrm{n}_{j+1}^*} + [\boldsymbol{N}_{(j)}]_{\mathrm{n}_{j+1}^*} = \boldsymbol{H}_j [\boldsymbol{X}_{(j)}]_{\mathrm{n}_{j+1}^*} \left( \otimes_{i=j-1}^{1} \boldsymbol{H}_i \right)^\top + [\boldsymbol{N}_{(j)}]_{\mathrm{n}_{j+1}^*}.$$

According to Lemma 5 and denoting the solution as $[\tilde{\boldsymbol{U}}_j]_{\mathrm{n}_{j+1}^*}$ with $k$ IHT or HTP iterations, we have

$$\|[\tilde{\boldsymbol{U}}_j]_{\mathrm{n}_{j+1}^*} - [\boldsymbol{U}_j]_{\mathrm{n}_{j+1}^*}\|_\mathrm{F} \le \alpha_j^k \|[\boldsymbol{U}_j]_{\mathrm{n}_{j+1}^*}\|_\mathrm{F} + \tau_j \|[\boldsymbol{N}_{(j)}]_{\mathrm{n}_{j+1}^*}\|_\mathrm{F},$$

where $\mu_j$ and $\tau_j$ relate to the RICs of matrix $\boldsymbol{H}_j$. The only step left is to bound $\|[\boldsymbol{N}_{(j)}]_{\mathrm{n}_{j+1}^*}\|_\mathrm{F}$ using $[\boldsymbol{E}_{j+1}]_{\mathrm{n}_{j+2}^*}$, where $E_{j+1} = \tilde{\boldsymbol{U}}_{j+1} - \boldsymbol{U}_{j+1}$.

We recall that Equation 6 is obtained by unfolding the measurement tensor formed from the matrix $\tilde{\boldsymbol{U}}_{j+1} = \boldsymbol{U}_{j+1} + \boldsymbol{E}_{j+1}$ along its $j$th mode. Hence, $\boldsymbol{N}_{(j)}$ is simply reordered version of $\boldsymbol{E}_{j+1}$. Consequently, the entries of the matrix $[\boldsymbol{N}_{(j)}]_{\mathrm{n}_{j+1}^*}$ are essentially entries of the $n_{j+1}^*$th row of $[\boldsymbol{E}_{j+1}]_{\mathrm{n}_{j+2}^*}$, leading to

$$\|[\boldsymbol{N}_{(j)}]_{\mathrm{n}_{j+1}^*}\|_\mathrm{F} \le \|[\boldsymbol{E}_{j+1}]_{\mathrm{n}_{j+2}^*}\|_\mathrm{F},$$

and we arrive at the desired result.

To elaborate, we first investigate the indices of the entries of $[\boldsymbol{N}_{(j)}]_{\mathrm{n}_{j+1}^*}$. The entries of the $n_j^*$th row of matrix $[\boldsymbol{N}_{(j)}]_{\mathrm{n}_{j+1}^*}$ are obtained by $i$) fixing $n_j = n_j^*$ (row index) and $n_{j+1} = n_{j+1}^*, \cdots, n_I = n_I^*$ (encapsulation), and $ii$) running $n_1, \cdots, n_{j-1}$ from one till $N_1, \cdots, N_{j-1}$, respectively. Thus, the entries of matrix $[\boldsymbol{N}_{(j)}]_{\mathrm{n}_{j+1}^*}$ can be obtained by $i$) fixing $n_{j+1} = n_{j+1}^*, \cdots, n_I = n_I^*$, $ii$) running $n_1, \cdots, n_{j-1}$ from one till $N_1, \cdots, N_{j-1}$, respectively, and $iii$) running $n_j = 1, \cdots, N_j$ (going over all rows). Given such knowledge, we start investigating the $n_{j+1}^*$th row of matrix $[\boldsymbol{E}_{j+1}]_{\mathrm{n}_{j+2}^*}$. The entries of this row are obtained by $i$) fixing $n_{j+1} = n_{j+1}^*$ (row index), $ii$) fixing $n_{j+2} = n_{j+2}^*, \cdots, n_I = n_I^*$ (fixed encapsulation), and $iii$) running $n_1, \cdots, n_j$ from one till $N_1, \cdots, N_j$, respectively. By comparing how indices are arranged, we can see that the entries of the matrix $[\boldsymbol{N}_{(j)}]_{\mathrm{n}_{j+1}^*}$ are essentially entries of the $n_{j+1}^*$th row of $[\boldsymbol{E}_{j+1}]_{\mathrm{n}_{j+2}^*}$, inferring $\|[\boldsymbol{N}_{(j)}]_{\mathrm{n}_{j+1}^*}\|_\mathrm{F} \le \|[\boldsymbol{N}_{j+1}]_{\mathrm{n}_{j+2}^*}\|_\mathrm{F}$. $\qquad\square$

As we have described before, Equation 6 is solved through multiple independent MMV problems. Thus, the error bound is also given regarding each individual MMV problem. Further, not only the $j$th step, but also the $j + 1$th step is solved through multiple independent MMV problems. Thus, we do not have the upper bound for $\boldsymbol{E}_{j+1}$ in Equation 4 as a whole but only the upper bound for each column block $[\boldsymbol{E}_{j+1}]_{\mathrm{n}_{j+2}}$. Fortunately, since all the noise entries in the $j$th step are contained as one single row of the noise block in the previous step, having the upper bound for each column block $[\boldsymbol{E}_{j+1}]_{\mathrm{n}_{j+2}}$ is sufficient to derive the noise bound for the $j$th step, which is shown in Lemma 6 and Equation 7.

Now, we proceed to the proof of Theorem 2. Generally speaking, Theorem 2 is obtained by recursively applying Lemma 6. Particularly, focusing on the $\boldsymbol{s}$ hierarchical sparse vectors, for the last step, i.e., *the first mode unfolding*, we solve

$$\boldsymbol{T}_{(1)} = \boldsymbol{H}_1 \boldsymbol{X}_{(1)} + \boldsymbol{N}_{(1)},$$

leading to $\prod_{j=2}^{I} N_j$ SMV problems. They are SMV because there is only one column in each column block, and hence the MMV problem reduces to the SMV problem. Lemma 6 indicates that

$$\begin{aligned}
\|\tilde{\boldsymbol{U}}_1 - \boldsymbol{U}_1\|_\mathrm{F} &\le \sum_{n_2, \cdots, n_I} \|[\tilde{\boldsymbol{U}}_1]_{\mathrm{n}_2} - [\boldsymbol{U}_1]_{\mathrm{n}_2}\|_\mathrm{F} \le \sum_{n_2, \cdots, n_I} \alpha_1^k \|[\boldsymbol{U}_1]_{\mathrm{n}_2}\|_\mathrm{F} + \tau_1 \|[\boldsymbol{E}_2]_{\mathrm{n}_3}\|_\mathrm{F} \\
&\le \sum_{n_2, \cdots, n_I} \alpha_1^k \|[\boldsymbol{U}_1]_{\mathrm{n}_2}\|_\mathrm{F} + \tau_1 \left( \alpha_2^k \|[\boldsymbol{U}_2]_{\mathrm{n}_3}\|_\mathrm{F} + \tau_2 \|[\boldsymbol{E}_3]_{\mathrm{n}_4}\|_\mathrm{F} \right) \\
&\le \sum_{n_2, \cdots, n_I} \left( \sum_{i=1}^{I} \prod_{j=1}^{i-1} \tau_j \alpha_i^k \|[\boldsymbol{U}_i]_{\mathrm{n}_{i+1}}\|_\mathrm{F} + \prod_{i=1}^{I} \tau_i \|[\boldsymbol{E}_I]_{\mathrm{n}_{I+1}}\|_\mathrm{F} \right).
\end{aligned}$$

We note that $I + 1$th level contains only one block, leading to $[\boldsymbol{E}_I]_{\mathrm{n}_{I+1}} = \boldsymbol{E}_I$. Using the relation $\tilde{\boldsymbol{U}}_I = \boldsymbol{U}_I + \boldsymbol{E}_I$ and Lemma 5 leads to $\|\boldsymbol{E}_I\|_{\mathrm{F}} \leq \alpha_I^k\|\boldsymbol{U}_I\|_{\mathrm{F}} + \tau_I\|\boldsymbol{N}_{(1)}\|_{\mathrm{F}}$. This concludes the proof for $\boldsymbol{s}$ hierarchical sparsity. For $s$ standard sparsity, the upper bound for all $s$ standard sparse vectors is the worst upper bound among all possible $\boldsymbol{s}$ corresponding to the sparsity level $s$. Therefore, taking the maximum over $\forall \boldsymbol{s} \in f_{\mathrm{N}}(s)$ concludes the proof.

For $\boldsymbol{s}$ Kronecker-supported sparsity, since the support is shared among different blocks in the same level, it is unnecessary to introduce multiple MMV problems, but to solve only one MMV problem. Thus, recursively applying Lemma 5 leads to the final result. For the last step, we solve

$$\boldsymbol{T}_{(1)} = \boldsymbol{H}_1\boldsymbol{X}_{(1)} + \boldsymbol{N}_{(1)},$$

which leads to the following relations,

$$
\begin{aligned}
\|\tilde{\boldsymbol{U}}_1 - \boldsymbol{U}_1\|_{\mathrm{F}} &\leq \alpha_1^k\|\boldsymbol{U}_1\|_{\mathrm{F}} + \tau_1\|\boldsymbol{N}_{(1)}\|_{\mathrm{F}} \\
&\leq \alpha_1^k\|\boldsymbol{X}_{(1)}\|_{\mathrm{F}} + \tau_1\left(\alpha_2^k\|\boldsymbol{U}_2\|_{\mathrm{F}} + \tau_2\|\boldsymbol{N}_{(2)}\|_{\mathrm{F}}\right) \\
&\leq \sum_{i=1}^{I}\prod_{j=1}^{i-1}\alpha_i^k\tau_j\|\boldsymbol{U}_i\|_{\mathrm{F}} + \prod_{i=1}^{I}\tau_i\|\mathbf{N}\|_{\mathrm{F}}.
\end{aligned}
$$

Thus, the proof is complete.

## H ADDITIONAL NUMERICAL EVALUATIONS

### H.1 COMPREHENSIVE STRUCTURED SPARSE VECTOR RECOVERY PERFORMANCE

This section presents a more comprehensive evaluation of our MSR framework compared to the state-of-the-art, consisting of complete results of Section 5 and a new set of results where we vary the number of measurements with a fixed SNR. We also include a new metric named support recovery rate (SRR) defined as

$$\mathrm{SRR} = \frac{|\operatorname{supp}(\hat{\boldsymbol{x}}) \cap \operatorname{supp}(\boldsymbol{x})|}{|\operatorname{supp}(\hat{\boldsymbol{x}}) \cup \operatorname{supp}(\boldsymbol{x})|},$$

where $\operatorname{supp}(\cdot)$ returns the set of positions of the nonzero entries of the argument vector, $|\cdot|$ returns the cardinality of the argument set, $\hat{\boldsymbol{x}}$ is the estimated sparse vector, and $\boldsymbol{x}$ is the ground truth.

Table 4: Average runtime. A complete version of Table 1. **Bold**: the best result.

| SNR | 3 dB | 7 dB | 11 dB | 15 dB | 19 dB | 23 dB |
|---|---|---|---|---|---|---|
| Recovery of $s$ sparse vectors | | | | | | |
| MSOMP-`Seq` | **0.4256** | **0.4119** | 0.3827 | 0.3329 | 0.2204 | **0.0568** |
| KroOMP | 130.5405 | 108.0526 | 76.6942 | 39.9844 | 11.5774 | 0.7525 |
| MSSBL-`Seq` | 1.8191 | 1.1016 | 0.5758 | **0.2218** | **0.1417** | 0.1141 |
| MSSBL-`Pl` | 0.4517 | 4.9263 | **0.2658** | 1.8292 | 0.1531 | 0.1281 |
| Recovery of $s$ hierarchically sparse vectors | | | | | | |
| MSSBL-`Seq` | 2.4930 | 2.0134 | 1.2501 | 0.6102 | 0.1965 | 0.1112 |
| MSSBL-`Pl` | 0.4962 | 0.4607 | 1.3664 | 0.2513 | 0.1447 | 0.1081 |
| MSHTP-`Seq` | **0.0379** | **0.0305** | **0.0297** | **0.0247** | **0.0186** | **0.0168** |
| HiHTP | 0.6512 | 0.5493 | 0.5204 | 0.5444 | 0.4398 | 0.4574 |
| HTP | 2.2436 | 1.7170 | 1.3256 | 0.8450 | 0.8264 | 0.5311 |
| MSIHT-`Seq` | 0.0500 | 0.0510 | 0.0532 | 0.0509 | 0.0450 | 0.0434 |
| IHT | 8.2437 | 8.2412 | 8.2554 | 8.2917 | 8.2889 | 8.2789 |
| Recovery of $s$ Kronecker-supported sparse vectors | | | | | | |
| MSOMP | 0.0042 | 0.0041 | 0.0040 | 0.0038 | 0.0026 | 0.0015 |
| MSHTP | 0.0011 | 0.0010 | 0.0011 | 0.0010 | 0.0010 | 0.0010 |
| MSSBL | 0.0728 | 0.0587 | 0.0447 | 0.0279 | 0.0119 | 0.0051 |
| SVD-KroSBL | 37.1233 | 26.9816 | 14.2405 | 8.6036 | 5.4067 | 4.0681 |
| AM-KroSBL | 55.9532 | 63.4676 | 75.9727 | 74.5840 | 51.7089 | 34.1331 |
| HTP | 0.9772 | 0.8347 | 0.4709 | 0.3465 | 0.2339 | 0.2323 |
| MSIHT | **0.0008** | **0.0007** | **0.0007** | **0.0007** | **0.0007** | **0.0007** |
| IHT | 6.0771 | 6.0760 | 6.0763 | 6.0690 | 6.0677 | 6.0535 |
| KSHTP | 0.1018 | 0.0730 | 0.0665 | 0.0811 | 0.0865 | 0.0881 |

We show a complete version of Figure 2 in Figure 7. We use Tensorlab (Vervliet et al., 2016) for tensor operation and `Seq` and `Pl` to represent the *sequential* and *parallel* (`parfor` function in

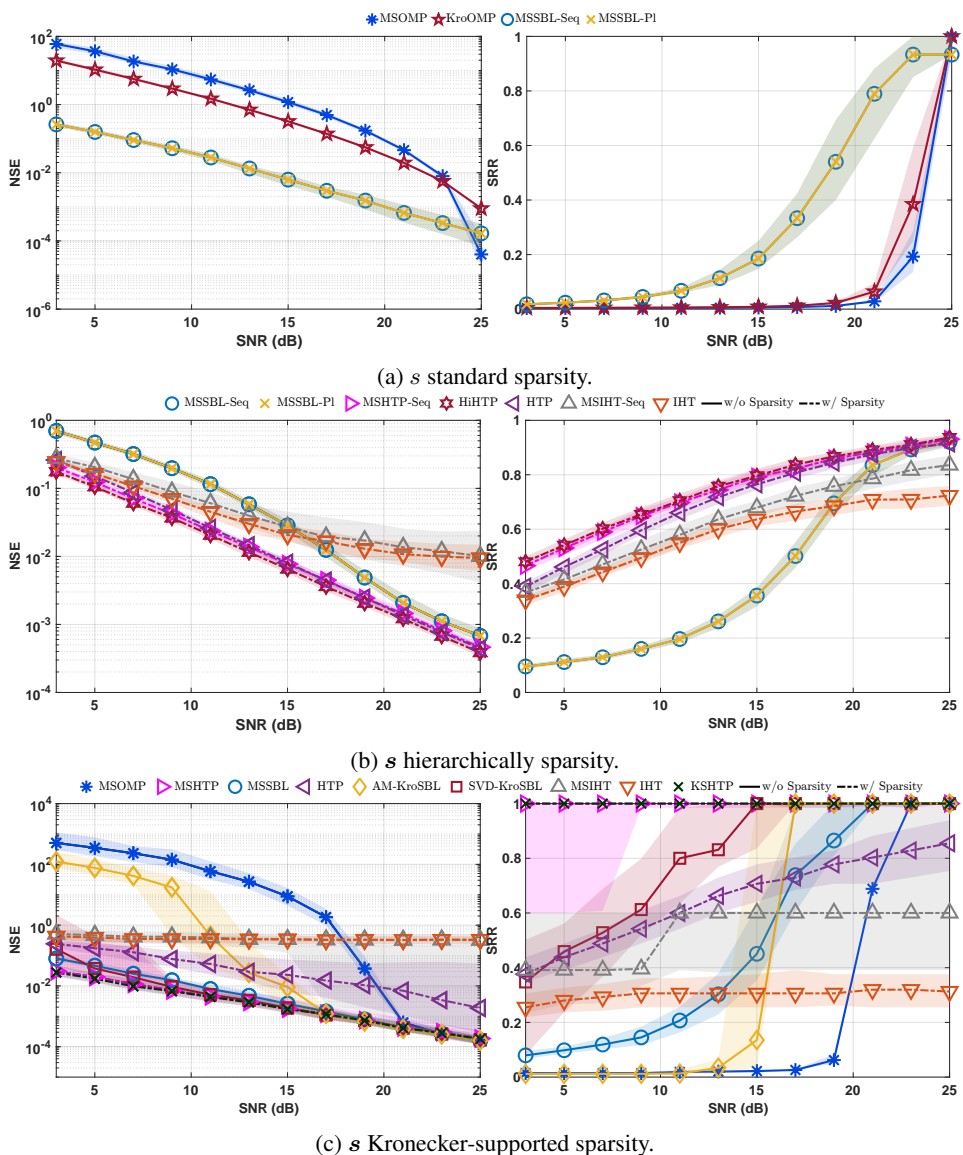

(a) $s$ standard sparsity.

(b) $s$ hierarchically sparsity.

(c) $s$ Kronecker-supported sparsity.

Figure 7: NSE and SRR as functions of SNR. A complete version of Figure 2.

Matlab (Inc., 2024)) implementation of Equation 5; they have the same recovery performance but different runtimes.

For fairness, we cap the number of EM iterations for SBL-based methods (MSSBL, AM-KroSBL, and SVD-KroSBL) to two hundred, for HTP based methods to one hundred, and for all IHT based methods to two hundred. For HTP-based algorithms, we stop the iterations if the detected support remains the same in two consecutive iterations (Foucart, 2011), while IHT-based algorithms are terminated when the normalized difference between two consecutive estimations is smaller than $10^{-6}$. For OMP-based algorithms, we stop when the norm of the residual is smaller than $\epsilon\|\boldsymbol{y}\|_2$. Here, the coefficient $\epsilon = 0.05$ is fixed for all OMP-based algorithms, which is empirically determined. We also prune small entries in hyperparameters for faster convergence for SBL-based algorithms.

In the recovery of $s$ standard sparse vectors, compared to Figure 2a, Figure 7a includes both the sequential and parallel implementation of our MSSBL. Regardless of different runtimes as in Table 4, sequential and parallel implementations provide identical NSE and SRR results. Regarding runtime, MSSBL-`Pl` is only faster than MSSBL-`Seq` in low SNR cases. This is because in high SNR cases, the parallel overhead dominates, including data transfer and communication cost. As we see in Fig-

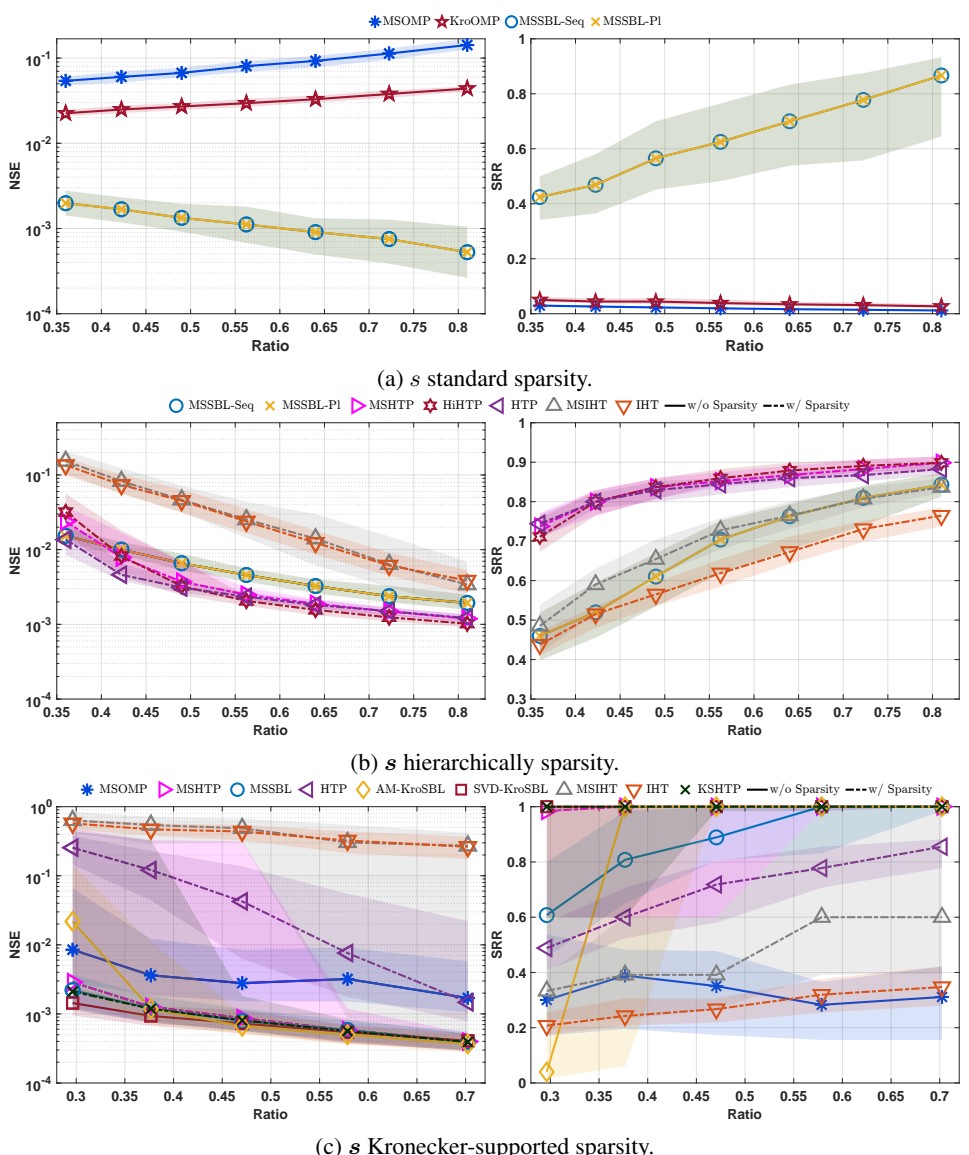

(a) $s$ standard sparsity.

(b) $s$ hierarchically sparsity.

(c) $s$ Kronecker-supported sparsity.

Figure 8: NSE and SRR as functions of the number of measurements.

Table 5: Average runtime in seconds. **Bold**: the best result.

| $M$ | 48 | 52 | 56 | 60 | 64 | 68 | 72 |
|---|---|---|---|---|---|---|---|
| \multicolumn{8}{c}{Recovery of $s$ sparse vectors} |
| MSOMP-Seq | **0.0735** | **0.0951** | **0.1265** | 0.1574 | 0.1937 | 0.2312 | 0.2918 |
| KroOMP | 0.7139 | 1.1941 | 2.1137 | 3.7952 | 7.4092 | 12.5439 | 22.4149 |
| MSSBL-Seq | 0.1726 | 0.1743 | 0.1439 | **0.1364** | **0.1332** | **0.1368** | **0.1321** |
| MSSBL-Pl | 0.1412 | 0.1464 | 0.1508 | 0.1517 | 0.1498 | 0.1527 | 0.3298 |

| $M$ | 48 | 52 | 56 | 60 | 64 | 68 | 72 |
|---|---|---|---|---|---|---|---|
| \multicolumn{8}{c}{Recovery of $s$ hierarchically sparse vectors} |
| MSSBL-Seq | 0.2978 | 0.2594 | 0.2031 | 0.1692 | 0.1552 | 0.1407 | 0.1274 |
| MSSBL-Pl | 0.1734 | 0.1614 | 0.1467 | 0.1363 | 0.1299 | 0.1248 | 0.1197 |
| MSHTP-Seq | **0.0204** | **0.0198** | **0.0190** | **0.0191** | **0.0178** | **0.0170** | **0.0168** |
| HiHTP | 0.3661 | 0.3622 | 0.3691 | 0.4189 | 0.4181 | 0.4599 | 0.5127 |
| HTP | 0.3071 | 0.3641 | 0.4753 | 0.4980 | 0.6407 | 0.6867 | 0.4566 |
| MSIHT-Seq | 0.0527 | 0.0502 | 0.0479 | 0.0457 | 0.0449 | 0.0421 | 0.0416 |
| IHT | 4.7097 | 5.4549 | 6.2188 | 7.1292 | 8.3298 | 9.1231 | 10.1874 |

| $M$ | 12 | 13 | 14 | 15 | 16 |
|---|---|---|---|---|---|
| \multicolumn{6}{c}{Recovery of $s$ Kronecker-supported sparse vectors} |
| MSOMP | 0.0017 | 0.0017 | 0.0019 | 0.0020 | 0.0023 |
| MSHTP | 0.0010 | 0.0010 | 0.0011 | 0.0011 | 0.0013 |
| MSSBL | 0.0187 | 0.0141 | 0.0118 | 0.0092 | 0.0083 |
| SVD-KroSBL | 5.3134 | 4.8928 | 4.7247 | 5.0780 | 5.3094 |
| AM-KroSBL | 52.0891 | 52.4779 | 49.6836 | 47.0243 | 42.6749 |
| HTP | 0.0715 | 0.1113 | 0.1658 | 0.2508 | 0.2745 |
| MSIHT | **0.0009** | **0.0008** | **0.0009** | **0.0007** | **0.0007** |
| IHT | 3.4581 | 4.0502 | 4.9489 | 6.0187 | 7.2859 |
| KSHTP | 0.1184 | 0.0908 | 0.1048 | 0.0885 | 0.0829 |

ure 3, when the computation cost dominates, there is a significant gain in computation time, as a trade-off for memory usage. In the recovery of $s$ hierarchical sparse vectors, compared to Figure 2b, Figure 7b includes the performance of MSSBL. MSSBL exhibits a worse performance in low SNR scenario because it does not require the true sparsity level $s$ as an input, while for IHT/HTP-based algorithms, this prior knowledge is necessary. However, MSSBL is still able to offer a comparable performance in high SNR scenarios, making it a powerful candidate when the prior knowledge $s$ is absent. In the recovery of $s$ Kronecker-supported sparse vectors, compared to Figure 2c, we include IHT/HTP-based algorithms in Figure 7c. KSHTP is the algorithm we explained in Equation HTP. Although the thresholding operator for Kronecker support is not optimal, KSHTP still offers the best SRR performance, followed by MSHTP. MSIHT has the least runtime, which is four orders less than its classic counterpart IHT. Overall, Figure 7 and Table 4 demonstrate that our MSR framework can offer similar or better performance with significantly reduced runtime.

We next evaluate the performance of different algorithms by fixing the SNR and varying the number of measurements. The setting is as follows. For the $s$ standard sparsity, we opt for $\boldsymbol{H} = \otimes_{i=I}^{1} \boldsymbol{H}_i$ with $I = 2$, and set $M = \{48, 52, 56, \cdots, 72\}$ and $N = 80$. The entries of $\boldsymbol{H}_i$ and the nonzero entries of $\boldsymbol{x}$ are drawn independently from the standard normal distribution. We set $s = 15$, and the support is randomly drawn from a uniform distribution. For $s$ hierarchically sparse vectors, we also opt for $I = 2$, and set $M = \{48, 52, 56, \cdots, 72\}$, $N = 80$, and $s = 15$. In the Kronecker-supported sparsity model, we opt for $I = 3$, and set $M = \{12, 13, \cdots, 16\}$, $N = 18$, and $s = 4$. We adopt the additive white Gaussian noise with zero mean with SNR (dB) = 20. Ratio is defined as $\bar{M}/\bar{N} = \prod_{i=1}^{I} M_i/N_i = (M/N)^I$. We consider NSE, SRR, and runtime for performance evaluation. We follow the same way to cap the number of iterations. Results in Figure 8 and Table 5 are obtained through two hundred independent trials. Overall, we observe similar trends as in Figure 7 and Table 4. Our MSR is able to provide comparable or better performance with reduced runtime, demonstrating the efficacy of exploiting the Kronecker product structure in the recovery process.

## H.2 COMPARISONS WITH TRADITIONAL COMPRESSED SENSING APPROACHES

In this section, we compare our MSR to traditional compressed sensing algorithms, including IHT, HTP, SBL, OMP, and the $\ell_1$ norm-based basis pursuit denoising (BPDN) (Foucart & Rauhut, 2013) (`basisPursuit` function in Matlab (Inc., 2024)).

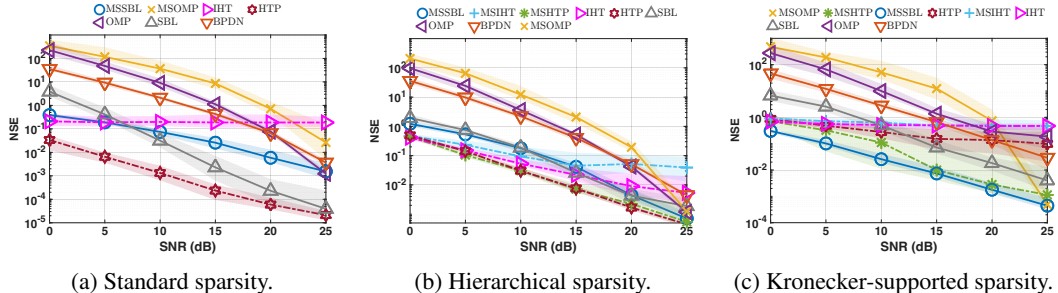

(a) Standard sparsity.   (b) Hierarchical sparsity.   (c) Kronecker-supported sparsity.

Figure 9: NSE as a function of SNR compared to traditional compressed sensing algorithms.

For all three models, we set $M_i = M$, $N_i = N$, and $s_i = s$ for $i \in [I]$. For the $s$ standard sparsity, we opt for $\boldsymbol{H} = \otimes_{i=I}^{1} \boldsymbol{H}_i$ with $I = 3$, $M = 12$, and $N = 15$. The entries of $\boldsymbol{H}_i$ and the nonzero entries of $\boldsymbol{x}$ are drawn independently from the standard normal distribution. We set $s = 8$, and the support is randomly drawn from a uniform distribution. For $\boldsymbol{s}$ hierarchically sparse vectors, we opt for $I = 2$, $M = 35$, $N = 40$, and $s = 8$ per dimension. Here, supports are generated by first selecting $s$ blocks uniformly at random, then assigning support within each block uniformly. In the $\boldsymbol{s}$ Kronecker-supported sparsity model, we opt for $I = 3$, $M = 12$, $N = 15$, and $s = 4$. In all models, the measurement noise is zero mean white Gaussian noise whose variance is determined by SNR (dB) $= 10 \log_{10} \mathbb{E}\{\|\boldsymbol{Hx}\|_2^2 / \|\boldsymbol{n}\|_2^2\}$ of $\{0, 5, 10, 15, 20, 25\}$. We follow the same condition to cap the iterative algorithms, and for BPDN, we cap the number of iterations at fifty. Compared to Section 5 and Appendix H.1, we downsize the measurement matrices mainly for computational feasibility. With the same condition as in Appendix H.1, it is hard to evaluate traditional algorithms such as BPDN and SBL. The NSE shown in Figure 9 are median and 25%/75% quartiles, while Table 6 shows the average runtime, both over fifty independent trials.

As shown in Figure 9a, HTP achieves the lowest runtime and high accuracy but relies on prior knowledge of the true sparsity level $s$, which is generally unavailable in practice. While SBL yields higher reconstruction accuracy than MSSBL in the high SNR regime, it incurs a runtime two orders of magnitude higher, limiting its scalability. MSSBL emerges as the most robust solution: it outperforms traditional methods (IHT, OMP, BPDN) and has lower runtime than SBL, offering a balance between reconstruction accuracy and computational efficiency, without requiring specific prior knowledge.

We illustrate the hierarchical sparsity recovery against traditional compressed sensing algorithms in Figure 9b. Compared to HTP, MSHTP achieves almost the same reconstruction accuracy but with one to two orders of less runtime. This also happens to MSSBL compared to SBL. MSSBL, in this case, is also a balanced option when the true sparsity level is unknown, without sacrificing efficiency significantly. OMP and IHT are only slightly worse than their counterparts, i.e., MSOMP and MSIHT, but the gain in runtime is significant by two orders of magnitude. Finally, Figure 9c contains the results for Kronecker-supported sparsity recovery against traditional benchmarks. MSSBL constantly achieves the best performance, seconded by MSHTP and MSOMP (high SNR case), with two to three orders less runtime than their counterparts.

### H.3 SPARSE VECTOR RECOVERY PERFORMANCE WITH VARYING NUMBER OF DIMENSIONS $I$

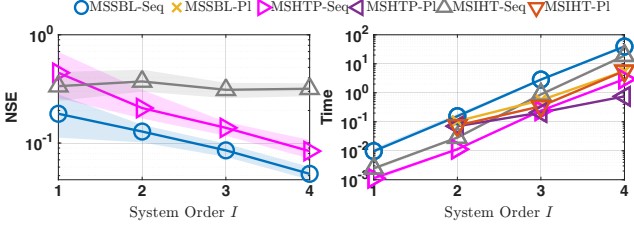

Figure 10: NSE and runtime of MSR as functions of system order $I$.

Table 6: Average runtime for comparison with traditional compressed sensing algorithms. **Bold**: the best result.

| SNR | 0 dB | 5 dB | 10 dB | 15 dB | 20 dB | 25 dB |
|---|---|---|---|---|---|---|
| | | | Recovery of $s$ sparse vectors | | | |
| MSOMP | 0.0631 | 0.0607 | 0.0618 | 0.0578 | 0.0385 | **0.0095** |
| MSSBL | 0.4547 | 0.4096 | 0.3356 | 0.1929 | 0.0969 | 0.0583 |
| HTP | **0.0210** | **0.0129** | **0.0100** | **0.0098** | **0.0091** | 0.0109 |
| IHT | 2.2580 | 2.2627 | 2.2629 | 2.2577 | 2.2473 | 2.2425 |
| OMP | 22.4379 | 20.1685 | 15.7469 | 8.9826 | 2.3411 | 0.0363 |
| SBL | 37.9208 | 21.5275 | 13.2480 | 10.9864 | 10.0749 | 9.4921 |
| BPDN | 102.8134 | 102.0363 | 101.4731 | 99.6498 | 96.1337 | 85.2775 |
| | | | Recovery of $s$ hierarchically sparse vectors | | | |
| MSOMP | 0.0246 | 0.0248 | 0.0232 | 0.0233 | 0.0173 | **0.0024** |
| MSHTP | **0.0047** | **0.0037** | **0.0031** | **0.0032** | **0.0033** | 0.0028 |
| MSIHT | 0.0066 | 0.0071 | 0.0069 | 0.0082 | 0.0088 | 0.0089 |
| MSSBL | 0.4694 | 0.3997 | 0.2533 | 0.1538 | 0.0424 | 0.0172 |
| HTP | 0.1965 | 0.1225 | 0.0777 | 0.0620 | 0.0363 | 0.0274 |
| IHT | 0.4816 | 0.4834 | 0.4842 | 0.4805 | 0.4814 | 0.4824 |
| OMP | 6.5235 | 5.7934 | 4.4745 | 2.6242 | 0.7864 | 0.0365 |
| SBL | 18.3118 | 16.1878 | 8.3721 | 3.3350 | 2.1724 | 1.8597 |
| BPDN | 20.9009 | 20.8322 | 20.3150 | 19.2756 | 17.5916 | 16.0579 |
| | | | Recovery of $s$ Kronecker-supported sparse vectors | | | |
| MSOMP | 0.0031 | 0.0027 | 0.0027 | 0.0025 | 0.0023 | 0.0014 |
| MSHTP | 0.0012 | 0.0011 | **0.0010** | 0.0010 | 0.0011 | 0.0010 |
| MSIHT | **0.0010** | **0.0009** | **0.0010** | **0.0008** | **0.0010** | **0.0009** |
| MSSBL | 0.0540 | 0.0500 | 0.0451 | 0.0322 | 0.0183 | 0.0057 |
| HTP | 0.3165 | 0.2945 | 0.2105 | 0.0945 | 0.0678 | 0.0344 |
| IHT | 1.8995 | 1.9024 | 1.8995 | 1.8991 | 1.9152 | 1.8953 |
| OMP | 23.6829 | 21.5929 | 16.4708 | 9.4103 | 2.6839 | 0.1793 |
| SBL | 46.1748 | 42.6398 | 29.0591 | 15.0069 | 11.0725 | 9.5650 |
| BPDN | 108.0371 | 108.1285 | 108.0272 | 106.9569 | 102.1098 | 91.1148 |

Here we present results where we fix the size of each factor matrix $\boldsymbol{H}_i$ but vary $I$ from $1$ to $4$, to demonstrate the ability of our MSR to handle arbitrary system order $I$.

We consider $N_i = N$ and $M_i = M$ for $i \in [I]$, opt for $N = 50$, and determine $M$ and $s$ through $M = \lceil (0.6\bar{N})^{1/I} \rceil$ and $s = \lceil 0.4N \rceil$ as in Section 5. We fix SNR at 20dB and results in Figure 10 shows how the NSE and runtime (median with $25\%/75\%$ quartiles) of MSSBL, MSHTP, and MSIHT scale with different $I$ for Kronecker-supported sparsity model.

In Figure 10, we observe similar trends as in Figure 3. Parallel and sequential implementations have identical recovery performance but different runtimes. Thus, for NSE, we only show the results for parallel implementation. The only exception in Figure 10 compared to Figure 3 is that for $I = 2$, parallel implementation for MSHTP and MSIHT requires more computation time than the sequential implementation. This is because when the problem is computationally light, parallel overhead dominates the time consumption, including data transfer and communication, rather than computation itself.

### H.4 APPLICATION: CHANNEL ESTIMATION FOR INTELLIGENT REFLECTING SURFACE-AIDED WIRELESS SYSTEM

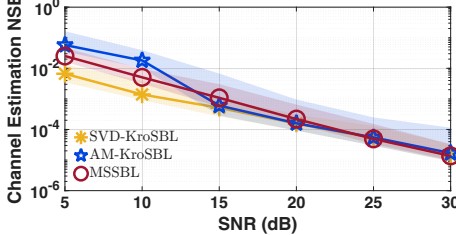

Figure 11: NSE of different schemes.

| SNR (dB) | AM-KroSBL | SVD-KroSBL | MSSBL |
|---|---|---|---|
| 5 | 138.9725 | 3.1272 | 0.0212 |
| 10 | 121.7270 | 2.8479 | 0.0147 |
| 15 | 77.7143 | 2.7911 | 0.0099 |
| 20 | 43.9860 | 2.7752 | 0.0066 |
| 25 | 26.3338 | 2.7479 | 0.0052 |
| 30 | 18.7202 | 2.7450 | 0.0052 |

Table 7: Average runtime in seconds.

An intelligent reflecting surface (IRS) is a reconfigurable meta-surface consisting of a large number of adjustable reflecting elements. By changing the reflection coefficients of elements, IRS can reflect the signal to a certain area to improve the coverage of the wireless communication systems operating at millimeter-wave/terahertz frequency bands. But properly configuring the IRS requires the channel state information. It is hence important to develop efficient and accurate channel estimation algorithms for the IRS-aided system. In this section, we consider an uplink narrowband IRS-aided MIMO system, consisting of a $T$ half-wavelength spacing-antenna transmitter mobile station (MS), an $R$ half-wavelength spacing-antenna receiver base station (BS), and an $L$ half-wavelength spacing-element uniform linear array IRS. If the channel matrices of the MS-IRS, and the IRS-BS channel are denoted as $\boldsymbol{\Phi}_{\mathrm{MS}} \in \mathbb{C}^{L \times T}$ and $\boldsymbol{\Phi}_{\mathrm{BS}} \in \mathbb{C}^{R \times L}$, respectively, according to the geometric channel model (You et al., 2022; Wang et al., 2020; Alkhateeb et al., 2014; He & Joseph, 2023), $\boldsymbol{\Phi}_{\mathrm{MS}} \in \mathbb{C}^{L \times T}$ and $\boldsymbol{\Phi}_{\mathrm{BS}} \in \mathbb{C}^{R \times L}$ can be formulated as

$$\boldsymbol{\Phi}_{\mathrm{MS}} = \sum_{p=1}^{P_{\mathrm{MS}}} \sqrt{\frac{LT}{P_{\mathrm{MS}}}} \beta_{\mathrm{MS},p} \boldsymbol{a}_L(\phi_{\mathrm{MS},p}) \boldsymbol{a}_T(\alpha_{\mathrm{MS}})^{\mathsf{H}}, \tag{8}$$

$$\boldsymbol{\Phi}_{\mathrm{BS}} = \sum_{p=1}^{P_{\mathrm{BS}}} \sqrt{\frac{RL}{P_{\mathrm{BS}}}} \beta_{\mathrm{BS},p} \boldsymbol{a}_R(\alpha_{\mathrm{BS},p}) \boldsymbol{a}_L(\phi_{\mathrm{BS}})^{\mathsf{H}}, \tag{9}$$

where $P_{\mathrm{MS}}$ and $P_{\mathrm{BS}}$ are the number of paths between MS and IRS, and IRS and BS, respectively. The angles $\phi_{\mathrm{MS},p}$, $\alpha_{\mathrm{MS}}$, $\alpha_{\mathrm{BS},p}$, and $\phi_{\mathrm{BS}}$ represent the $p$th AoA of the IRS, and the $p$th AoD of the MS, the $p$th AoA of the BS, and the AoD of the IRS, respectively, while $\beta_{\mathrm{MS},p}$ and $\beta_{\mathrm{BS},p}$ are the complex path gains. Steering vector $\boldsymbol{a}_Q(\psi) \in \mathbb{C}^Q$ for any integer $Q$ and angle $\psi$ is defined as $\boldsymbol{a}_Q(\psi) = 1/\sqrt{Q}[1, e^{j\pi \cos \psi}, \cdots, e^{j\pi(Q-1)\cos \psi}]^{\top}$. Then, the cascaded MS-IRS-BS channel is given by $\boldsymbol{\Phi}_{\mathrm{BS}} \operatorname{diag}(\boldsymbol{\theta}) \boldsymbol{\Phi}_{\mathrm{MS}}$ for a given IRS configuration $\boldsymbol{\theta} \in \mathbb{C}^L$ whose $i$th entry of $\boldsymbol{\theta}$ models the reflection of the $i$th IRS element. Channel estimation problem targets to estimate the cascaded channel $\boldsymbol{\Phi}_{\mathrm{BS}} \operatorname{diag}(\boldsymbol{\theta}) \boldsymbol{\Phi}_{\mathrm{MS}}$ given any $\boldsymbol{\theta}$, which is sufficient for subsequent tasks such as beamforming (Wang et al., 2020).

Wireless channels operating on millimeter-wave/terahertz bands are intrinsically sparse due to severe path loss. To reveal this sparsity, we adopt three sparsifying bases $\boldsymbol{A}_R$, $\boldsymbol{A}_L$, and $\boldsymbol{A}_T$, corresponding to the angular domain of the array at BS, IRS, and MS, respectively. Such bases contain steering vectors evaluated over $N$ grid angles $\{\psi_n\}_{n=1}^N$ such that $\cos(\psi_n) = 2n/N - 1$ (Mao et al., 2022), defined as $\boldsymbol{A}_Q = [\boldsymbol{a}_Q(\psi_1), \boldsymbol{a}_Q(\psi_2), \ldots, \boldsymbol{a}_Q(\psi_N)] \in \mathbb{C}^{Q \times N}$ for any integer $Q > 0$. Then, Equation 8 and 9 reduce to

$$\boldsymbol{\Phi}_{\mathrm{BS}} = \boldsymbol{A}_R \boldsymbol{x}_{\mathrm{R}} \boldsymbol{x}_{\mathrm{L,d}}^{\mathsf{H}} \boldsymbol{A}_L^{\mathsf{H}} \quad \text{and} \quad \boldsymbol{\Phi}_{\mathrm{MS}} = \boldsymbol{A}_L \boldsymbol{x}_{\mathrm{L,a}} \boldsymbol{x}_{\mathrm{T}}^{\mathsf{H}} \boldsymbol{A}_T^{\mathsf{H}}, \tag{10}$$

where vectors $\boldsymbol{x}_{\mathrm{R}}, \boldsymbol{x}_{\mathrm{L,d}}, \boldsymbol{x}_{\mathrm{L,a}}, \boldsymbol{x}_{\mathrm{T}} \in \mathbb{C}^N$ are the unknown channel representations over the known sparsifying bases. They are sparse due to the intrinsic sparsity of the channel.

Channel estimation is performed by processing the received pilot signals, given the knowledge of the sent pilot signals and the training IRS configurations. Suppose we allocate $K$ time slots for channel estimation, over which the channel is considered to be constant. We vary IRS configurations for $K_{\mathrm{I}}$ times, and for each different configuration, we transmit the same set of pilot signal $\boldsymbol{G} \in \mathbb{C}^{T \times K_{\mathrm{P}}}$ over $K_{\mathrm{P}}$ time slots such that $K = K_{\mathrm{I}} K_{\mathrm{P}}$. The received signal $\boldsymbol{Y}_k \in \mathbb{C}^{R \times K_{\mathrm{P}}}$ corresponding to the $k$th training configuration $\boldsymbol{\theta}_k$ is

$$\boldsymbol{Y}_k = \boldsymbol{\Phi}_{\mathrm{BS}} \operatorname{diag}(\boldsymbol{\theta}_k) \boldsymbol{\Phi}_{\mathrm{MS}} \boldsymbol{G} + \boldsymbol{N}_k, \tag{11}$$

where $\boldsymbol{N}_k \in \mathbb{C}^{R \times K_{\mathrm{P}}}$ is the noise. Substituting Equation 10 into Equation 11, and vectorizing the received signal $\{\boldsymbol{Y}_k\}_{k=1}^{K_{\mathrm{I}}}$ followed by some algebraic operations (He & Joseph, 2023), we have

$$\boldsymbol{y} = (\boldsymbol{H}_{\mathrm{L}} \otimes \boldsymbol{H}_{\mathrm{T}} \otimes \boldsymbol{H}_{\mathrm{R}}) \boldsymbol{x} + \boldsymbol{n} = \boldsymbol{H} \boldsymbol{x} + \boldsymbol{n} \in \mathbb{C}^{RK}, \tag{12}$$

where $\boldsymbol{H}_{\mathrm{L}} \in \mathbb{C}^{K_{\mathrm{I}} \times N}$ is formed by the first $N$ columns of $\boldsymbol{\Theta}^{\top}(\boldsymbol{A}_L^{\top} \odot \boldsymbol{A}_L^{\mathsf{H}})^{\top}$ whose $N^2$ columns are just $N$ repetitions the columns of $\boldsymbol{H}_{\mathrm{L}}$, $\boldsymbol{H}_{\mathrm{T}} = \boldsymbol{X}^{\top} \boldsymbol{A}_T^*$, and $\boldsymbol{H}_{\mathrm{R}} = \boldsymbol{A}_R$, with $\odot$ being the Khatri-Rao product. We collect $K_{\mathrm{I}}$ IRS configurations $\{\boldsymbol{\theta}_k\}_{k=1}^{K_{\mathrm{I}}}$ in matrix $\boldsymbol{\Theta} \in \mathbb{C}^{L \times K_{\mathrm{I}}}$ and define $\boldsymbol{x} = \boldsymbol{x}_{\mathrm{L}} \otimes \boldsymbol{x}_{\mathrm{T}}^* \otimes \boldsymbol{x}_{\mathrm{R}} \in \mathbb{C}^{N^3}$ with $\boldsymbol{x}_{\mathrm{L}} \in \mathbb{C}^N$ being the scaled version of the first $N$ entries of $\boldsymbol{x}_{\mathrm{L,a}} \otimes \boldsymbol{x}_{\mathrm{L,d}}^*$, corresponding to the removal of redundant columns in $\boldsymbol{\Theta}^{\top}(\boldsymbol{A}_L^{\top} \odot \boldsymbol{A}_L^{\mathsf{H}})^{\top}$. Denoting

$\boldsymbol{H}_\text{L}$ as $\boldsymbol{H}_3$ with $K_\text{I}$ as $M_3$, $\boldsymbol{H}_\text{T}$ as $\boldsymbol{H}_2$ with $K_\text{P}$ as $M_2$, $\boldsymbol{H}_\text{R}$ as $\boldsymbol{H}_1$ with $R$ as $M_1$, and $N = N_3 = N_2 = N_1$, the channel estimation problem is transformed into a KCS problem following the form of Equation 1 with unknown $\boldsymbol{s} = (s_3, s_2, s_1) = (P_\text{MS}, 1, P_\text{BS})$ *Kronecker-supported sparse* vector $\boldsymbol{x}$. With the estimated $\hat{\boldsymbol{x}}$, the estimated channel for a given IRS configuration $\boldsymbol{\theta}$, i.e., $\hat{\boldsymbol{\Phi}}_\text{BS} \operatorname{diag}(\boldsymbol{\theta}) \hat{\boldsymbol{\Phi}}_\text{MS}$, is obtained by reshaping $(\hat{\boldsymbol{\Phi}}_\text{MS}^\top \odot \hat{\boldsymbol{\Phi}}_\text{BS}) \boldsymbol{\theta}$ with known size $R \times T$, where $\hat{\boldsymbol{\Phi}}_\text{MS}^\top \odot \hat{\boldsymbol{\Phi}}_\text{BS}$ with known size $RT \times L$ is reconstructed as $\operatorname{vec}(\hat{\boldsymbol{\Phi}}_\text{MS}^\top \odot \hat{\boldsymbol{\Phi}}_\text{BS}) = (\boldsymbol{\Phi}_\text{A} \otimes \boldsymbol{A}_T^* \otimes \boldsymbol{A}_R) \hat{\boldsymbol{x}}$ with $\boldsymbol{\Phi}_\text{A}$ being the first $N$ columns of $(\boldsymbol{A}_L^\top \odot \boldsymbol{A}_L^\mathsf{H})^\top$.

We set $T = 6$, $R = 16$, and $L = 256$. To set up the sparsifying bases, we opt for $N = 18$. Pilot signals contained in $\boldsymbol{G}$ are randomly generated quadrature phase shift keying symbols using a uniform distribution while the IRS training configuration is randomly drawn from uniform distribution $\{\pm 1/\sqrt{L}\}$ (Lin et al., 2021). Regarding the pilot signals and training IRS configurations, we consider $K_\text{I} = K_\text{P} = 10$, making $\boldsymbol{H}_3 \in \mathbb{C}^{10 \times 18}$, $\boldsymbol{H}_2 \in \mathbb{C}^{10 \times 18}$, and $\boldsymbol{H}_1 \in \mathbb{C}^{16 \times 18}$. To model the scatters, we set $P_\text{MS} = P_\text{BS} = 3$ and all angles $\phi_{\text{MS},p}$, $\alpha_\text{MS}$, $\alpha_{\text{BS},p}$, and $\phi_\text{BS}$ are drawn uniformly and independently from the grid points, while path gains $\beta_{\text{MS},p}$ and $\beta_{\text{BS},p}$ are drawn independently from complex standard normal distribution (He & Joseph, 2023). We compare MSSBL with SVD-/AM-KroSBL in He & Joseph (2023) with the same way to cap iterative algorithms. Metrics include channel estimation NSE given by $\frac{1}{K_\text{I}} \sum_{k=1}^{K_\text{I}} \frac{\|\hat{\boldsymbol{\Phi}}_\text{BS} \operatorname{diag}(\boldsymbol{\theta}_k) \hat{\boldsymbol{\Phi}}_\text{MS} - \boldsymbol{\Phi}_\text{BS} \operatorname{diag}(\boldsymbol{\theta}_k) \boldsymbol{\Phi}_\text{MS}\|_\text{F}^2}{\|\boldsymbol{\Phi}_\text{BS} \operatorname{diag}(\boldsymbol{\theta}_k) \boldsymbol{\Phi}_\text{MS}\|_\text{F}^2}$ and runtime.

Figure 11 shows 25%/50%/75% quartiles of NSE, while Table 7 shows the average runtime, both over fifty independent trials. We observe that all three algorithms provide comparable channel estimation performance, while MSSBL has two orders less runtime than SVD-KroSBL and four orders less runtime than AM-KroSBL, making it more efficient in this application scenario.

## H.5 APPLICATION: FOREMAN VIDEO SEQUENCE RECOVERY

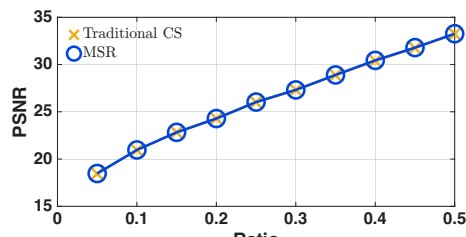

| Ratio $\zeta$ | Traditional CS | MSR |
|---|---|---|
| 0.05 | 77.5294 | 10.5590 |
| 0.10 | 91.2761 | 9.0006 |
| 0.15 | 70.1564 | 9.7536 |
| 0.20 | 75.8293 | 9.9201 |
| 0.25 | 67.9980 | 11.6894 |
| 0.30 | 99.2622 | 11.1384 |
| 0.35 | 79.0998 | 11.1111 |
| 0.40 | 109.8495 | 11.1645 |
| 0.45 | 92.8983 | 7.7705 |
| 0.50 | 66.3104 | 6.7719 |

Figure 12: PSNR of different schemes.    Table 8: Runtime in seconds.

This section presents the recovery results for *Foreman video sequence* to demonstrate MSR's ability to deal with a real-world dataset and its superiority over the traditional compressed sensing algorithm. In the experiment, we follow the settings in Duarte & Baraniuk (2011a;b) and to make this paper self-contained, we provide a brief overview of these settings.

Tested frames are generated by cropping around the center to form a frame size of $128 \times 128$ pixels and there are in total eight frames used in the experiment. To spatially sparsify the image content within a single frame, we vectorize each frame and adopt a $2D$ inverse discrete wavelet transform basis $\boldsymbol{W}_1 \in \mathbb{R}^{16384 \times 16384}$ applied to the sparse coefficients of each video frame. For the sparsity in the temporal dimension, we turn to an $1D$ inverse discrete wavelet transform basis $\boldsymbol{W}_2 \in \mathbb{R}^{8 \times 8}$. It exploits the correlation between frames to sparsify the signal over time. Suppose the video sequence is denoted by $\boldsymbol{\theta} \in \mathbb{R}^{131072}$ where $131072 = 8$ (frames) $\times 128$ (row pixels) $\times 128$ (columns pixels), then its relation to sparsifying bases and the sparse coefficient vector $\boldsymbol{x} \in \mathbb{R}^{131072}$ is $\boldsymbol{\theta} = (\boldsymbol{W}_2 \otimes \boldsymbol{W}_1) \boldsymbol{x}$. To compress the video sequence, we use a measurement matrix as $\boldsymbol{I} \otimes \boldsymbol{S}$, where $\boldsymbol{I}$ means that there is no temporal compression while $\boldsymbol{S} \in \mathbb{R}^{M \times 16384}$ from a subsampled permuted Hadamard transform denotes the spatial compression. Here, $M$ is the number of measurements taken in one frame. This leads to the following measurement/sparsifying model

$$\boldsymbol{y} = (\boldsymbol{W}_2 \otimes \boldsymbol{S} \boldsymbol{W}_1) \boldsymbol{x}. \tag{13}$$

The goal is to obtain sparse coefficient $\boldsymbol{x}$ using the compressed measurement $\boldsymbol{y}$, and finally we reconstruct the video sequence as $\boldsymbol{\theta} = (\boldsymbol{W}_2 \otimes \boldsymbol{W}_1) \boldsymbol{x}$.

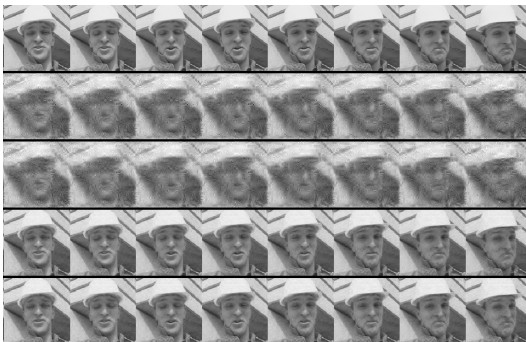

Figure 13: Comparison of reconstructed Foreman video frames. Row 1: ground truth. Row 2: traditional CS with $\zeta = 0.1$. Row 3: MSR with $\zeta = 0.1$. Row 4: traditional CS with $\zeta = 0.5$. Row 5: MSR with $\zeta = 0.5$.

Denoting $\boldsymbol{W}_2$ as $\boldsymbol{H}_2$, $\boldsymbol{SW}_1$ as $\boldsymbol{H}_1$, and $I = 2$, we note that Equation 13 is mathematically equivalent to Equation 1, hence can be solved using our MSR. The benchmark in this experiment is $\ell_1$-based basis pursuit (Duarte & Baraniuk, 2011a), where Equation 13 is treated as a traditional compressed sensing problem and the Kronecker structure of $\boldsymbol{W}_2 \otimes \boldsymbol{SW}_1$ is ignored. For a fair comparison, we adopt the same $\ell_1$ solver (van den Berg & Friedlander, 2008; 2019) in MSR as in Duarte & Baraniuk (2011a) with the same stopping criterion. We determine the number of measurements $M$ as $M = \lfloor 16384\zeta \rceil$ with measurement ratio $\zeta \in \{0.05, 0.10, \cdots, 0.45, 0.50\}$. We use the peak signal-to-noise ratio (PSNR) and runtime as evaluation metrics. Results are shown in Figure 12 and Table 8, where traditional CS refers to traditional compressed sensing algorithm. We observe that MSR achieves the same PSNR as that of traditional CS with roughly one order of magnitude less runtime, effectively demonstrating the efficacy of our MSR on a real-world dataset. We also compare all used video frames in Figure 13.

