# OpenReview forum: "Hierarchical Multi-Stage Recovery Framework for Kronecker Compressed Sensing"
_ICLR.cc/2026/Conference — ICLR 2026 Poster_

### Official Review · Reviewer_MA64 · 2025-10-18

**Soundness:** 3
**Presentation:** 3
**Contribution:** 4
**Rating:** 8
**Confidence:** 3

**Summary:**

In the Kronecker compressed sensing (KCS) framework, the measurement matrix is composed by the Kronecker product of multiple factor matrices as $ H = H_I \otimes \cdots \otimes H_1$. While this allows for efficient measurement of signals with multidimensional structures, the dimensionality increases rapidly (e.g., $O(N^I))$, which makes existing methods computationally expensive. They also fail to fully utilize the individual factor matrix structures and hierarchical sparsity.
This paper addresses these difficulties by introducing the {\em hierarchical view}.
The contribution of this paper is composed of the following three parts.
First, it is theoretically demonstrated that the Kronecker product structure is a "hierarchical block partition" and that each factor matrix $H_i$ measures the sparsity of a different "hierarchical level." Taking advantage of the Kronecker structure, an algorithm, Multi-Stage Recovery (MSR), that performs reconstruction sequentially (or in parallel) for each layer is proposed. This reduces the computational cost for signal reconstruction to $O((MN)^I)$ to $O(MN^I)$.
Second, a theoretical guarantee is developed by introducing a new type RIP-based condition.
Finally, for all three sparse models, MSR methods (MSOMP, MSHTP, MSSBL, etc.) show comparable or superior accuracy to existing methods (KroOMP, HiHTP, KroSBL, etc.), while reducing computational time by 1–3 orders of magnitude.

**Strengths:**

By introducing a new perspective into the Kronecker compressed sensing framework, we propose a signal restoration algorithm that can significantly reduce the required computational effort compared to conventional methods. At the same time, we also conduct novel performance evaluations, theoretically guaranteeing performance, and demonstrate the practical usefulness of the proposed algorithm through numerical experiments.

**Weaknesses:**

Kronecker compressed sensing is only possible when the observed signal has multidimensional structural sparsity or separable statistical structure, which limits the practical applications in which the proposed method is useful.

**Questions:**

This paper refers to radar imaging and wireless communication as examples of applications where the proposed method is useful, because the sparsity of each signal can be separated into dimensions. Can the usefulness of this method be verified for real data in these fields?

---

> ### Author Response · Authors · 2025-11-19
> **Response to reviewer's concern on missing real-world data-based verification**
>
> We sincerely thank the reviewer for the positive feedback.
>
> **W1**: Kronecker compressed sensing is only possible when the observed signal has multidimensional structural sparsity or separable statistical structure, which limits the practical applications in which the proposed method is useful.
>
> **A1**: The reviewer has precisely identified the motivation of KCS. We agree that our method, under the framework of KCS, is not a universal solution for all compressed sensing problems, but a specialized framework designed for multidimensional signals. This is a deliberate trade-off. For specific but still widely existing applications, such as wireless channel estimation, a general method that relies on traditional compressed sensing methods can be practically difficult due to the signal’s multidimensional nature [Duarte & Baraniuk, 2011a]. Thus, Kronecker compressed sensing and our approach trade the generality of traditional compressed sensing for better efficiency over specialized problems. For completeness, we list the following applications where KCS with structured sparsity appears.
>
> - Standard sparsity and KCS:
>   - Compressive imaging [Rivenson & Stern (2009)].
>   - Hyperspectral imaging [Duarte & Baraniuk (2011a); Li & Bernal (2017)]
>   - Compressible image and video representation and recovery [Friedland et al. (2014)].
> -  Hierarchical sparsity and KCS:
>    - Channel estimation problem for a massive multiple-input multiple-output system [Roth et al. (2020); Wunder et al. (2019)].
> - Kronecker-supported sparsity and KCS:
>   - Image processing [Caiafa & Cichocki (2012); Caiafa & Cichocki (2013); Zhao et al. (2019)]
>   - Wireless communication [He & Joseph (2025a); Chang & Su (2021); Xu et al. (2022)].
>
>
> **Q1**: This paper refers to radar imaging and wireless communication as examples of applications where the proposed method is useful, because the sparsity of each signal can be separated into dimensions. Can the usefulness of this method be verified for real data in these fields?
>
> **A2**: We thank the reviewer for this constructive feedback regarding the practical applicability. We added three sets of numerical results with model-based validation, relying on well-established system models that are highly representative of the intended applications, and demonstrating improvements against baselines. We added
> - hierarchical sparsity evaluation in Section 5 for channel estimation in a wideband massive multiple-input multiple-output system using the channel model in [Haghighatshoar & Caire,
> 2017; Chen & Yang, 2016; Wunder et al., 2019] with accuracy improvement and runtime reduction, and
> - Kronecker-supported sparsity evaluation in Appendix H.4 for channel estimation in a narrowband intelligent reflecting surface-aided system using the channel model in [He & Joseph, 2023; You et al., 2022] with similar accuracy but significant runtime reduction,
> - standard sparsity evaluation in Appendix H.5 for real-world Foreman video sequence reconstruction [Duarte & Baraniuk (2011a)], showing similar peak signal-to-noise ratio but significant runtime reduction compared to the benchmark.

---

> > ### Comment · Reviewer_MA64 · 2025-11-27
> >
> > Thank you for your answer. I keep my evaluation as it is.

---

> > > ### Author Response · Authors · 2025-11-28
> > > **Thank you for your support**
> > >
> > > We thank the reviewer for the continued effort and positive confirmation. We deeply appreciate your constructive feedback, which has helped us strengthen our work.

---

### Official Review · Reviewer_Wv5h · 2025-10-24

**Soundness:** 2
**Presentation:** 2
**Contribution:** 3
**Rating:** 6
**Confidence:** 4

**Summary:**

The authors consider the problem of recovering sparse vectors $x$ from Kronecker measurements, i.e., measurement matrix of the form $H=H_1\otimes \dots \otimes H_N$. The overarching idea is to perform the recovery in stages: By unfolding of the measurement $y=Hx$ in different manners, the recovery problem can be viewed as a number of  MMV problems of the form $y= H_j U_j$, that can be solved in an iterative fashion. They coin this class of algorithms MSR, multi-stage recovery. Importantly, their method is improved if the $x$ also is assumed to have a Kronecker sparse structure, since the MMV problems become coupled.

They derive a convergence guarantee for their method that is valid under the assumption that each $H_i$ has the RIP. Their bound can be applied to different types of sparsity in $x$, and in particular improves when $x$ Kronecker-sparse. It however potentially suffers from the potential of error propagation between the folded stages.

**Strengths:**

The main strength of the paper is that it proposes a method which performs better for Kronecker-based sparsity than for general hierarchical sparsity, which with the exception of He and Joseph, 2025, seems to be novel.

The recovery guarantee that the authors provide is interesting in that it only requires each constituent matrix to have a constant RIP ($\delta_{s_i}(H_i)<1/\sqrt{3}$) for convergence in the noiseless case. This is actually a lot better than previous guarantees -- using the bound in Theorem 1 to get an RIC (which is what is needed for e.g. the HiHTP to converge), one essentially need the sums of the $\delta_{s_i}(H_i)$ to be less than $1/\sqrt{3}$, which intuitively means that each constant must be smaller than $1/\sqrt{I\cdot 3}$, which will be very low for a high number of Kronecker levels. It is also interesting to see that their technique, through Corollary 1, improves on the previously quite weak bounds for standard sparsity.

**Weaknesses:**

The paper is sometimes hard to follow, mainly due to the complexity of the notation related to tensor-algebra and the different sparsity notions. I think that refinement of the illustrations would go a long way to alleviate the problems. The color coding in Figure 4 is in my opiniion rather confusing than helpful.

The RIC analysis offers some new interesting perspectives, but it seems very close to the one performed in [Roth et. al; 2020] and [Roth et al; 2018]. The relations between the two should be better explained, see questions below.

The superiority of the proposed methods compared to the SOTA, both for hierarchically sparse vectors and Kronecker-sparse vectors, are mainly argued for via comparison of run-times. This metric depends a lot of the specifics of the implementation. Could something more objective be measured and presented, such as the number of iterations? I particularly stumble upon this since the discussion regarding time-complexity on page 5 seems to contain mistakes for HiHTP, see below.

**Questions:**

** Relation of RIP analysis to [Roth et al; 2020]** Although the formulation of the results are different, it seems like the formulation and proof of Theorem 1 in the manuscript at hand is essentially the same as the formulation and proof of Theorem 4. Can the authors comment on this, and in particular pinpoint exactly how the analyses are different?

** Complexity discussion and HiHTP ** In the complexity discussion, the authors claim that the space- and time-complexity for HiHTP ($I=2$) both are $M^2N^2$. However [Roth et al; 2020] claim in their paper that the per-iteration time- complexity of their algorithm (for generic measurement matrices) is (in this notation) $MN^2$. Can the authors clarify?

---

> ### Author Response · Authors · 2025-11-19
> **Response to reviewer's concerns on notation clarification, RIC comparison to prior work, and complexity analysis**
>
> We sincerely thank the reviewer for the meticulous review.
>
> **W1**: The notation related to tensor-algebra and the different sparsity notions is complex. The color coding in Figure 4 is in my opinion rather confusing than helpful.
>
> **A1**: To clarify tensor-algebra (mainly on tensor unfolding), different sparsity notions, and the confusing figure for the proof of Lemma 1, we made the following changes.
>
> - We provide a new Figure 5 to replace the original Figure 4, containing an example with specific values for $I$, $M_i$'s, and $N_i$'s. In Figure 5, we show an example of tensor unfolding (Figure 5(c)), and explain carefully how Lemma 1 holds.
> - We contain a new Figure 6, where we also illustrate different sparsity notions with specific values in Appendix B. In Figure 6, we not only explain how sparsity is defined, but also explain some other notions, such as encapsulations and set $[[\mathbf{x}_{\mathrm{n_j}}]]$, which are needed for the definition of sparsity pattern.
>
> **W2&Q1**: Although the formulation of the results is different, it seems like the formulation/proof of Theorem 1 here is essentially the same as that of Theorem 4. Can the authors specify exactly how the analyses differ?
>
> **A2**: The high-level proof strategy of Theorem 1 in our paper and Theorem 4 in [Roth et al. (2020)] is similar in that both aim to sequentially unwrap the effect of the Kronecker product. The key difference lies in how this is done: we employ tensor representations/operations such as tensor unfolding, enabling a straightforward, flip-operator-free proof. This formulation clearly demonstrates how the sparse signal $\mathbf{x}$ (or its tensor form and its unfolding) is measured by factor matrix $\mathbf{H}_j$  through a linear transformation $\mathbf{X}\_{(j)}\left(\mathbf{I}\_{\prod\_{i=I}\^{j+1}N_i}\otimes\left( \otimes\_{i=j-1}^{1} \mathbf{H}_i \right)  \right)^\top$ whose row sparsity is dictated by the sparsity of our hierarchical block partition. The aspect of this multi-stage measurement framework is missing in [Roth et al. (2020)], which limits its focus solely on hierarchical sparsity. In contrast, our multi-stage measurement framework provides a general perspective that defines generalized sparsity, where standard, hierarchical, and Kronecker-supported sparsity are special cases. Our unified view also explains why standard RIP cannot be improved beyond hierarchical sparsity, clarifies the maximum achievable sparsity level, and shows why the corresponding bounds are fundamentally tight. It further provides insight into why proofs for Kronecker-supported sparsity can be strengthened, drawing analogies to standard RIP and MMV analyses.
>
> We revised Appendix D for better clarification.
>
> **W3&Q2**: Objective measurement such as the number of iterations and clarification on the per-iteration complexity of HiHTP ($I=2$).
>
> **A3**: Regarding the number of iterations, for fairness, we use identical exit conditions and a common iteration cap (100) for all HTP-based methods. These details are now provided in Appendix H.1. Yet, we use runtime as a key performance metric because iterative algorithms inherently trade off performance and number of iterations: fewer iterations reduce runtime but also degrade reconstruction accuracy. Our goal is to demonstrate that our methods can achieve equal or better accuracy (e.g., NSE) more quickly, which is the most meaningful practical measure.
>
> Besides, the number of iterations alone is an incomplete metric as per-iteration complexities vary across algorithms, e.g., HiHTP can have a larger per-iteration complexity than ours. Thus, two methods with the same iteration count may still have very different computational costs. Overall, based on theoretical complexity analysis in Section 3 and consistently shorter runtimes of our algorithms for a similar or better performance with respect to benchmarks, we believe our claim is well supported.
>
> About the per-iteration complexity of HiHTP, a potential notational confusion arises in [Roth et al. (2020)] because the symbol $m$ is used in two different ways: in their Sections I and II (where the complexity analysis appears) and III.(a), it denotes the total number of measurements of the full matrix, but later in III.(b), it is also used for the dimension of *a factor matrix*.
>
> To clarify, in the complexity analysis, $m$ refers to the *total number of rows/measurements* of the overall, generic measurement matrix $\mathbf{A} \in \mathbb{K}^{m \times d}$, where $d=nN$. They report a complexity of $\mathcal{O}(mNn)$. In our paper, $\bar{M}$ refers to the total number of rows/measurements, while $M$ refers to the number of rows/measurements of a single factor matrix. For the $I=2$ case discussed, the overall measurement matrix has dimensions $\bar{M} \times \bar{N}$, where $\bar{M} = M^I = M^2$. Substituting $m=M^2$, their complexity $\mathcal{O}(mNn)$ yields the same per-iteration complexity we claim.

---

> > ### Comment · Reviewer_Wv5h · 2025-11-21
> > **Thank you for the clarifications**
> >
> > First, thank you for clearing out the confusion about the $m$ and the $M$ regarding the per-iteration complexity of HiHTP. I now understand the discrepancy.
> > Also thank you for providing more details about the experiments in the appendix. It is now easier to judge whether the time elapsed is a reasonable metric for the difference in performance of the methods. Since the differences are so bit, it seems like it is not the experiment set up that is the only cause for the discrepancy.
> >
> > I also appreciate the explaination regarding the similarities and differences to Roth et al. It seems like the proof ideas indeed are similar, but it is also clear that the new perspectives of the authors are bringing novel insights.
> >
> > My opinion is post-rebuttal that there is worth in publishing the work, and I have therefore updated my score.

---

> > > ### Author Response · Authors · 2025-11-24
> > > **Thank you for the positive feedback and score update**
> > >
> > > We sincerely thank you for the continued engagement. We are glad that our response successfully clarified the confusion, and your constructive feedback has significantly improved our manuscript.

---

### Official Review · Reviewer_64D7 · 2025-10-27

**Soundness:** 3
**Presentation:** 2
**Contribution:** 2
**Rating:** 4
**Confidence:** 2

**Summary:**

This paper proposes a multi-stage sparse recovery framework for Kronecker compressed sensing, tailored to three sparsity models, with theoretical guarantees and simulations showing comparable performance and faster runtime than state-of-the-art methods.

**Strengths:**

1）Developed a versatile multi-stage sparse recovery algorithmic framework tailored to three different sparsity models (standard, hierarchical, and Kronecker-supported) for the Kronecker compressed sensing problem.

2）Provided theoretical recovery guarantees by analyzing the restricted isometry property of Kronecker product matrices under different sparsity models, and demonstrated comparable recovery performance with significantly reduced run time through simulations.

**Weaknesses:**

1）Compared to traditional compressed sensing, Kronecker Compressed Sensing (KCS) has not garnered sufficient attention, primarily due to its excessive complexity—including challenges in measurement matrix construction, signal reconstruction and hardware implementation—while lacking significant performance advantages that could justify its complexity. One notable strength of the method proposed in this paper is its reduced reconstruction complexity. However, we are particularly interested in how it compares to traditional methods in terms of runtime and reconstruction accuracy, yet the paper appears to lack relevant experimental evaluations.

2）The construction of measurement matrices remains a major challenge in KCS. While the paper claims to improve Restricted Isometry Property (RIP) conditions, there seems no  concrete method for constructing matrices that satisfy these enhanced conditions.

**Questions:**

Besides the two major concerns mentioned above, I have the following questions:

1) The paper introduces three sparsity models, but it does not specify which types of real-world data these models are tailored for. Despite asserting broad applicability, the paper lacks simulations using authentic datasets.

2) As previously mentioned, a critical comparison with traditional compressed sensing methods in terms of reconstruction accuracy is essential, given the high computational complexity of KCS.

3) In the experiments, the analysis is limited to cases with a number of stages no greater than three, namely $I\leq 3$. Why was this restriction imposed? Ideally, the performance variations across different stage numbers should have been explored.

---

> ### Author Response · Authors · 2025-11-19
> **Response to reviewer's concerns on comparisons with traditional CS, guidance on constructing matrices with improved RIP conditions, real-world applications, and experiments with $I> 3$**
>
> We sincerely thank the reviewer for the constructive comments.
>
> **W1&Q2**: Comparison between methods in this paper and traditional methods in terms of runtime/accuracy is essential.
>
> **A1**: We provide additional results, particularly comparing our approach with traditional methods (OMP, IHT, HTP, SBL, and $\ell_1$ based basis pursuit denoising) regarding runtime/accuracy in Appendix H.2. For hierarchical/Kronecker-supported sparsity, a similar/better performance with lower runtime is shown, while for standard sparsity, our method trade off between accuracy, speed, and requirement of sparsity level.
>
> **W2**: There seems to be no concrete method for constructing matrices that satisfy these enhanced conditions.
>
> **A2**: We agree that construction of RIP-satisfying matrices is challenging, not just for KCS, but for compressed sensing (CS) in general. Our result (Theorem 1) shows that KCS does *not* require a new or specialized construction: it is sufficient that each individual factor matrix $\mathbf{H}_i$ satisfies the standard RIP. This simplifies the construction process and aligns with the well-established matrix constructions in the CS literature. While deterministic constructions are limited, it is well-known that random matrices  (e.g., sub-Gaussian) satisfy the classic RIP with high probability if the measurement bound is met. Using Theorem 1, we can derive the measurement bounds for individual factor matrices, as discussed in Section 4 under *Measurement bounds for classical methods*. Practically, we can generate each factor matrix $\mathbf{H}_i$ using standard CS constructions that ensure the classic RIP, and then build our final KCS measurement matrix $\mathbf{H}$ by their Kronecker product, which our theory guarantees satisfies the required RIP conditions.
>
> **Q1**: The paper introduces three sparsity models, but it does not specify which types of real-world data these models are tailored for. The paper lacks simulations using authentic datasets.
>
> **A3**: Regarding real-world scenarios these models are tailored for, we list the following applications.
> - Standard sparsity and KCS:
>   - Compressive imaging [Rivenson & Stern (2009)].
>   - Hyperspectral imaging [Duarte & Baraniuk (2011a); Li & Bernal (2017)]
>   - Compressible image and video representation and recovery [Friedland et al. (2014)].
> -  Hierarchical sparsity and KCS:
>    - Channel estimation problem for a massive multiple-input multiple-output system [Roth et al. (2020); Wunder et al. (2019)].
> - Kronecker-supported sparsity and KCS:
>   - Image processing [Caiafa & Cichocki (2012); Caiafa & Cichocki (2013); Zhao et al. (2019)]
>   - Wireless communication [He & Joseph (2025a); Chang & Su (2021); Xu et al. (2022)].
>
> We revised Section 1 to clarify these applications. We added two sets of numerical results with model-based validation, relying on well-established, highly representative system models of the intended applications, and demonstrating improvements over baselines. We added
> - hierarchical sparsity evaluation in Section 5 for channel estimation in a wideband massive multiple-input multiple-output system using the channel model in [Haghighatshoar & Caire,
> 2017; Chen & Yang, 2016; Wunder et al., 2019] with accuracy improvement and runtime reduction, and
> - Kronecker-supported sparsity evaluation in Appendix H.4 for channel estimation in a narrowband intelligent reflecting surface-aided system using the channel model in [He & Joseph, 2023; You et al., 2022] with similar accuracy but significant runtime reduction.
>
> **Q3**: The analysis is limited to $I\leq 3$. The performance variations across different $I$ should have been explored.
>
> **A4**: We clarify that our theoretical analysis (Theorem 1 and 2) is not limited to $I\leq 3$.  Particularly, Theorem 2 shows that the error bounds for all three sparsity patterns are related to $I$, implying that error propagates differently when $I$ changes. However, we focus on $I\leq 3$ in our numerical evaluations due to computational feasibility. Since the problem dimension scales exponentially with $I$, for a moderate-sized factor matrix, e.g., 64-by-80 $\mathbf{H}_i$ as in Section 5, evaluating $I\leq 4$ is almost computationally infeasible for traditional methods on our machine. Moreover, in all the practical applications of KCS we considered, $I$ is typically 2 or 3. Yet, investigating the performance variations across different $I$ is important from a theoretical perspective and for potential future applications that may involve higher system orders.
>
> To address your concern and complete our evaluation, besides Figure 3 (fixed $I$, varying $N$), we present additional results with fixed size of $\mathbf{H}_i$’s but varying $I=$ 1 to 4 in Appendix H.3. We see the accurate recovery of MSSBL/MSHTP/MSIHT and that the runtime is exponentially in $I$, matching complexity analysis in Appendix F. It shows the general ability of our MSR to handle higher system order $I$ with reasonable run time.

---

> > ### Comment · Reviewer_64D7 · 2025-11-24
> >
> > I would like to thank the authors for their explanations and additional experiments. However, I have doubts regarding the practical value of KCS in the following two aspects:
> >
> > 1) KCS method has not received sufficient attention in the compressed sensing field due to its high computational complexity. As noted in the paper, the primary advantage of KCS is  to reducing runtime, but this comes at the cost of significantly higher complexity: an exponential multiple compared to traditional methods. Regarding runtime optimization, GPU-based parallel implementations already exist for conventional reconstruction algorithms such as ISTA [1] and OMP [2]. I believe these parallelized algorithms operate much faster than KCS while maintaining considerably lower complexity.
> >
> >      [1] https://github.com/rfeinman/pytorch-lasso
> >
> >      [2] Ariel Lubonja. Efficient batched cpu/gpu implementation of orthogonal matching pursuit for python, arxiv, 2024.
> >
> >
> > 2) The effectiveness and performance of KCS on real-world data remain unclear. As observed, the data used in the simulations are synthetic rather than real-world. I suspect this is because, for real-world data, the hierarchical parameters about sparsity are not easily identifiable, such that the performance of KCS cannot be guaranteed.

---

> ### Author Response · Authors · 2025-11-26
> **Response to reviewer's concerns on practical value of KCS**
>
> **Q1**: KCS method has not received sufficient attention in the compressed sensing field due to its high computational complexity. As noted in the paper, the primary advantage of KCS is to reduce runtime, but this comes at the cost of significantly higher complexity: an exponential multiple compared to traditional methods. Regarding runtime optimization, GPU-based parallel implementations already exist for conventional reconstruction algorithms such as ISTA [1] and OMP [2]. I believe these parallelized algorithms operate much faster than KCS while maintaining considerably lower complexity.
>
> **A1**: We thank the reviewer for the follow-up questions and would like to clarify several misconceptions underlying the reviewer's comments regarding KCS and our MSR algorithms.
>
> *KCS vs. Conventional CS*: KCS is a measuring and sparsifying scheme for multidimensional signals, using Kronecker matrices for dimension-specific measurement. In contrast, the traditional CS scheme uses *unstructured and dense* matrices operating simultaneously on multiple dimensions, which can become infeasible for high-dimensional signals. For an $I$-dimensional signal of size $\bar{N} = N^I$, CS has a memory complexity of $O(N^I)$ while KCS requires only $I$ matrices, each with $N$ columns, leading to a memory complexity of $O(IN)$. This demonstrates the clear practical and storage advantages of KCS.
>
> *KCS and Computational Complexity*: KCS itself does not impose computational complexity; it depends entirely on the recovery algorithm. Claiming that KCS has received limited attention in CS primarily due to its high computational complexity is inaccurate. KCS has been widely studied in multidimensional sparse signal processing. For instance, [Duarte & Baraniuk (2011a)] has many hundreds of citations, showing the broad impact and relevance of KCS. Similarly, the note that "the primary advantage of KCS is reducing runtime at the cost of higher complexity" mischaracterizes the KCS scheme and seems contradictory. The primary goal of KCS is to enable feasible measurement and recovery of high-dimensional signals. Reduced runtime reflects lower, not higher, computational complexity.
>
> *KCS Algorithms and Computational Complexity*: KCS problems can be solved either with conventional CS algorithms or tailored algorithms like our MSR framework. The complexity of both CS and KCS-tailored algorithms scales with the dimension of the signal ($\bar{N} = N^I$). So, KCS’s complexity is not an exponential multiple of standard methods; what appears to be "exponential complexity" is the intrinsic dimensionality of the signal, not a consequence of our MSR. **Our MSR reduces complexity by exploiting the Kronecker matrix structure**, as detailed in Section 3 and Appendix F, which is consistent with the runtime benefits presented in Section 5 and Appendix H.
>
> *Accelerated CS algorithms and MSR*: GPU-parallel implementations of CS solvers (e.g., ISTA, OMP) can accelerate recovery; these methods are *complementary, not competitive* with our MSR. Each MSR subproblem can directly use these solvers (Step 4) for further speedups. To illustrate, suppose a CS solver recovers a $\bar{N}$-dimensional sparse signal from $\bar{M}$ measurements with complexity $O(\bar{M}^a \bar{N}^b)$, for some $a,b \geq 1$. For an $I$-dimensional signal with $\bar{M} = M^I$ and $\bar{N} = N^I$, its complexity is $O(M^{aI} N^{bI})$. In contrast, when MSR uses this CS algorithm in Step 4, the worst-case complexity of MSR is $O(M^aN^b\frac{N^I-M^I}{N-M})$, which is strictly smaller than $O(M^{aI} N^{bI})$ when $I > 1$ and $1<M<N$. When $I=1$, MSR reduces to the considered accelerated CS method (see Appendix F for details).
>
> In short, KCS is specifically designed for multidimensional signals where CS is infeasible. Our MSR framework, tailored to KCS, reduces both time and memory complexity by exploiting the multidimensional structure and integrates seamlessly with any accelerated CS solvers.
>
> **Q2**: The effectiveness and performance of KCS on real-world data remain unclear. As observed, the data used in the simulations are synthetic rather than real-world. I suspect this is because, for real-world data, the hierarchical parameters about sparsity are not easily identifiable, such that the performance of KCS cannot be guaranteed.
>
> **A2**: The effectiveness and performance of KCS on real-world data have been demonstrated in [Duarte & Baraniuk (2011a)]. To illustrate the advantage of our MSR on real data, we have included a new set of simulation results on Foreman video sequence reconstruction in Appendix H.5. This addresses the reviewer’s concern that MSR may not be practical due to hierarchical sparsity parameters. Importantly, this experiment focuses on recovering a sparse vector; there is no hierarchical structure, and the sparsity level does not need to be known in advance, while the good reconstruction accuracy and lower runtime performance of MSR are observed.

---

> > ### Comment · Reviewer_64D7 · 2025-11-28
> >
> > Thank you for your further explanations. I am familiar with compressed sensing but not with KCS. From my current understanding, KCS aims to sense multidimensional signals along each dimension (or a subset of dimensions) to explore the inherent sparsity and correlation within each dimension. Meanwhile, this process allows factorizing the underlying large-sized measurement matrix into a sequence of small ones, thereby reducing sensing complexity. However, it is not guaranteed that high correlation/sparsity exists in every dimension/section. Manually selecting appropriate dimensions seems inconvenient, and an equivalent implementation using deep networks may be a better alternative.
> >
> > In light of the authors' dedicated work, I am inclined to **raise the score**, yet the system does not seem to offer a score revision option. **I kindly request that the AC consider my opinions during the decision-making process.**

---

> ### Author Response · Authors · 2025-11-28
> **Thank you for the positive feedback**
>
> We sincerely thank the reviewer for the continued time and effort, and for the positive endorsement to the AC. We are glad that our clarifications have resolved your concerns.
>
> We fully agree that high correlation/sparsity is not guaranteed in every dimension for all applications, and data-driven deep neural networks are indeed a promising alternative. That said, for applications like those we demonstrated, where sparsity is inherent, our approach remains a compelling and efficient solution.
>
> Regarding the score revision: We understand the current situation regarding system limitations. While your official comment here is already really helpful, if you are willing, could you please explicitly state your intended score in this thread? This would provide a precise record to facilitate the AC's final decision.
>
> Thank you again for helping us improve the paper!

---

### Official Review · Reviewer_27dS · 2025-10-31

**Soundness:** 3
**Presentation:** 3
**Contribution:** 2
**Rating:** 6
**Confidence:** 3

**Summary:**

The paper studies the performance of Kronecker compressed sensing (CS) for three different hierarchical/structured sparsity patterns. New algorithms that mix reshaping of signal, observation, and sensing tensors and joint sparsity algorithms applied to tensor slices allow for improvements in performance versus agnostic baselines.

**Strengths:**

The paper covers theoretical analysis, algorithmic contributions, and numerical performance for several different signal setups.

The analytical results expand over existing contributions in Kronecker CS by proposing and leveraging custom restricted isometry properties, obtaining better scaling laws for the number of measurements necessary from random CS matrix constructions. The results also include the commonly observed tolerance to the presence of noise in the measurements.

The proposed algorithms exploit the structure of Kronecker matrices for CS and sparsity to reduce the computational complexity of recovery vs. their agnostic counterparts.

The numerical comparisons are convincing.

**Weaknesses:**

It is not always clear if these proposed sparsity structures find readily available applications where Kronecker CS is feasible and an improvement over baselines. All experiments are based on synthetic data.

There appears to be no discussion of the role of the sparsity transformation, which implicitly would have to follow a Kronecker product formulation as well.

**Questions:**

Are there applications where the newly proposed hierarchical sparsity structures arise naturally?

---

> ### Author Response · Authors · 2025-11-19
> **Response to reviewer's concerns about real-world motivation and discussion of sparsity transformations**
>
> We sincerely thank the reviewer for the positive comments.
>
> **W1&Q1**: It is not always clear if these proposed sparsity structures find readily available applications where Kronecker CS is feasible and an improvement over baselines. All experiments are based on synthetic data.
>
> **A1**: We list the applications of Kronecker CS (KCS) with different sparsity structures below:
> - Standard sparsity and KCS:
>   - Compressive imaging [Rivenson & Stern (2009)].
>   - Hyperspectral imaging [Duarte & Baraniuk (2011a); Li & Bernal (2017)]
>   - Compressible image and video representation and recovery [Friedland et al. (2014)]
> -  Hierarchical sparsity and KCS:
>    - Channel estimation problem for a massive multiple-input multiple-output system [Roth et al. (2020); Wunder et al. (2019)]
> - Kronecker-supported sparsity and KCS:
>   - Image processing [Caiafa & Cichocki (2012); Caiafa & Cichocki (2013); Zhao et al. (2019)]
>   - Wireless communication [He & Joseph (2025a); Chang & Su (2021); Xu et al. (2022)]
>
> We revised Section 1 to clarify these applications. We added three sets of numerical results with model-based validation, relying on well-established system models that are highly representative of the intended applications, and demonstrating improvements against baselines. We added
> - hierarchical sparsity evaluation in Section 5 for channel estimation in a wideband massive multiple-input multiple-output system using the channel model in [Haghighatshoar & Caire,
> 2017; Chen & Yang, 2016; Wunder et al., 2019], where we observe accuracy improvement and runtime reduction, and
> - Kronecker-supported sparsity evaluation in Appendix H.4 for channel estimation in a narrowband intelligent reflecting surface-aided system using the channel model in [He & Joseph, 2023; You et al., 2022] with similar reconstruction performance but significant runtime reduction,
> - standard sparsity evaluation in Appendix H.5 for real-world Foreman video sequence reconstruction [Duarte & Baraniuk (2011a)], showing similar peak signal-to-noise ratio but significant runtime reduction compared to the benchmark.
>
> **W2**: There appears to be no discussion of the role of the sparsity transformation, which implicitly would have to follow a Kronecker product formulation as well.
>
> **A2**: We assume that sparsity transformation refers to the transformation of the signal to an appropriate basis to obtain its sparse representation, i.e., signal $\mathbf{\theta} =\mathbf{S}\mathbf{x}$ where $\mathbf{x}$ is sparse and matrix $\mathbf{S}$ is the transformation (or sparsifying basis) that maps these sparse coefficients into the signal domain. In KCS, the transformation $\mathbf{S}$ naturally inherits a Kronecker product structure as well.
>
> The primary motivation for KCS is to handle multidimensional signals that can be sparsified in a way that respects their dimension-wise structure. Traditional compressed sensing employs a measurement matrix that operates on all signal dimensions simultaneously, and a sparsifying basis that is unstructured, and sparsifies the signal across all dimensions at once, failing to capture the multidimensional structure. KCS handles the first measurement challenge by using a Kronecker product matrix, performing a dimension-wise measurement, and similarly handles the second challenge by sparsifying the signal dimension-wise [Duarte & Baraniuk, 2011a].
>
> As a concrete example, consider estimating the channel between a base station and a user in a wireless communication system. This channel is inherently multidimensional, since its properties depend on the angular domain of the base station array (one dimension) and the angular domain of the user array (a second dimension). Due to transmission physics, the channel is typically sparse in the angular domain, meaning that there exist only a few strong paths corresponding to a limited number of specific angles at the base station array and the user array. The traditional approach would require a large basis to sparsify the channel. However, using separate transformation matrices for each dimension (the angular domain of the base station and the user) naturally sparsifies the signal in each dimension independently, leading to a Kronecker product sparsifying basis (see Appendix H.4 for details). If the sparsifying basis for each dimension is $\mathbf{S}_i$, combining the measurement model for $\mathbf{\theta}$ as $$\mathbf{y} = (\mathbf{M}_I\otimes \cdots \otimes \mathbf{M}_1)\mathbf{\theta} = (\mathbf{M}_I\otimes \cdots \otimes \mathbf{M}_1)(\mathbf{S}_I\otimes \cdots \otimes \mathbf{S}_1)\mathbf{x} = (\mathbf{M}_I\mathbf{S}_I\otimes \cdots \otimes \mathbf{M}_1\mathbf{S}_1)\mathbf{x},$$ where $\mathbf{M}_i$ is the $i$th dimension measurement matrix. Defining the effective measurement matrix $\mathbf{H}_i =\mathbf{M}_i \mathbf{S}_i$ shows that this case reduces to Equation 1.

---

### Author Response · Authors · 2025-12-01
**Post-rebuttal summary for the AC**

We summarize the revisions and discussions.

**Reviewer 27dS, 64D7, MA64 questioned the applications of KCS and our approach’s usefulness.**
- We revised Sec. 1 for applications of Kronecker compressed sensing (KCS) with considered sparsity patterns.
- We validated the practical utility using three new applications:
  - Well-representative model-based:
    - Sec. 5: wideband massive MIMO channel estimation (hierarchical sparsity).
    - App. H.4: narrowband IRS-aided system channel estimation (Kronecker-supported sparsity).
  - Real-world dataset-based:
    - App. H.5: Foreman video sequence reconstruction (standard sparsity).

Reviewer 27dS did not raise further concerns. Reviewer 64D7 appreciated the added experiments and explanations, and Reviewer MA64 acknowledged our answer.

**Reviewer 27dS questioned whether the sparsity transformation follows a Kronecker structure.**

We confirmed that the Kronecker structure of the sparsity transformation is the natural consequence of dimension-wise sparsifying, leading to the same math model as Eq. 1.

Reviewer 27dS did not raise further questions.

**Reviewer 64D7 requested a comparison to traditional compressed sensing (CS) algorithms.**

We added new results benchmarking our framework against OMP, IHT, HTP, SBL, and Basis Pursuit in App. H.2, showing similar/better performance with lower runtime for structured sparsity, and a balanced option in the standard case.

**Reviewer 64D7 asked for a concrete method to construct measurement matrices that satisfy the enhanced RIP conditions.**

We clarified that our Theorem 1 shows that no specialized construction is needed. Following the common practice with random matrix $\mathbf{H}_i$, and computing their Kronecker product is sufficient.

**Reviewer 64D7 questioned scalability with different system orders $I$.**

We evaluated performance for $I=1$ to $4$ in App. H.3, showing our framework is general and not limited to $I \leq 3$. We also discussed the insight from our Theorem 2 on the performance variations across different $I$.

**Reviewer 64D7 questioned whether KCS is truly beneficial due to high computational complexity (exponential multiple compared to traditional methods) and may even be slower than existing GPU-accelerated traditional CS algorithms.**

We clarified that KCS is a measuring/sparsifying scheme for multidimensional signals, not an algorithmic framework. Its complexity depends on the recovery algorithm, not intrinsically higher. The “exponential” complexity arises from the multidimensional signal size and is present in both KCS and traditional methods. Our framework is a tailored algorithm for KCS, exploiting the Kronecker structure to reduce runtime and complexity (Sec. 5, App. H and F). Finally, GPU-accelerated CS solvers are complementary to our framework, since they can be used seamlessly inside our framework to further speed up recovery. So our framework is practically efficient instead of computationally prohibitive.

Reviewer 64D7 appreciated the added experiments and explanations, and was inclined to raise the score.

**Reviewer Wv5h pointed out confusion regarding figures/sparsity notions.**

We replaced the confusing figure with Fig. 5 on tensor unfolding and added Fig. 6 in App. B to explain sparsity notions.

**Reviewer Wv5h requested clarification of our analysis and Roth et al. (2020).**

We revised App. D to acknowledge the similarity and explicitly discuss the differences, specifically highlighting the tensor operations, the multi-stage measurement view, and its induced practical and theoretical insights.

**Reviewer Wv5h questioned runtime as a metric as it depends on implementation and suggested also using a more objective measure like the number of iterations, because the reviewer spotted a discrepancy between the per-iteration complexity of HiHTP reported in Roth et al. (2020) and in our work.**

We clarified that we focus on runtime because iterative methods trade iterations for recovery quality. We aim to show an equal or better NSE with less runtime. The iteration count alone is not comparable, since per-iteration costs differ across algorithms. Two methods with the same number of iterations can still have very different computational costs. We also adopted the same stopping rule and iteration cap for all algorithms within the same family, e.g., all HTP-based ones, for fairness. The discrepancy is due to a notation confusion in Roth et al. (2020). Upon clarification, our work and Roth et al. (2020) reduce to the same result.

Reviewer Wv5h stated that there is worth in publishing the work and raised the score.

**Reviewer MA64 critiqued the limited applicability of KCS to multidimensional signals.**

We admitted that KCS is not a universal solution but intentionally trades generality for efficiency in multidimensional settings where traditional CS can be infeasible. We also included applications where KCS is readily feasible.

Reviewer MA64 acknowledged our clarification.

---

### Meta-Review · Area_Chair_LBCx · 2026-01-19

**Summary:**

This paper proposes a hierarchical multi-stage recovery (MSR) framework for Kronecker compressed sensing. It leverages the multi-level structure of Kronecker measurement matrices to efficiently recover sparse signals under standard, hierarchical, and Kronecker-supported sparsity models.

- Reviewer 27dS, 64D7, MA64 questioned the applications of KCS and our approach’s usefulness.

- Reviewer 64D7 requested a comparison to traditional compressed sensing (CS) algorithms.

- Reviewer 64D7 questioned whether KCS is truly beneficial due to high computational complexity (exponential multiple compared to traditional methods) and may even be slower than existing GPU-accelerated traditional CS algorithms.

- Reviewer Wv5h questioned the use of runtime as a metric, as it depends on implementation and suggested also using a more objective measure like the number of iterations, because the reviewer spotted a discrepancy between the per-iteration complexity of HiHTP reported in Roth et al. (2020) and in our work.

- Reviewer MA64 critiqued the limited applicability of KCS to multidimensional signals.

**Reviewer Concerns:**

In the rebuttal, the authors have thoroughly addressed the concerns raised by the reviewers, providing clarifications, additional experiments, and detailed explanations.

**Reviewer Scores:**

After the rebuttal, Reviewer 64D7 appreciated the additional experiments and clarifications, indicating an inclination to raise their score. Reviewer Wv5h acknowledged the value of the work for publication and subsequently increased their score.

---

### Decision · Program_Chairs · 2026-01-26

Accept (Poster)